



# Towards a global actual evapotranspiration product for the Copernicus Land Monitoring Service

Radoslaw Guzinski[1], Héctor Nieto[2], José Miguel Barrios[3], Walid Ghariani[1],
Francoise Gellens-Meulenberghs[3], Jan De Pue[3], and Roselyne Lacaze[4]

[1]DHI A/S, Hørsholm, Denmark
[2]ICA-CSIC, Madrid, Spain
[3]RMI, Brussels, Belgium
[4]HYGEOS, Lille, France

**Correspondence:** Radoslaw Guzinski (rmgu@dhigroup.com)

**Abstract.** Copernicus Land Monitoring Service (CLMS) produces biogeophysical maps of the global land surface. The CLMS portfolio so far did not include actual evapotranspiration (ETa), despite it being a direct link between the energy, water and carbon cycles and its importance for global food security, efficient water resources management and weather forecasting. However, a global CLMS ETa product is currently under development and will enter operational production by the end of 2025.

It will have a spatial resolution of 300 m, dekadal (10-daily) temporal resolution, will consist of evaporation and transpiration sub-products and (like all other CLMS products) will be distributed under free and open data policy. It will be based mainly on Copernicus input data with primary satellite imagery coming from the observations of OLCI and SLSTR sensors on board of Sentinel-3 satellites. Such product will fill a gap in currently existing global and operational ETa products, thus satisfying a wide range on potential users' needs. In this paper, we describe the various design choices taken during the development of

the ETa product, ranging from cloud masking and gap-filling, through derivation of biophysical traits, radiation components and weather forcings to spatial sharpening of the land surface temperature observations. Those data were then used to drive two evapotranspiration models: TSEB-PT and ETLook. A prototype implementation of the ETa processing chain was used to produce ETa data across a globally representative range of climatic zones and plant functional types, which was validated against measurements from 104 Eddy Covariance flux tower sites. The resulting overall best root mean squared error (RMSE)

of 0.80 mm/day (relative RMSE of 47%), bias of -0.12 mm/day (relative bias of 7%) and coefficient of determination of 0.84 compare well with a similar global ETa dataset and are encouraging for the upcoming operational production of ETa maps.

## 1   Introduction

Copernicus (https://www.copernicus.eu, last accessed: 01/09/2025) is the Earth observation (EO) program of the European Union that provides free and open access to data acquired by the Sentinel satellites (and other contributing missions) and to

higher level and non-space products through the Copernicus services. One of the main objectives of the program is to bring operational mindset to EO by guaranteeing long-term future continuity of the provided data and predictable planning of future evolution of data and services. The Copernicus Land Monitoring Service (CLMS - https://land.copernicus.eu, last accessed:



01/09/2025) produces a series of qualified global biogeophysical products on the status and evolution of the land surface. The products are used to monitor vegetation, water cycle, energy budget and terrestrial cryosphere. Production and delivery are

carried out in a timely manner and are complemented by the constitution of long-term time series.

Actual evapotranspiration (ETa) has so far not been included in the CLMS portfolio. This will change when the CLMS near-real-time (NRT) ETa product will enter operational production by the end of 2025. Such product will be of importance for many applications and research questions. The ETa is one of the Essential Climate Variables (ECV) as defined by the Global Climate Observing System (Bojinski et al., 2014). It binds different processes occurring at the Earth's surface (related

to energy, water and carbon cycles) and evolves coherently with other CLMS variables and associated products (e.g. vegetation indices, gross primary productivity, soil moisture, surface temperature). Since actual evapotranspiration is a direct proxy of plant water use it can be utilized for consistent irrigation water use monitoring across natural and political boundaries, and is therefore essential for Sustainable Development Goal (SDG) reporting, e.g. of SDG indicators 6.4.1 and 6.4.2 (O'Connor et al., 2020). ETa can also be useful for forest fire risk/spread forecasting (Vidal et al., 1994), drought monitoring (Anderson

et al., 2011), hydrological modelling (Zhang et al., 2020; Larsen et al., 2016), irrigation accounting (Zhang and Long, 2021) and yield modelling (Jurečka et al., 2021; Gómez-Candón et al., 2021). In addition, spatially distributed fields of ET at an adequate spatial resolution can help to improve the weather forecasting (Boone et al., 2025), in particular in irrigated fields in semi-arid climates where the additional water supply in the soil affects the surrounding microclimate and thus the boundary layer conditions (Udina et al., 2024; Lunel et al., 2024).

One of the initial primary users of the CLMS ETa product will be the Food and Agriculture Organisation (FAO) of the United Nations. FAO has been producing and disseminating ETa datasets, including a global product, through its "WAter Productivity through Open-access of Remotely sensed derived data" (WaPOR - https://www.fao.org/in-action/remote-sensing-for-water-p roductivity/en - last accessed 30/07/2025) project. The WaPOR data has been used in multiple applications such as improving water use productivity or monitoring agricultural practices and water consumption (Chukalla et al., 2022; Hajirad et al., 2023;

Seijger et al., 2023). However, production of global operational satellite-based datasets is not in the core mandate of the FAO and therefore long-term continuity of the WaPOR project cannot be ensured. For this reason, FAO expressed strong interest to the European Commission for the introduction of a global ETa dataset in the CLMS portfolio.

In order to satisfy this wide range of potential users' needs, and for consistency with other global CLMS products, the CLMS ETa product will have a spatial resolution of 300 m and a dekadal temporal resolution. Other product requirements are shown

in Table 1. Those requirements also closely match the specifications of the WaPOR global ETa dataset and fill a gap in currently available satellite-based ETa products. Apart from WaPOR (the long-term continuity of which is not guaranteed), global operational ETa datasets with closest matching spatio-temporal specifications are the MODIS and VIIRS ETa products (Román et al., 2024). They have a higher temporal resolution (8-days) but lower spatial resolution (500 m) and use a modeling approach which does not make direct use of land surface temperature (LST) measurements (Mu et al., 2011). Another operational and

global product which utilizes MODIS and VIIRS data is produced by United States Geological Survey using SSEBop energy balance model (Senay et al., 2020) with dekadal temporal resolution and 1 km spatial resolution. Other ETa datasets have either much lower spatial and temporal resolutions (e.g. ETa product based on geostationary observations produced by the





**Table 1.** Specifications (requirements) of the Copernicus Land Monitoring Service actual evapotranspiration product.

| Property | Specification |
|---|---|
| Spatial resolution | 300 m |
| Temporal resolution | Dekadal (dekad: days 1–10, 11–20, 21–end of month) |
| Spatial coverage | Global |
| Temporal coverage | 2019 – Near Real Time |
| Timeliness | Within 2 days (optimally 1 day) after the end of each dekad |
| Products | - Actual evapotranspiration [mm/day]<br>- Soil evaporation [mm/day]<br>- Canopy transpiration [mm/day]<br>- Latent heat fluxes [W/m$^2$]<br>- Sensible heat flux [W/m$^2$] |

EUMETSAT Satellite Application Facility on Land Surface Analysis - Barrios et al. (2024)), do not have global coverage (e.g. OpenET - Melton et al. (2022)) or are not produced operationally and in NRT (e.g. ETMonitor - Zheng et al. (2022)).

Preparatory activities required to develop an operational CLMS ETa product recommended that two ET modelling frameworks should be further investigated. The first one is the Sen-ET framework (Guzinski et al., 2020, 2021) developed to model ETa with Copernicus data at various spatial scales and using the Two-Source Energy Balance Priestley-Taylor (TSEB-PT) ET model (Norman et al., 1995; Kustas and Norman, 1999; Anderson et al., 2024). The second is the WaPOR framework developed by FAO through the WaPOR project and using the ETLook ETa model (Bastiaanssen et al., 2012). Both models, although conceptually different, estimate evaporation and transpiration and use LST as one of core input forcings.

This paper aims to give an overview of the design choices made for the upcoming CLMS ETa product. In Section 2 , we present the design of the CLMS ETa processing chain, starting with input data sources through their pre-processing to brief description of the two ETa models and method used for gap-filling of the resulting maps. This is followed by Section 3 in which a prototype ETa dataset, produced with both TSEB-PT and ETLook, is compared against measurements from 104 Eddy Covariance (EC) flux tower sites. In Section 4, we compare the results with other similar ET datasets, justify the design choices described in Section 2 and outline suggestions for product improvement. Finally, conclusions are presented in Section 5.

## 2 Data and methods

### 2.1 Input data sources

Modelling actual evapotranspiration with satellite observations and TSEB-PT or ETLook models is a complex task requiring diverse set of input forcing variables (Table 2). Those are derived from shortwave optical imagery needed to estimate biophys-



**Table 2.** Input forcing variables for Copernicus Land Monitoring Service actual evapotranspiration product. PAR indicates photosynthetically active radiation part of the spectrum (400-700 nm) while NIR indicates near infra-red part of the spectrum (700-2500 nm).

| Group | Variable | Units |
|---|---|---|
| Biophysical traits | Leaf Area Index (LAI) | $m^2 \ m^{-2}$ |
| | Fraction of LAI that is green | – |
| | Mean leaf inclination angle (LIDF) | º |
| | Leaf PAR reflectance | – |
| | Leaf PAR transmittance | – |
| | Leaf NIR reflectance | – |
| | Leaf NIR transmittance | – |
| Soil properties | Soil PAR reflectance | – |
| | Soil NIR reflectance | – |
| Boundary condition | Radiometric surface temperature | K |
| Weather | 100m air temperature | K |
| | 100m wind speed | $m \ s^{-1}$ |
| | 100m water vapour pressure | hPa |
| | Surface pressure | hPa |
| | Direct PAR irradiance | $W \ m^{-2}$ |
| | Diffuse PAR irradiance | $W \ m^{-2}$ |
| | Direct NIR irradiance | $W \ m^{-2}$ |
| | Diffuse NIR irradiance | $W \ m^{-2}$ |
| | Longwave irradiance | $W \ m^{-2}$ |
| | Daily shortwave irradiance | $W \ m^{-2}$ |
| | Daily reference ET | $mm \ day^{-1}$ |
| | Daily precipitation | $mm \ day^{-1}$ |
| Canopy structure | Canopy height | m |
| | Fractional cover of clumped canopy | – |
| | Canopy width to height ratio | $m \ m^{-1}$ |
| | Effective leaf size | m |
| | Maximum stomata conductance | $m \ s^{-1}$ |

ical properties of the surface (e.g. leaf area index - LAI and albedo), thermal infrared observations of land surface temperature (LST) which is the boundary condition for the land surface - air energy exchange and a proxy for root-zone soil moisture, weather forcings, which drive (e.g. solar irradiance) and modulate (e.g wind speed) the energy exchange between the land surface and the air, and ancillary data (e.g. digital elevation models and canopy height maps) that cannot be derived from the

other data sources.

Copernicus products should, to the largest extent possible, be based on other Copernicus data. This is to ensure the free and open license conditions, long-term future continuity and consistency across the CLMS portfolio. In case of the CLMS ETa



**Table 3.** Input data sources for the Copernicus Land Monitoring Service actual evapotranspiration near-real-time product.

| Data type | Sensor / model | Product name | Source |
|---|---|---|---|
| Thermal imagery | Sentinel-3 SLSTR | SL_2_LST | Copernicus Data Space Ecosystem |
| Shortwave imagery | Sentinel-3 OLCI + SLSTR | cgl_TOC v2.3.4 | Copernicus Land Monitoring Service |
| Meteorological data | ECMWF IFS | CAMS global atmospheric composition forecasts | Copernicus Atmosphere Monitoring Service |
| Landcover map | PROBA-V (Sentinel-2) | Global Dynamic Land Cover 2019 | Copernicus Land Monitoring Service |
| Digital elevation model | TanDEM-X | Copernicus DEM | Copernicus Data Space Ecosystem |
| Canopy height map | GEDI LiDAR + Sentinel-2 | ETH_GlobalCanopyHeight_10m_2020_version1 | ETH Zurich |

product, this means observations from optical (both shortwave and thermal infrared) sensors on board of Sentinel-3 satellites and Copernicus Digital Elevation Model (DEM - European Space Agency and Airbus (2022)), both available from the Coper-
nicus Data Space Ecosystem (https://dataspace.copernicus.eu/ - last accessed 25/08/2025), and products provided by CLMS and Copernicus Atmosphere Monitoring Service (Peuch et al., 2022) (Table 3). The only exception is the canopy height map, which is derived from fusion of GEDI LiDAR measurements and Sentinel-2 imagery (Lang et al., 2023).

The subsections below describe the pre-processing methods selected to convert the input data from Table 3 into ET model input forcing variables shown in Table 2.

## 2.2 Sentinel-3 cloud-masking and gap-filling

Biophysical characterization of land-surface is based on imagery obtained by two sensors on-board Sentinel-3 satellites: short-wave Ocean and Land Colour Intrument (OLCI); and combined shortwave and thermal infrared Sea and Land Surface Temperature Radiometer (SLSTR). Both of those sensors operate in the optical spectral domain, which is blocked by clouds and therefore cloud detection and masking needs to be performed before further analysis of this data.

Atmospherically corrected and geolocated Top-Of-Canopy (TOC) reflectances derived from shortwave optical detectors on OLCI and SLSTR sensors are generated and distributed by the CLMS (Copernicus Land Monitoring Service, 2025b). The cloud mask of this product is based on quality flags from Level 1 SLSTR and OLCI products and IDEPIX approach (Wevers et al., 2022). No further cloud masking is performed during the ETa production and instead the recommendations on using the annotation flags from Section 5.1 of Product User Manual (Copernicus Land Monitoring Service, 2025d) are followed.

Land Surface Temperature (LST) product (SL_2_LST___) is a Level 2 product based on thermal observations by the SLSTR sensor. It comes with quality layers indicating the presence of clouds. As recommended in the Copernicus Sentinel-3 SLSTR Land User Handbook (https://sentiwiki.copernicus.eu/__attachments/1672112/OMPC.ACR.HBK.002-Sentinel3SLST RLandHandbook2024-1.4.pdf, last accessed 01/08/2025), the probabilistic cloud mask is used during ETa production. This cloud mask uses a semi-Bayesian approach by estimating a probability of a clear-sky using radiative transfer modelling and
meteorological conditions at the time of satellite overpass and observational climatology.

The TOC reflectance is used to derive land surface biophysical traits (Sec 2.3) and albedo (Sec 2.4) and to sharpen the LST (Sec 2.5). The reflectance values, as well as biophysical traits, usually show a clear seasonal cycle and spatial similarity, and





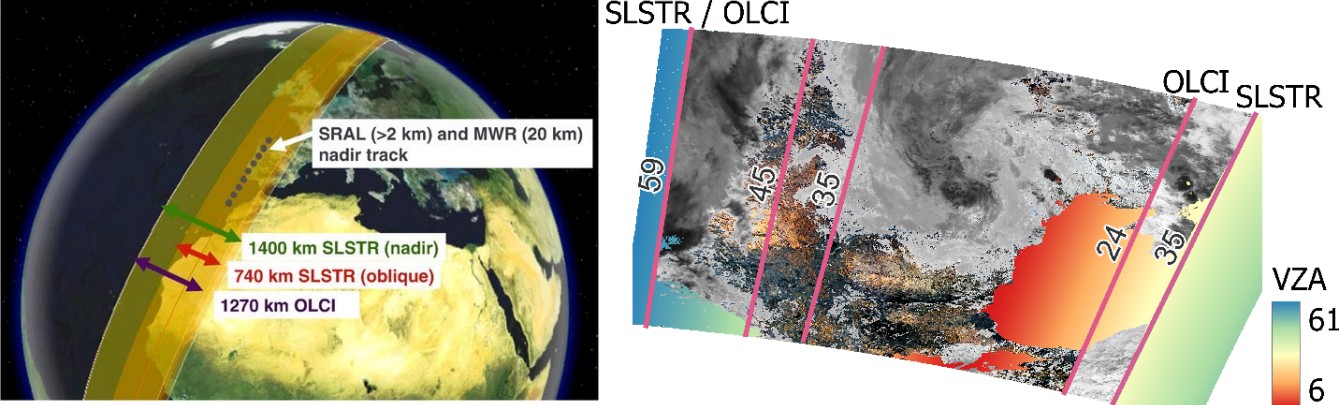

**Figure 1.** Widths and locations of swaths of Sentinel-3 OLCI and SLSTR instruments (courtesy of Donlon et al. (2012)) (left) and maximum view zenith angles (VZA) of those swaths overlaid on an example OLCI (colour) and SLSTR (greyscale) image (right). Also indicated are potential limits of VZA on the western side of the swath (45° and 35°).

have same or similar values in both cloudy and sunny conditions (e.g. leaf area index does not change day to day depending on cloudiness). Therefore, gaps in this data are highly suitable for filling using spatio-temporal gap-filling, e.g. StarFM (Gao
et al., 2006), and smoothing methods, e.g. Whittaker-Eilers smoother used by WaPOR (Eilers, 2003). On the other hand, LST is highly variable in both time (the temperature can change significantly at short time intervals even though it also has a strong seasonal cycle) and space. In addition, LST under clouds is different to LST in clear-sky conditions and using gap-filled values can lead to energy imbalance at the land surface. Therefore, LST is usually not gap-filled, especially if it is to be used as input into ETa models.

In the case of the CLMS ETa product, we are using TOC reflectance and LST observations acquired by sensors onboard the same satellite platform and thus observing the land surface (and the clouds) at the same time. Therefore, since LST is not gap-filled, there is also no need to gap-fill TOC reflectance acquired at the same time and location. However, the Sentinel-3 OLCI sensor has a swath width of 1270 km, while the nadir pointing scan of SLSTR sensor has a swath width of 1400 km (Donlon et al., 2012). Both sensors are tilted to reduce the sun glint effect and therefore larger part of the swath is located
westwards of the satellite orbit path (Fig. 1 left).

The sensors are positioned such that their swaths align at the western edge, while on the eastern edge the coverage of OLCI ends at VZA of 24° while coverage of SLSTR ends at VZA of 35° (Fig. 1 right). When modelling ETa, LST observations acquired at high VZA (e.g. above 45° as in the Sen-ET approach (Guzinski et al., 2020)) are usually omitted due to increased uncertainty caused by longer atmospheric path, LST anisotropy and larger pixel footprint. However, SLSTR observations from
the eastern edge of the swath with VZA between 24° and 35° could still potentially be used if OLCI data for that part of the swath was gap-filled.

To assess whether it is worth to gap-fill the missing part of OLCI's eastern swath, and whether maximum VZA should be limited to 45° or 35° (threshold mission requirement value for planned Copernicus Land Surface Temperature Monitoring

 

mission (Koetz et al., 2021)), we performed an analysis of number of monthly cloud-free Sentinel-3 LST acquisitions over

Europe and Africa (Fig. 2). This analysis was performed using 5 years of data (2020–2024) for months of March, June, September and December (apart from March 2020 and December 2024) and while limiting VZA of the western edge of swath to either 45° or 35° and eastern edge to either 35° or 24°.

Figure 2 shows a clear seasonal trend in the number of cloud-free SLSTR observations. In June and September, most of Europe has sufficient number of cloud-free observations for each dekad (at least 2 but often more than 4), while in March and,

especially in December, there are areas where less than two or even less than one cloud free observations are available in each dekad. In the equatorial region of Africa, there are predominantly cloudy conditions throughout the year but they shift north in June and September and south in December and March. The difference between number of monthly cloud-free observations with eastern VZA limited to 35° or 24° is in large majority of cases less than 2, except in the Sahara region and southern Africa where there is sufficient number of observations in either case. The difference between limiting western swath to either 45° or

35° is more pronounced and can push some regions to having less than 1 observation per dekad.

Based on this analysis, it was decided to limit the western VZA to 45° and eastern VZA to 24°. This means that the extra processing that would be required to gap-fill the OLCI swath is avoided. Even though gap-filling of input imagery will not be performed, the filling of cloud gaps in the produced ETa maps is required to fulfill the requirements of the ETa product. Therefore gap-filling of outputs will be performed as described in Section 2.9.

**2.3   Biophysical traits**

A sound and generalizable approach to obtain the biophysical traits required by the ETa models is the inversion of canopy radiative transfer models (RTM), such as the combined PROSPECT (Jacquemoud and Baret, 1990; Feret et al., 2008; Féret et al., 2017, 2021) and 4SAIL (Verhoef et al., 2007) RTMs. Together those models can simulate the canopy spectra from 400 to 2500 nm at every nm based on the inputs listed in Table 4 (Jacquemoud et al., 2009). One computationally efficient inversion

method consists of training a regression model based on large number of PROSPECT and 4SAIL simulations (Weiss et al., 2000, 2002; Verrelst et al., 2012). Then this regression model is applied to the spectral imagery to obtain the biophysical traits maps.

In case of CLMS ETa, we are using a vectorized version of PROSPECT-D (Féret et al., 2017) and 4SAIL models (Nieto, 2025). This vectorized version allows us to run the RTMs very efficiently and thus enables us to generate specific regression

models for each scene, given their actual observation and illumination conditions. This contrasts with other similar algorithms, such as the BiophysicalOp in the Sentinel-2 toolbox of Sentinel Application Platform (Djamai et al., 2019), which applies the same generic hybrid inversion approach to all scenes. The biophysical processing framework is summarized as:

1. Estimating the proportion of spectral direct and diffuse irradiance using the 6S model (Vermote et al., 1997) and mean sun zenith and azimuth angles together with mean aerosol optical thickness and total column water vapor of Sentinel-3

TOC scene.

2. Generating ca. 40000 PROSPECT-D+4SAIL simulated spectra by:





**Figure 2.** Mean number of monthly cloud-free SLSTR observations for period 2020-2024 (apart from March 2020 and December 2024) for March (top-left), June (top-right), September (bottom-left) and December (bottom-right). The titles of the sub-plots indicate view zenith angle (VZA) limits on the western and eastern edges of the swath (e.g. W45–E35 indicates a VZA limit of 45° on the western edge and 35° on the eastern edge). The last column of each panel shows the difference in cloud-free observations per month between limiting eastern swath to either 35° or 24°. Less than 3 observations per month means on average less than one observation per 10-day aggregation period (dekad), less than 6 per month means less than 2 per dekad, less than 12 per month means less than 4 per dekad.





**Table 4.** List of biophysical traits and ancillary information required by PROSPECT-D+4SAIL radiative transfer models together with range of values which are simulated or set during the simulation. Ranges are based on information extracted from the LOPEX database (Hosgood et al. 1993) and the Sentinel-2 Biophysical ATBD (Weiss et al., 2020)

| Type | Variable | Units | Lower range | Upper range |
|---|---|---|---|---|
| Leaf trait | Leaf structure parameter | – | 1 | 3 |
| | Chlorophyll a+b concentration (Cab) | $\mu g\,cm^{-2}$ | 0 | 110 |
| | Carotenoids concentration (Car) | $\mu g\,cm^{-2}$ | 0 | 30 |
| | Antocyanins concentration (Ant) | $\mu g\,cm^{-2}$ | 0 | 40 |
| | Brown pigments (Cbrown) | arbitrary | 0 | 2 |
| | Dry matter content (Cm) | $g\,cm^{-2}$ | 0.0017 | 0.0031 |
| | Leaf water content (Cw) | $g\,cm^{-2}$ | 0 | 0.0525 |
| Canopy trait | Total LAI | $m^2\,m^{-2}$ | 0 | 8 |
| | LIDF: Campbell (1990) mean leaf angle | ° | 20 | 80 |
| | Hotspot parameter | – | 0.05 | 1 |
| Ancillary inputs | Soil spectrum | – | ECOSTRESS spectra library | |
| | Direct/diffuse irradiance | – | Scene mean | |
| | Solar angles | ° | Scene mean | |
| | Sensor angles | ° | Scene mean | |

(a) Creating the same amount of Monte Carlo random samples (Saltelli et al., 1999) of biophysical traits and observation angles, based on prescribed plausible ranges listed in Table 4.

(b) Running PROSPECT-D+4SAIL for each of these samples together with the generated proportion of spectral direct and diffuse irradiance and the mean solar angles of the scene.

(c) Convolving the simulated narrowband spectra using the Sentinel-3 OLCI and SLSTR spectral response functions in order to obtain a set of 40000 simulated Sentinel-3 TOC reflectances.

3. Training a random forest regression in which the dependent variables are the randomly generated biophysical traits and the explicative variables the correspondent simulated TOC reflectances and observation angles.

4. Applying the random forest regression to the actual TOC reflectance scene.

As outputs we obtain 8 products: leaf area index (LAI), Campbell (1990) mean leaf angle, leaf chlorophyll a+b concentration, leaf carotenoids concentration, leaf antocyanins concentration, brown pigments, leaf dry matter content, and leaf water content. The retrieved leaf pigments concentration gives us an idea about the canopy greenness, and we have observed an increase of Sentinel-3 estimated antocyanins concentration during vegetation curing in summer and the leaf senescence in fall. Indeed antocyanins are red pigments and thus may become dominant over other leaf pigments in those situations (Féret et al., 2017).





For that reason, we can express the fraction of LAI that is green ($f_g$) using an empirical piecewise linear relation to the antocyanins ($Ant$):

$$f_g = \begin{cases} 1, & \text{if Ant} \leq 5\mu\text{g cm}^{-2} \\ 1 - 0.8\frac{\text{Ant}-5}{20}, & \text{if Ant} > 5\mu\text{g cm}^{-2} \text{ and Ant} \leq 25\mu\text{g cm}^{-2} \\ 0.2, & \text{if Ant} > 25\mu\text{g cm}^{-2} \end{cases} \tag{1}$$

## 2.4 Albedo and net shortwave radiation

Net radiation ($R_n$) provides the energy that drives all other energy fluxes at the land-surface (including ET) and can be approximated as:

$$R_n = S_n + L_n = S^{\downarrow}(1-\alpha) + \epsilon\left(L^{\downarrow} - \sigma\text{LST}^4\right) \tag{2}$$

where $S_n$ ($S^{\downarrow}$) and $L_n$ ($L^{\downarrow}$) are the shortwave and longwave net radiation (incoming irradiances) respectively, $\alpha$ and $\epsilon$ are surface albedo and emissivity, respectively, $LST$ is the Land Surface Temperature, and $\sigma \approx 5.67 \times 10^{-8}$ (W m$^{-2}$ K$^{-4}$) is

the Stefan-Boltzmann constant. $\alpha$ and $\epsilon$ (as well as LST) can be estimated, with a certain degree of accuracy, from Earth Observation (EO) data.

Considering the larger magnitude of shortwave irradiance ($S^{\downarrow}$) compared to the longwave irradiance ($L^{\downarrow}$), and the fact that $L_n$ is usually computed internally by each ET model, we will focus on method for deriving $S_n$. In particular, this study concentrates on the canopy and leaf properties that influence albedo and radiation partitioning between soil and canopy. The

albedo ($\alpha$) is defined as the proportion of incident shortwave radiation that is reflected by the surface. The shortwave net radiation ($S_n$) is therefore the balance between the incident shortwave irradiance ($S^{\downarrow}$) and the reflected shortwave radiance ($S^{\uparrow} = \alpha S^{\downarrow}$)

The spectral properties of the surface are key in determining the albedo. Leaves, due to their photosynthetic activity, absorb a large proportion of light due to the presence of chlorophylls, as well as other leaf pigments. On the other hand, soils can

have a large range of albedo values, depending on their mineral composition, texture and topsoil moisture. Therefore, the albedo of a vegetated surface, and also radiation partitioning between soil and canopy, will depend not only on leaf chlorophyll concentration but also on canopy density and, in a lesser degree, on the soil albedo in situations of sparse vegetation or initial growth stages. Indeed, most of the Earth's surface show certain anisotropic behaviour when reflecting radiation i.e. it scatters different amounts of radiation depending on the scattering direction. Vegetation, as it is mainly composed by an array of leaves,

is also affected by this anisotropic behaviour. Therefore, plants will reflect radiation differently depending on their structural characteristics as well as the illumination geometry (i.e. the solar position) and the scattering direction (i.e. the sensor position), changing their albedo with time.

To compute the net shortwave radiation and its partitioning between the canopy ($Sn_C$) and the soil ($Sn_S$) the model of Campbell and Norman (1998) is used. The key aspect of this model is the calculation of the transmitted shortwave radiation





through the canopy ($\tau_C$), which is wavelength dependent due to vegetation absorbing a greater portion of photosynthetically active radiation (PAR - 400-700 nm) than near infra-red (NIR - 700-2500 nm) wavelengths. $\tau_C$ is partitioned into two components (direct/diffuse) and in two spectral band (PAR/NIR).

$$
\begin{aligned}
S_{n,C} = &\,(1 - \tau_{C,DIR,PAR})(1 - \rho_{C,DIR,PAR})PAR_{DIR}+\\
&(1 - \tau_{C,DIR,NIR})(1 - \rho_{C,DIR,NIR})NIR_{DIR}+\\
&(1 - \tau_{C,DIF,PAR})(1 - \rho_{C,DIF,PAR})PAR_{DIF}+\\
&(1 - \tau_{C,DIF,NIR})(1 - \rho_{C,DIF,NIR})NIR_{DIF}
\end{aligned}
\tag{3a}
$$

$$
\begin{aligned}
S_{n,S} = &\,\tau_{C,DIR,PAR}(1 - \rho_{S,PAR})PAR_{DIR}+\\
&\tau_{C,DIR,NIR}(1 - \rho_{S,NIR})NIR_{DIR}+\\
&\tau_{C,DIF,PAR}(1 - \rho_{S,PAR})PAR_{DIF}+\\
&\tau_{C,DIF,NIR}(1 - \rho_{S,NIR})NIR_{DIF}
\end{aligned}
\tag{3b}
$$

where $\rho_{C,DIR}$ is the canopy directional-hemispherical reflectance, $\rho_{C,DIF}$ is the canopy bihemispherical reflectance, $\tau_{C,DIR}$ canopy directional-hemispherical transmittance, $\tau_{C,DIF}$ canopy bihemispherical transmittance, which depend on leaf spectral properties (absortance) and canopy structure (LAI and Campbell (1990) leaf inclination distribution parameter). On the other
hand, $\rho_S$ is the soil bihemispherical reflectance. In all cases $\tau$ and $\rho$ are separated between the PAR and NIR regions of the solar spectrum. The calculation of canopy tranmittances and reflectances is shown in Appendix A.

Campbell RTM requires the leaf absorptance ($\zeta = 1 - \rho - \tau$) as input for calculating canopy transmittance and reflectance. To estimate this parameter, we are using the leaf traits retrievals from the biophysical processor described in Section 2.3. With the information of these traits, we can run the PROSPECT-D leaf RTM in forward mode to get a spectral reflectance ($\rho$)
and transmittance ($\tau$) that could then be integrated to the required broadband regions. However, running PROSPECT-D for a large array of pixels is too computationally expensive. For that reason, a PROSPECT-D emulator (Guzinski et al., 2021) was developed (Rivera et al., 2015): we generated broadband leaf reflectances and transmittances in the PAR and NIR from a large range of PROSPECT-D simulations, covering all plausible range of leaf traits from Table 4, which was then used to train a random forest model that relates the leaf traits to the broadband PAR and NIR leaf reflectance and transmittance. Figure 3
depicts the importance that each leaf trait has on the broadband reflectance emulator, showing that the most important pigment is the chlorophyll a+b concentration, due to its strong absorption of PAR radiation, followed by the leaf dry matter and water contents, which are the main absorbers of the NIR/SWIR radiation.

Soil reflectance has a smaller role in most situations, since the radiation reaching the ground is smaller due to the interception of light by the canopy. However, in sparse and semi-arid conditions, where vegetation is scarce or even not present, soil albedo
plays a significant role. Furthermore, these semi-arid areas are usually characterized by brighter soils since they are composed by sands, salts and with a small fraction of organic matter. For that reason, Guzinski et al. (2023) developed a method to unmix the broadband soil reflectance ($\rho_{soil,\lambda}$) from the TOC reflectance:



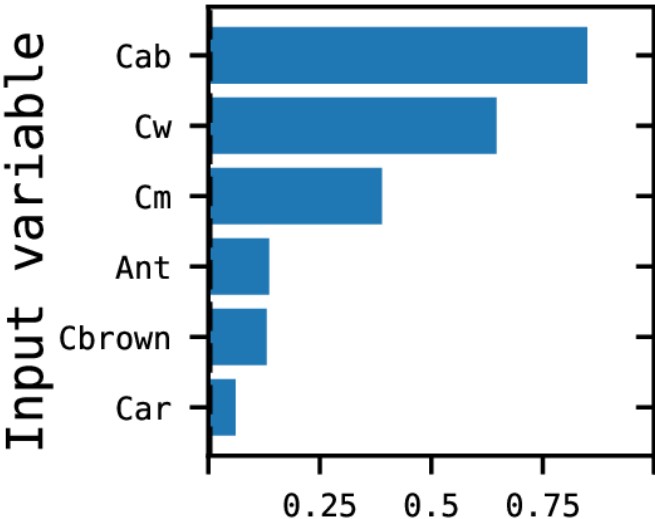

**Figure 3.** Mean permutation importance for leaf biochemical components in the PROSPECT-D leaf radiative transfer emulator. Full variable names can be found in Table 4.

$$\rho_{soil,\lambda} = \frac{\rho_{surface,\lambda} - [1 - P_0(\theta_v)]\,\rho_{leaf,\lambda}}{P_0(\theta_v)} \tag{4}$$

where $P_0(\theta_v)$ is the canopy gap fraction at the satellite observation angle $\theta_v$ computed by the Beer-Lambert law ($e^{-\kappa(\theta_v)\mathrm{LAI}}$),
$\rho_{surface,\lambda}$ is the broadband surface reflectance, and $\rho_{leaf,\lambda}$ is the leaf broadband reflectance computed previously.

In order to convert the narrowband reflectances of the TOC product into broadband reflectances we have applied the approach proposed by Liang (2001), who derived a linear regression model between the satellite spectral bands and the broad bands corresponding to the PAR region (400–700 nm), NIR region (700–2500 nm) and the solar spectrum (SW - 400–2500 nm). Similarly to Liang (2001), to overcome the limited amount of *in situ* measurements of albedo, we again made use of the
PROSPECT-D+4SAIL canopy RTM coupled with the 6S atmospheric RTM (Vermote et al., 1997) to run a large number of simulations covering all plausible illumination, atmospheric and canopy conditions. Once the RTM simulations are performed (at 1 nm step), we convolved the results to both Sentinel-3A and Sentinel-3B specific spectral response functions, and integrated these simulated spectra to the broadband regions defined above. Then, one multivariate linear regression model per broadband region and per platform is fitted with the broadband reflectances as dependent variables and the Sentinel-3 simulated TOC
reflectances as explanatory variables. The resulting coefficients are shown in Table 5, showing the weight of each Sentinel-3 band has on the broadband reflectances. It is worth noting that the coefficients are consistent with the expected behaviour, since the spectral bands located in the PAR region have no influence on the NIR broadband reflectances, nor the spectral bands in the NIR/SWIR influence on the PAR broadband model.



**Table 5.** Narrowband to Broadaband conversion coefficients for Sentinel-3A and Sentinel-3B platforms

| Platform | BAND | O2 | O3 | O4 | O5 | O6 | O7 | O8 | O9 | O10 | O11 | O12 | O16 | O17 | O18 | O21 | S5 | S6 |
|---|---|---|---|---|---|---|---|---|---|---|---|---|---|---|---|---|---|---|
| Sentinel-3A | SW | 0.10 | -0.18 | 0.23 | 0.09 | 0.08 | 0.03 | 0.13 | -0.41 | 0.32 | 0.04 | 0.09 | 0.01 | 0.21 | -0.13 | 0.19 | 0.12 | 0.02 |
| | PAR | 0.10 | 0.14 | -0.01 | 0.24 | 0.17 | 0.24 | 0.15 | -0.26 | 0.21 | 0.00 | 0.00 | 0.00 | 0.00 | 0.00 | 0.00 | 0.00 | 0.00 |
| | NIR | 0.00 | 0.00 | 0.00 | 0.00 | 0.00 | 0.00 | 0.00 | 0.00 | 0.00 | 0.04 | 0.17 | 0.00 | 0.55 | -0.39 | 0.40 | 0.19 | 0.07 |
| Sentinel-3B | SW | 0.10 | -0.18 | 0.23 | 0.09 | 0.08 | 0.03 | 0.14 | -0.42 | 0.32 | 0.04 | 0.09 | 0.01 | 0.21 | -0.13 | 0.19 | 0.12 | 0.02 |
| | PAR | 0.10 | 0.14 | -0.01 | 0.25 | 0.17 | 0.24 | 0.16 | -0.28 | 0.21 | 0.00 | 0.00 | 0.00 | 0.00 | 0.00 | 0.00 | 0.00 | 0.00 |
| | NIR | 0.00 | 0.00 | 0.00 | 0.00 | 0.00 | 0.00 | 0.00 | 0.00 | 0.00 | 0.04 | 0.17 | 0.00 | 0.55 | -0.38 | 0.40 | 0.19 | 0.07 |

## 2.5 Land surface temperature sharpening

The CLMS ETa product specification states a spatial resolution of 300 m. However, the spatial resolution of the SLSTR LST product is 1 km. Therefore, a method to perform thermal sharpening of the LST is required. Most such methods are based on machine learning approaches, of various complexities, which capture the relationships between shortwave spectral reflectance (and possibly other ancillary data) and the LST. The shortwave reflectance is of higher spatial resolution and is resampled to the resolution of LST to be used as explanatory variables for model training. Once trained, the model is applied to the reflectance

at its original resolution to obtain a representation of LST at that resolution. In most approaches, this is followed by a bias correction step to ensure conservation of energy between the LST maps at both the original and sharpened spatial resolutions.

Both the Sen-ET and WaPOR (version 3) modelling frameworks use a Data Mining Sharpener (DMS) thermal sharpening approach (Gao et al., 2012) as implemented in the pyDMS Python package (https://github.com/radosuav/pyDMS, last accessed 30/07/2025). It is a quite complex method, which however works well even for sharpening by a ratio of 50 i.e.

sharpening Sentinel-3 LST from 1 km to 20 m using Sentinel-2 reflectance (Guzinski and Nieto, 2019; Guzinski et al., 2023; Sánchez et al., 2024). Based on this work, we are applying it to sharpen Sentinel-3 LST using CLMS TOC product with 300 m spatial resolution. The DMS regression models are trained on the whole Sentinel-3 TOC tile (10°by 10°) as well as on subsets of 30 by 30 LST pixels in a moving window fashion.

In Sen-ET framework all relevant reflectance bands from Sentinel-2 or Sentinel-3 (depending on the target spatial resolution)

covering visible, near-infrared and shortwave-infrared parts of the spectrum were used as explanatory variables. For Sentinel-3 this means the following 17 bands from CLMS TOC product: Oa02, Oa03, Oa04, Oa05, Oa06, Oa07, Oa08, Oa09, Oa10, Oa11, Oa12, Oa16, Oa17, Oa18, Oa21, S5N, S6N. This configuration of variables, in addition to DEM and cosine of solar zenith angle, is called "DMS - Reflectance" in Section 4.3.2. During the WaPOR project, feature engineering was performed and spectral bands were converted to indices before evaluating their usefulness in DMS. This resulted in following 11 spectral

bands, reflectance indices and DEM-related products being used as explanatory variables: Oa04 (blue), Oa17 (NIR), Modified Normalized Difference Water Index (MNDWI), Plant Senescence Reflectance Index (PSRI), Normalized Difference Moisture Index (NMDI), Visible Atmospherically Resistant Index Red Edge (VARI_RED_EDGE), Bare Soil Index (BI), elevation, cosine solar zenith angle, aspect and slope. More details and the list of evaluated indices are available in the WaPOR wiki (https://bitbucket.org/cioapps/wapor-et-look/wiki/Intermediate_Data_Components/LST, last accessed: 22/07/2025).





**Table 6.** List of required CAMS global atmospheric composition forecast fields and their topographic correction status.

| Variable | Unit | Topographic correction |
|---|---|---|
| 10 m u-component of wind | m s$^{-1}$ | No |
| 10 m v-component of wind | m s$^{-1}$ | No |
| 2 m dewpoint temperature | K | Yes – altitude |
| 2 m temperature | K | Yes – altitude |
| Geopotential height | m$^2$ s$^{-2}$ | No |
| Surface pressure | Pa | Yes – altitude |
| Total aerosol optical depth at 550 nm | – | No |
| Total column water vapour | kg m$^{-2}$ | No |
| Surface solar radiation downwards | J m$^{-2}$ | Yes – altitude and orientation |
| Surface thermal radiation downwards | J m$^{-2}$ | No |
| Total precipitation | m | No |

This combination of explanatory variables is called "DMS - WaPOR" in Section 4.3.2. Finally, since we do not expect strong influence of aspect and slope on LST those two variables were removed from the WaPOR list and the resulting combination of 9 variables (called "DMS - WaPOR selected" in Section 4.3.2) is used in the ETa processing chain to sharpen the 1 km Sentinel-3 LST to the required 300 m spatial resolution..

## 2.6  Weather forcing

Weather forcing is critical for accurate estimation of ET. Due to the 2-day timeliness requirements for the CLMS ETa product, the weather data source is Copernicus Atmosphere Monitoring Service (CAMS) forecasts (Peuch et al., 2022), produced by the European Center for Medium Range Weather Forecasts and distributed freely and openly through the CAMS Data Store. CAMS data contain surface meteorological parameters covering the whole Earth on a 0.4° grid and hourly temporal resolution.

Instantaneous weather forcing at the satellite overpass are used to drive both ET models and include air temperature, vapor 285 pressure, wind speed, surface pressure, and clear-sky solar irradiance (Table 6). All instantaneous data were obtained by linear interpolation between two CAMS hourly forecasts to the time of Sentinel-3 SLSTR acquisition over the area of interest. Daily weather forcing is used to drive the ETLook ET model and to extrapolate and interpolate the instantaneous estimates of ET and include solar irradiance as well as air and dew temperatures, wind speed, and pressure, which are then used to calculate the FAO-56 reference ET (Allen, 1998) required for the gap filling (see Section 2.9). They are being integrated over a 24-h period 290 starting at midnight local time.

The pre-processing of CAMS weather forcing, including topographic correction, was done using the open source Python software meteo_utils (Nieto et al., 2025b) and as described in Guzinski et al. (2021). The only differences from Guzinski et al. (2021) being the use of Copernicus DEM (COP-DEM_GLO-90-DTED - European Space Agency and Airbus (2022)) resampled to a resolution of 300 m for topographic correction and the use of REST2 model (Gueymard, 2008) to estimate clear



sky solar irradiance. In addition daily total precipitation is calculated using 24-h integration of CAMS hourly total precipitation forecasts.

## 2.7 Structural and ancillary parameters

Resistance energy balance models need additional ancillary inputs, such as canopy height or roughness. The latter influences the efficiency of the turbulent transport of heat and water between the land surface and the overlying air (Raupach, 1994;

Alfieri et al., 2019). Vegetation structure and density are thus important for estimating turbulent transport of momentum, heat and water vapour in the canopy air space (Garratt and Hicks, 1973; Thom, 1972; Raupach, 1994; Shaw and Pereira, 1982).

For that reason, the CLMS Global Dynamic Land Cover map (Copernicus Land Monitoring Service, 2015) is used to assign vegetation parameters, which are difficult to estimate directly from other Earth Observation data (Guzinski et al., 2021). Those parameters, and values assigned to different land cover classes, are listed in Table 7. The TSEB-PT model requires all of the

parameters, apart from stomatal resistance, while ETLook requires only vegetation height and stomatal resistance. Values of all parameters, except for vegetation height of forested land covers, were adapted from Guzinski et al. (2020).

Vegetation height is one the most important of the ancillary parameters, especially for the TSEB model (Burchard-Levine et al., 2020), as it influences the aerodynamic resistance to heat transport. In order to better estimate the obstacle (canopy) height in different land covers we use a framework that differs depending on predominant plant functional type of each land

cover. For for annual plant functional types, canopy height is dynamically computed considering its growth as:

$$h_c = h_{min} + (h_{max} - h_{min}) \times min(\frac{LAI}{LAI_{max}}, 1) \tag{5}$$

where the symbols are described in Table 7.

For forest plant functional types we set a static canopy height based on a 10-m spatial resolution (resampled to 300 m resolution) global forest canopy height map developed by combining the Global Ecosystem Dynamics Investigation (GEDI)

LiDAR and Sentinel-2 observations using a probabilistic deep learning model (Lang et al., 2023). Whenever the land cover map indicated a forest while the GEDI-based canopy height map was below the $h_{min}$ parameter, the minimum value was enforced.

## 2.8 ET modelling

### 2.8.1 TSEB-PT model

The Two-Source Energy Balance (TSEB) modelling scheme was proposed by Norman et al. (1995) and afterwards refined and applied in a multitude of applications and studies (Anderson et al., 2024) including in the Sen-ET framework (Guzinski et al., 2020) and implemented as open source in pyTSEB Python package (Nieto et al., 2025a). In this modelling scheme the directional radiometric LST $(T_R(\theta))$ is split into the temperatures of vegetation and soil based on the vegetation cover and LST





**Table 7.** Land Cover (LC) Look-Up-Table for ancillary and structural parameters required by ET models, adapted from Guzinski et al. (2020). $h_{min}$ (m) is the minimum canopy height; $h_{max}$ (m) is the maximum canopy height occurring when leaf area index (LAI) reaches its optimal maximum $\mathrm{LAI}_{max}$ (for annual plant functional types only); $f_c$ is the at-nadir fraction of the ground occupied by a clumped canopy ($f_c = 1$ for a homogeneous canopy); $w_c/h_c$ is the canopy shape parameter, representing the canopy width to canopy height ratio; $l_w$ (m) is the average leaf size; and $r_{st}$ (s m$^{-1}$) is the minimum stomatal resistance. LC classes come from the CLMS Dynamic Land Cover map. Note that snow/ice and water surfaces are masked in the current implementation.

| LC | $h_{min}$ (m) | $h_{max}$ (m) | $\mathrm{LAI}_{max}$ (-) | $f_c$ (-) | $w_c/h_c$ (-) | $l_w$ (m) | $r_{st}$ (s m$^{-1}$) | Description |
|---|---|---|---|---|---|---|---|---|
| 20 | 2.0 | 2.0 | 0.0 | 1.0 | 1.0 | 0.05 | 175 | Shrubs |
| 30 | 0.1 | 1.0 | 4.0 | 1.0 | 1.0 | 0.02 | 150 | Herbaceous vegetation |
| 40 | 0.0 | 1.0 | 5.0 | 1.0 | 1.0 | 0.02 | 125 | Cultivated and managed vegetation/agriculture (cropland) |
| 50 | 10.0 | 10.0 | 0.0 | 0.0 | 0.0 | 0.00 | 400 | Urban / built up |
| 60 | 0.1 | 0.1 | 0.0 | 0.1 | 1.0 | 0.01 | 100 | Bare / sparse vegetation |
| 90 | 0.0 | 2.0 | 5.0 | 1.0 | 1.0 | 0.10 | 150 | Herbaceous wetland |
| 100 | 0.3 | 0.3 | 0.0 | 1.0 | 1.0 | 0.01 | 180 | Moss and lichen |
| 111 | 8.0 | GEDI | GEDI | 0.8 | 0.5 | 0.05 | 200 | Closed forest, evergreen needle leaf |
| 112 | 8.0 | GEDI | GEDI | 0.8 | 1.0 | 0.15 | 200 | Closed forest, evergreen, broad leaf |
| 113 | 8.0 | GEDI | GEDI | 0.8 | 0.5 | 0.05 | 200 | Closed forest, deciduous needle leaf |
| 114 | 8.0 | GEDI | GEDI | 0.8 | 1.0 | 0.15 | 200 | Closed forest, deciduous broad leaf |
| 115 | 8.0 | GEDI | GEDI | 0.8 | 0.8 | 0.10 | 200 | Closed forest, mixed |
| 116 | 8.0 | GEDI | GEDI | 0.8 | 0.8 | 0.10 | 200 | Closed forest, unknown |
| 121 | 5.0 | GEDI | GEDI | 0.3 | 0.5 | 0.05 | 200 | Open forest, evergreen needle leaf |
| 122 | 5.0 | GEDI | GEDI | 0.3 | 1.0 | 0.15 | 200 | Open forest, evergreen, broad leaf |
| 123 | 5.0 | GEDI | GEDI | 0.3 | 0.5 | 0.05 | 200 | Open forest, deciduous needle leaf |
| 124 | 5.0 | GEDI | GEDI | 0.3 | 1.0 | 0.15 | 200 | Open forest, deciduous broad leaf |
| 125 | 5.0 | GEDI | GEDI | 0.3 | 0.8 | 0.10 | 200 | Open forest, mixed |
| 126 | 5.0 | GEDI | GEDI | 0.3 | 0.8 | 0.10 | 200 | Open forest, unknown |

observation geometry.

$$T_R(\theta) \approx [f(\theta)T_C^4 + [1 - f(\theta)]T_S^4]^{0.25} \tag{6}$$

$$f(\theta) = 1 - exp(\frac{-0.5\Omega(\theta)LAI}{cos\theta}) \tag{7}$$

where $\theta$ is the view zenith angle of the thermal observation and $\Omega(\theta)$ is the clumping factor of the vegetation at view angle $\theta$ (Kustas and Norman, 1999) and has a value of less than 1 for clumped vegetation and subscripts C and S denote canopy and soil respectively.



Based on this split, the energy fluxes of vegetation and soil (net radiation - $R_n$, sensible heat flux - H, latent heat flux - $\lambda E$, soil heat flux - G) are estimated separately, before being combined to obtain the bulk surface fluxes.

$$R_{n,C} = H_C + \lambda E_C \tag{8}$$

$$R_{n,S} = H_S + \lambda E_S + G \tag{9}$$

$$R_n = (H_C + H_S) + (\lambda E_C + \lambda E_S) + G = H + \lambda E + G \tag{10}$$

The soil (canopy) sensible heat flux is computed from the gradient between the soil (canopy) temperature ($T_S$ and $T_C$ respectively) and the air temperature at the sink-source height. This transfer of heat between the two components and the atmosphere is modulated by resistances to heat exchange organized in a series resistance network (in analogy to electrical systems) which depend on aerodynamic and meteorological conditions.

        Since there are multiple solutions to $T_C$ and $T_S$ satisfying equation 6, an iterative approach is employed. The initial as-
sumption is that green canopy transpires at potential rate based on Priestley-Taylor formulation (Priestley and Taylor, 1972):

$$\lambda E_C = \alpha_{PT} f_g \frac{\Delta}{\Delta + \gamma} R_{n,C} \tag{11}$$

where $\alpha_{PT}$ is the Priestley-Taylor coefficient, $\Delta$ is the slope of the vapour pressure to air temperature curve (mbar K$^{-1}$) and $\gamma$ is the psychrometric constant (mbar K$^{-1}$). In all land-covers $\alpha_{PT}$ has an initial value of 1.26, except for forests ($h_c \geq 5m$)
where it is lowered to account for reduction in stem conductivity (and therefore stomatal conductance) with height following Komatsu (2005): $\alpha_{PT} = -0.269 ln(h_c) + 1.31$. If unrealistic fluxes are obtained ($\lambda E_C < 0$ and $\lambda E_S < 0$) then the canopy transpiration (i.e., $\alpha_{PT}$) is sequentially reduced and soil and canopy temperatures and fluxes are recalculated until realistic values are obtained. This implementation of the TSEB scheme is called the TSEB-PT model.

        TSEB-PT outputs instantaneous fluxes at the time of thermal image acquisition. The modelled instantaneous latent heat flux
($\lambda E_{inst}$), calculated during clear-sky conditions, is extrapolated to daily ET values as $\lambda E_{daily} = \lambda E_{inst} \times \frac{S_{daily}^{\downarrow}}{S_{inst}^{\downarrow}}$, with $S_{daily}^{\downarrow}$ and $S_{inst}^{\downarrow}$ are the daily and instantaneous shortwave irradiances, respectively.

### 2.8.2 ETLook model

ETLook model (Bastiaanssen et al., 2012) is used in the WaPOR framework and is described in detail in Section 5 of "WaPOR Data Manual, Evapotranspiration v2.2" (FRAME Consortium, 2020). Similarly to TSEB-PT, ETLook is a two-source model,
meaning that it derives soil evaporation (E) and vegetation transpiration (T) as two separate fluxes, and it ensures conservation of energy at the land-surface. The model assumes potential rates of daily E and T based on the Penman–Monteith equation (Monteith, 1965) that are throttled down to actual E and T using stress factors:





$$E = \frac{\Delta(R_{n,S} - G) + \rho c_p \frac{\Delta_e}{r_{a,S}}}{\Delta + \gamma(1 + \frac{r_S}{r_{a,S}})} \qquad (12a)$$

$$T = \frac{\Delta(R_{n,C}) + \rho c_p \frac{\Delta_e}{r_{a,C}}}{\Delta + \gamma(1 + \frac{r_C}{r_{a,C}})} \qquad (12b)$$

where $\Delta_e$ (mbar) is vapor pressure deficit, $\rho$ (kg m$^{-3}$) is the air density, $c_p$ (J kg$^{-1}$ K$^{-1}$) is specific heat of dry air, $r_{a,S}$ and $r_{a,C}$ are aerodynamic resistances for soil and canopy respectively and $r_S$ and $r_C$ are resistances of soil and canopy. All resistances are in s m$^{-1}$.

The resistances of soil and canopy to energy transfer are calculated taking the stress factors into account:

$$r_S = b(S_e^{top})^c \qquad (13a)$$

$$r_C = (\frac{r_{s,min}}{LAI_{eff}})(\frac{1}{S_t S_v S_r S_m}) \qquad (13b)$$

where $b$ and $c$ are soil resistance parameters, $r_{s,min}$ (s m$^{-1}$) is the minimum stomatal resistance, $LAI_{eff}$ is effective leaf area index. Soil evaporation is limited by top-soil moisture ($S_e^{top}$), while plant transpiration is affected by air temperature stress ($S_t$), vapour pressure stress ($S_v$), radiation stress ($S_r$) and root-zone soil moisture stress ($S_m$)(Jarvis, 1976).

The soil moisture required for E and T stress factors is derived using a trapezoid constructed in the LST - vegetation
fractional cover ($f_C$) space (Yang et al., 2015). The trapezoid corner values as set based on theoretical calculations by inverting the Penman-Monteith equation for both dry and moist bare soil and vegetated conditions. Then the soil moisture of a pixel is estimated using the relative location of LST and $f_C$ of that pixel within that trapezoid.

## 2.9 Output gap-filling

The CLMS ETa product consists of 5 sub-products: instantaneous sensible and latent heat fluxes (in Wm$^{-2}$), and dekadal
(10-day mean) evapotranspiration and its components evaporation and transpiration (in mm day$^{-1}$). The instantaneous heat fluxes represent values modeled at the time of Sentinel-3 satellite overpass and, therefore, do not undergo any gap-filling. On the other hand, the water fluxes need to undergo gap-filling. Otherwise, the dekadal aggregation would only take into account fluxes modeled during clear-sky conditions within the aggregation period. This would firstly result in frequent gaps and secondly in a systematic overestimation of the dekadal aggregate.

The gap-filling of ET product, i.e. estimation of ET during cloudy conditions, is usually performed using a reference quantity that can be derived for any date regardless of cloud conditions. This implies that this reference quantity is mostly dependent on weather forcing. In the Sen-ET approach the choice was made to use reference evapotranspiration calculated using the FAO-56 method (Allen, 1998). A ratio of modeled daily ETa to reference ET (called crop-stress coefficient $K_{s,c}$) is calculated on dates for which daily ETa is available and is used to recreate ETa on the target date which needs to be gap-filled, as described
in Guzinski et al. (2021). This method was further developed in Guzinski et al. (2023) to better account for soil drying by performing linear interpolation of $K_{s,c}$ (applicable only in non-NRT modelling when ETa after the target date is available)




and for soil wetting through rainfall by setting $K_{s,c}$ to the maximum value observed during the gap-filling period if a simple water-balance approach indicates that the soil is wet. Taking rainfall into account is important especially for longer gap-filling periods (Delogu et al., 2021) and in climates in which rainfall initiates the growing season (e.g. rainy season in the Sahel and other semi-arid areas). For the CLMS ETa processing chain we further improved the robustness of this method, especially for longer gap-filling windows, by replacing the maximum $K_{s,c}$ observed within the gap-filling window with the 80th percentile of ratios observed within the gap-filling period. If the latest know $K_{s,c}$ before the target date was larger than 80th percentile then the value of that last know $K_{s,c}$ was preserved. This was done to ensure that taking rainfall into account would only increase, and not reduce, the gap-filled ET values. To accommodate longer periods with few satellite observations (see Fig 2) a 60-day gap-filling window is used during ETa production.

Once the gap-filled ET is estimated, the split into evaporation and transpiration is performed using the ratio of evaporation or transpiration to ET either from the closest non-gap-filled preceding date (in case of NRT processing) or linearly interpolated ratio from the closest non-gap-filled preceding and succeeding dates (in case of non-NRT processing).

## 3 Prototype product validation

The main aim of the validation is to evaluate the performance of the prototype product through comparison of dekadal ETa values, as modelled by the TSEB-PT and ETLook algorithms, with ETa data derived from one year of in-situ measurements from stations covering main worldwide climatic zones and plant functional types. In addition to assessing the performance of the two ETa models driven by input forcing as described in Section 2, the statistical metrics were also computed for an ETa dataset composed by the average of the TSEB-PT and ETLook estimates. This additional dataset is referred to as Ensemble ETa. The statistical metrics used in the analysis were: bias, root mean squared error (RMSE) and coefficient of determination ($r^2$). This comparison focused only on the variables that could be extracted from measurements at eddy covariance stations (ETa, $\lambda E$, H). Evaporation and transpiration were generated by the models but could not be contrasted against *in situ* observations.

### 3.1 Validation data collection and preparation

The search for eddy covariance sites was intended to collect data representing the largest possible diversity in climate regions and plant functional types. Different eddy covariance networks and datasets were explored in search for validation data for recent years. The year 2020 was selected for this analysis considering the overall *in situ* data availability.

When building the reference database, the priority was given to datasets in which $\lambda E$ and H data had been corrected for the energy balance closure (EBC) problem (Foken et al., 2011). A commonly applied procedure delivering corrected values of $\lambda E$ and H is the ONEflux processing pipeline (Pastorello et al., 2020). Data generated by this procedure could be obtained from the AmeriFlux (Novick et al., 2018), ICOS (Integrated Carbon Observations System - Heiskanen et al. (2022)) and OzFlux (Isaac, 2014) networks. In search for more diversity in the reference datasets, other networks/datasets were queried even if the processing chain was not accounting for the EBC. Thus, data from the European Fluxes Database Cluster (EFDC - https://www.europe-fluxdata.eu/, last accessed: 01/09/2025), l'observatoire AMMA-CATCH (Analyse Multidisciplinaire de





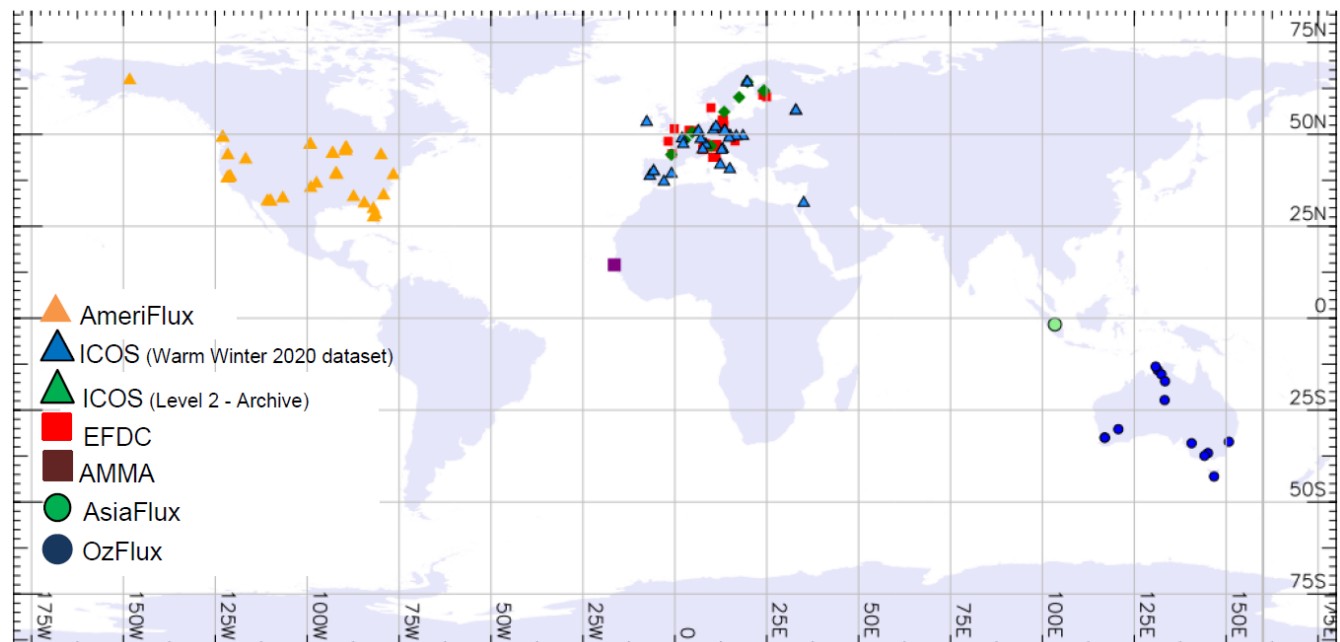

**Figure 4.** Eddy covariance sites and networks used for quality assessment.

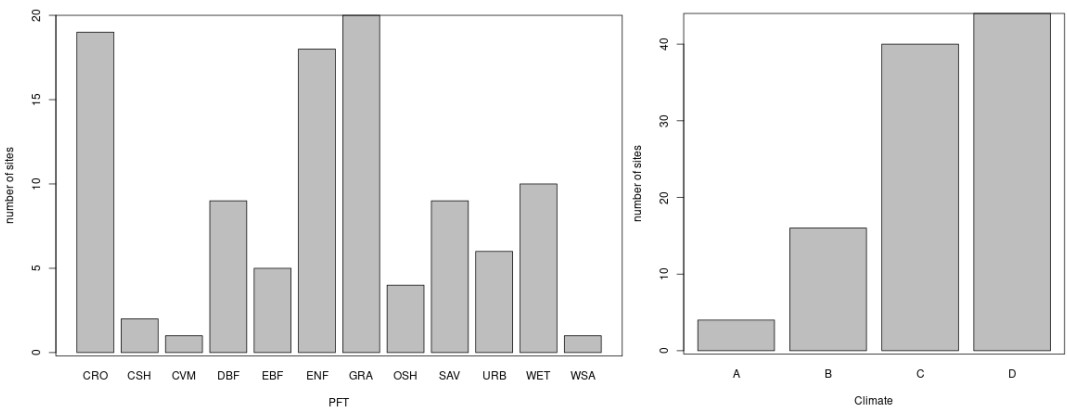

**Figure 5.** Number of eddy covariance sites grouped per PFT (left) and per climate region (right). (CRO: Cropland; CSH: Closed Shrubland; CVM: Crop Vegetation Mosaic; DBF: Deciduous Broadleaved Forest; EBF: Evergreen Broadleaved Forest; ENF: Evergreen Needleleaved Forest; GRA: Grassland; OSH: Open Shrubland; SAV: Savanna; URB: Urban; WET: Wetland; WSA: Woody Savanna) (A: Tropical; B: Dry; C: Continental; D: Temperate).

la Mousson Africaine - Couplage de l'Atmosphère Tropicale et du Cycle Hydrologique -Galle et al. (2018); AMMA-CATCH
(1990)) and the AsiaFlux (Mizoguchi et al., 2009) networks were also included in the analysis.



Figure 4 shows the location of the eddy covariance sites considered in this study and the network/dataset the data were taken from, while Figure 5 provides an overview of the representation of climate regions and plant functional types (PFT) in this set of eddy covariance sites. Further details of the location, climate, PFT and network of each site can be found in Appendix B.

Figures 4 and 5 show that the available eddy covariance networks are not equally distributed across the globe which results
in imbalanced representativeness of climate regions and plant functional types. However, at least one station was located in all major climate zones and PFTs. Despite only 3 sites being outside of Europe, North America and Australia, there are other sites which are located in similar climatic zones, e.g. tropics (northern Australia) as well as arid and semi-arid (Spain, southern Australia and western US). Therefore, we believe that the validation should be representative also of the geographical areas for which in-situ ET data is currently missing.

The data acquired from the AmeriFlux, ICOS and OzFlux networks was available at half-hourly and daily time steps and format aspects like variable naming, units, quality flags, etc. were uniform. Daily $\lambda E$ data were discarded if the value of the quality flag was lower than 0.6 (i.e. less than 60% of sub-daily data was not measured or had a good quality gap filling) and/or the value of $\lambda E$ was outside the realistic range. A dekadal ETa value was computed if the dekad was composed of at least 7 valid daily ETa values. The computation of ETa from $\lambda E$ and air temperature was conducted as follows:

$$lv = (2.501 - 0.00237 T_a) \times 10^6 \tag{14}$$

$$ET = 3600 \times 24 \times \frac{\lambda E}{lv} \tag{15}$$

where $lv$ is the latent heat of vaporization, $T_a$ is the average daily air temperature, $\lambda E$ is the daily average latent heat flux and $ET$ is the daily evapotranspiration in mm/day.

The data from other networks was delivered at half-hourly time step only. Therefore, an additional aggregation step needed
to be considered to obtain daily ETa values. In doing this, the half-hourly data were filtered on the basis of quality flag and occurrence within the realistic range. For each day, the number of valid timeslots between sunrise and sunset was computed (the sunrise and sunset times change as function of geographic location and time of the year). Missing data during the day were computed by linear interpolation if the number of valid timeslots during daytime was at least 50% of the total number of timeslots in that period. Otherwise, the day was discarded. The criterion for aggregating the daily ETa values to dekadal values
was the same as indicated in the previous paragraph.

The last preparatory step was matching the modelled and reference values on the basis of timeslot and location. This is a straightforward step for the dekadal ETa values. The data at satellite overpass time ($\lambda E$ and H) were matched to the nearest timeslot of the eddy covariance half-hourly datasets. The analysis for $\lambda E$ and H at satellite overpass time was conducted for TSEB-PT only as the ETLook algorithm does not generate those variables.





**Table 8.** Bias (mm/day), RMSE (mm/day), and $r^2$ scores for dekadal ETa per model (all sites combined), and summary statistics at site level. rBias and rRMSE are relative metrics (i.e., divided by mean measured ET).

| Model | N | Bias | RMSE | rBias | rRMSE | $r^2$ | Summary scores at site level | | | | | |
| | | | | | | | Bias | | RMSE | | $r^2$ | |
| | | | | | | | min | max | min | max | mean | St.Dev |
|---|---|---|---|---|---|---|---|---|---|---|---|---|
| TSEB-PT | 3108 | -0.12 | 0.88 | -0.07 | 0.52 | 0.78 | -1.19 | 1.41 | 0.28 | 1.77 | 0.76 | 0.25 |
| ETLook | 3108 | -0.44 | 1.08 | -0.26 | 0.64 | 0.78 | -0.72 | 2.56 | 0.27 | 2.82 | 0.79 | 0.24 |
| Ensemble | 3108 | -0.28 | 0.80 | -0.16 | 0.47 | 0.84 | -0.66 | 1.98 | 0.28 | 2.13 | 0.82 | 0.21 |
| WaPOR | 3108 | 0.12 | 0.85 | 0.07 | 0.50 | 0.81 | -1.01 | 1.53 | 0.20 | 1.85 | 0.77 | 0.28 |

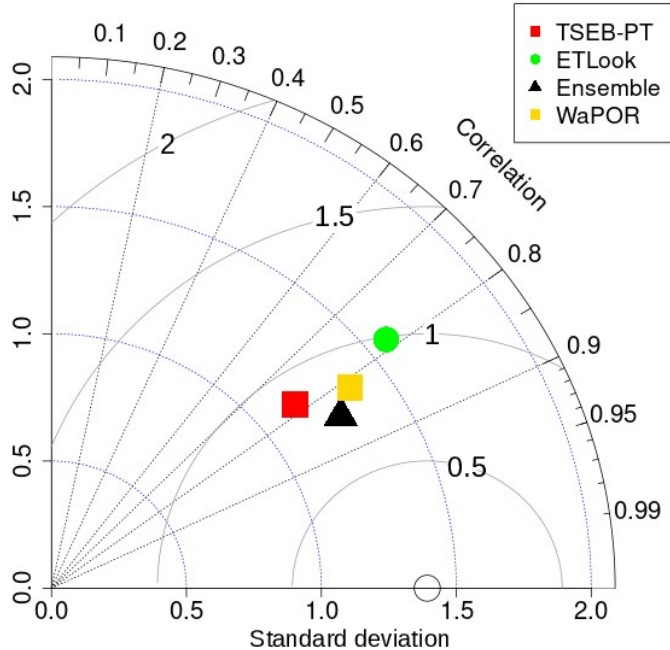

**Figure 6.** Taylor-plot general overview. The circle on the x-axis is the standard deviation of the eddy covariance towers measurements.

## 3.2 Validation results

### 3.2.1 Dekadal ETa values

The procedure described in the previous section resulted in a dataset of 3108 records (104 sites) of dekadal ETa values obtained from eddy covariance measurements. A first appraisal on the performance of the ETa models can be obtained from the statistical scores shown in Table 8. A graphical representation of the performance of the models under consideration is presented in the Taylor plot of Figure 6.





The points to be highlighted in these statistical scores are:

– The modelled ETa series under consideration exhibit high correlation with the reference dataset ($r^2$ around or above 0.8).

– The RMSE of the Ensemble dataset is the lowest in the evaluation and the TSEB-PT yielded the lowest absolute bias.

– The bias values of ETLook, TSEB-PT and the Ensemble ETa are negative.

Around 78% of the data records used in calculating the scores of Table 8 and Figure 6 had been corrected for the EBC problem. The same analysis was conducted for the subset of data records composed only of EBC-corrected data. The results were very similar to those presented above and are shown in Appendix C.

The plots of Figure 7 show a more detailed view of these results. The bias and $r^2$ are represented as axes in plots showing the values per site with indication of the climate region and PFT each site belongs to. The visual inspection of these plots

shows that in correspondence to the results presented above, the majority of study sites are located in the region of the highest correlation and bias values not exceeding the absolute value of 1 mm/day. The concentration of results within the region of high correlation and low bias seems to be more pronounced in the Ensemble series. Moreover, the majority of sites in the Continental and Temperate climate regions appear in the section of the plots with high correlation and low bias in all ETa series; although various sites in temperate regions appear in the low correlation region of the plots too. Conversely, the models

performed less good in the Tropical and Dry regions.

A complementary view on the model performance when grouping the sites per climate region is presented in the Taylor plots of Figure 8. Those plots confirm that the highest performance of all models was exhibited in the sites of the Temperate and Continental regions. It is important to note that the abundance of sites in these two climate regions is much higher than that of the Tropical and Dry regions (see Figure 4). The plots of Figure 8 point at the Ensemble and TSEB-PT dekadal ETa as the

most suited options to best represent the different climate regions.

Figure 7 also shows that the accuracy of dekadal ETa modelling can often be associated to the PFT. Figure 9 shows Taylor plots with the performance of the models when grouped per PFT. Although the analysis per PFT can offer valuable insights on the performance of the models, one has to be cautious as one PFT class can contain a large diversity of ecosystem conditions.

The CRO, GRA and WET plots of Figure 9 reveal that both models exhibited similar performance in the generation of

dekadal ETa values in those groups. The three classes together account for 42% of the evaluated sites spread across different climate regimes and represent a wide range of conditions. The Ensemble ETa series exhibits slightly better correlation scores.

In savannas (SAV, WSA) and shrublands (CSH, OSH), more pronounced differences between the TSEB-PT and ETLook emerge. SAV sites (which are well represented in the dataset) show that TSEB-PT tends to underestimate the variability of the fluxes, whereas it is generally overestimated by ETLook. These biases compensate each other in the Ensemble model, which

furthermore achieves a higher correlation than TSEB-PT and ETLook separately.

This behaviour is even more pronounced in the forest sites (DBF, EBF, ENF): TSEB-PT underestimates the variability of the flux (i.e. the amplitude of the seasonal cycle), whereas it is overestimated by ETLook. The difference between both models can be largely attributed to the difference in transpiration, of which the value of ETLook can be almost double the value of TSEB-PT, as illustrated in the plotted time series (Figure 10 and Figure 11).



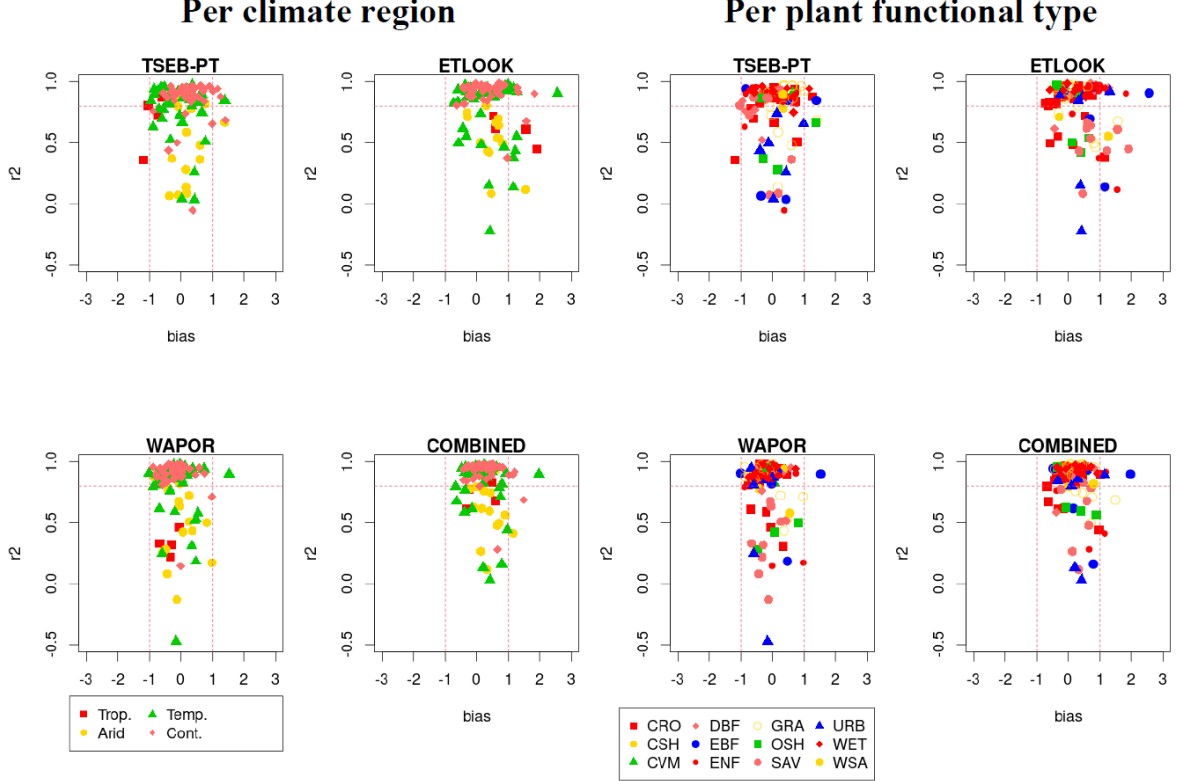

**Figure 7.** Bias and $r^2$ at validation sites indicating the climate region and PFT for ETLook, TSEB-PT, WaPOR and the Ensemble dekadal ETa series.

Figure 10 shows the time series of two EBF sites. These plots illustrate the significant difference amongst models' results and show the high ETa values of ETLook, which cause the large error depicted in the Taylor plots for this group. Note that the IT-Cp2 site (one of the examples in Figure 10) is the site with the largest error in the EBF class.

     The sites in the DBF group are all located in the Temperate or Continental climate region. The DBF group, with 10 sites, exhibited larger differences in the scores for TSEB-PT and ETLook. In the DBF class, the estimates of both models seemed
to complement each other as ETLook estimates correlated better with the reference dataset than TSEB-PT but the error of the latter was smaller. In this case, the Ensemble formula appears as a suitable option for generating estimates closer to the reference ETa values. The Figure 11 shows examples of dekadal values in DBF (US-xST, DE-HoH) sites.

     It should be noted that ETa of urban areas is produced even though neither of the two ET models was designed for this particular land cover type. This is for consistency with other CLMS products. The URB plot in Figure 9 indicates that, as
expected, the TSEB-PT and ETLook output correlate poorly with the reference data in urban areas and the difference between the two models is substantial.





**Figure 8.** Taylor plots on the performance of the dekadal ET from TSEB-PT, ETLook, the Ensemble dataset and WaPOR in 2020. Sites grouped per Climate region. A: Tropical, B: Arid, C: Temperate, D: Continental.





**Figure 9.** Taylor plots on the performance of the dekadal ET from TSEB-PT, ETLook, the Ensemble dataset and WaPOR in 2020. Sites grouped per PFT. CRO, CSH, CVM, DBF, EBF, ENF, GRA, OSH, SAV, URB, WET, WSA.



**Figure 10.** Dekadal ET, E and T as modelled by TSEB-PT and ETLook and the WaPOR product and dekadal ET from the eddy covariance towers in two EBF sites: AU-Whr (Whroo) and IT-Cp2 (Castelporziano2) sites.





**Figure 11.** Dekadal ET, E and T as modelled by TSEB-PT and ETLook and the WaPOR product and dekadal ET from the eddy covariance towers in two DBF sites: US-xST (NEON-STEI) and DE-HoH ( Hohes Holz).




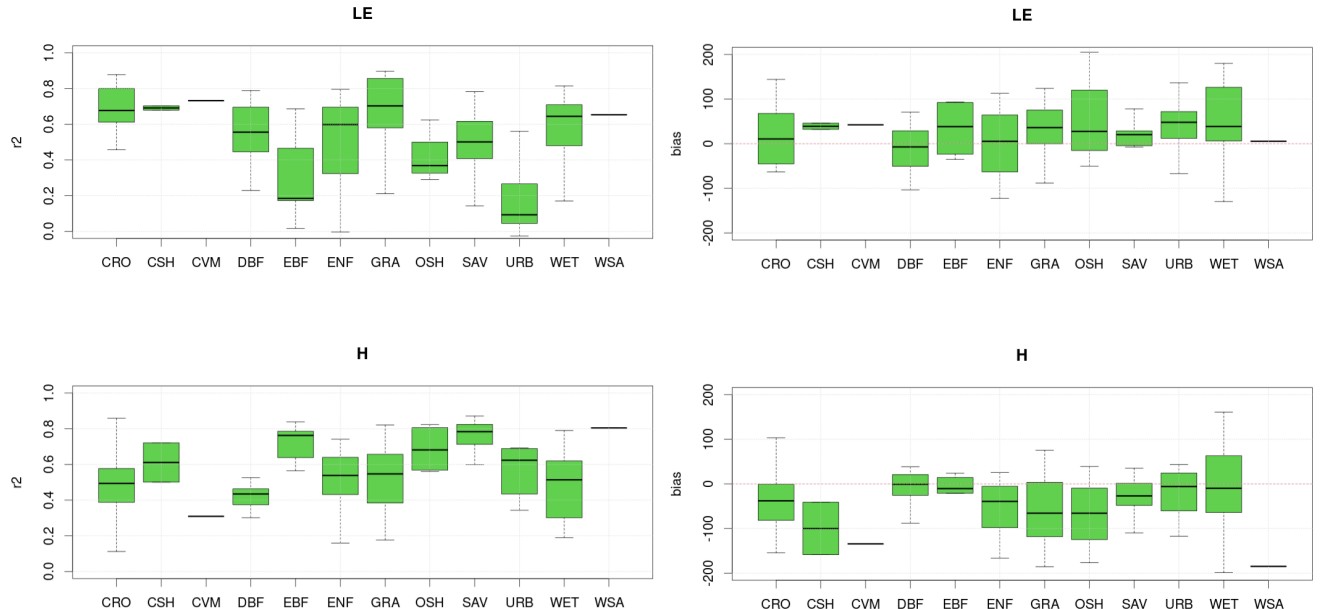

**Figure 12.** Boxplots of $r^2$ (left) and bias (right) values of LE and H at overpass time as modelled by the TSEB-PT model.

### 3.2.2 $\lambda E$ and H at satellite overpass time

Estimates of $\lambda E$ and H at satellite overpass time were generated by the TSEB-PT model only. This section provides an overview of the goodness of fit of the modelled instantaneous $\lambda E$ and H estimates as compared with the values at the closest half-hourly

timeslot reported at the eddy covariance towers. The satellite overpass time, in local time, varied along the year for the different sites. For instance, the AU-Cum site registered values between 9:05 and 10:52 local time; US-Ton, between 10:03 and 11:52; ES-LM1, between 11:27 and 13:13; etc.

The Figure 12 shows the range of $r^2$ and bias values obtained for each PFT class. A number of aspects in this plot can be connected to the analysis of dekadal ETa presented in the previous section. For instance, the low $r^2$ of $\lambda E$ in the urban sites

and, to a certain degree, in the evergreen forest classes. It is also notable that the bias in $\lambda E$ is positive in the majority of the sites and corresponds to dominant negative bias in the H estimates.

The length of the boxes in the boxplots of Figure 12 suggests an important degree of variability in the correlation and error of the modelled values amongst the sites of each PFT class. The differences in the time window at which satellite overpass takes place and the large heterogeneity in ecosystem properties within each class can partly explain this variability.

The Figure 13 shows examples of the modelled $\lambda E$ and H at satellite overpass time in contrast to the corresponding eddy covariance values. The scatterplots illustrate the more pronounced negative bias in H in some sites as well as the poor estimation of the fluxes in urban areas.





**Figure 13.** Scatterplots of LE and H ($Wm^{-2}$) as measured at the eddy covariance stations and as modelled by TSEB-PT at satellite overpass time in BE-Lon (Lonzee) (CRO), ES-LM1 (Las Majadas) (SAV), US-xSB (Ordway-Swisher Biological Stations) (ENF), FR-Fon (Fontainebleau) (DBF), AU-Rgf (Ridgefield) (CRO) and AT-Inn (Innsbruck) (URB).





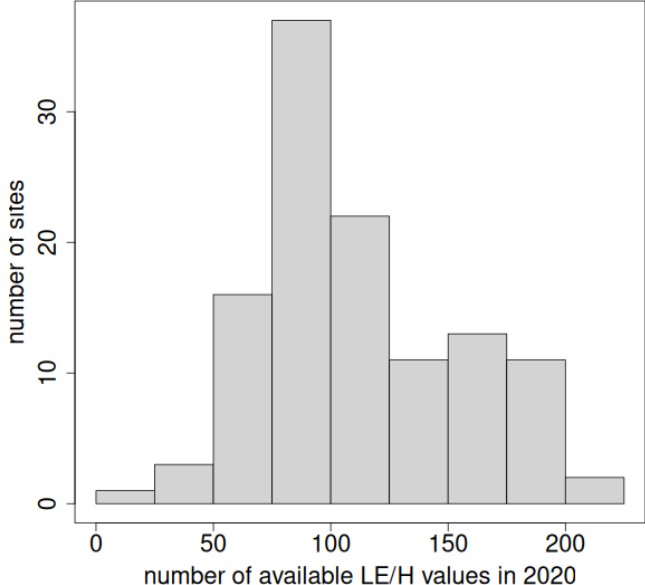

**Figure 14.** Number of study sites by the number of available modelled LE/H values at satellite overpass time in the year 2020.

A noteworthy feature of the $\lambda E$/H modelled dataset is the number of actual values available. Figure 14 provides an overview of the number of available values (regardless of the quality) generated by the TSEB-PT at satellite overpass time. The Figure shows that most of the sites have less than 150 values in a year and only very few exceed 200. This is an important consideration towards the generation of operational $\lambda E$ and H products as the number of gaps can be high and highlights the importance of robust gap-filling scheme for the production of dekadal ETa dataset.

## 4 Discussion

### 4.1 Comparison with WaPOR and OpenET ETa products

WaPOR global ETa dataset is the only existing operational and global ETa product with specifications very similar to the CLMS ETa. Furthermore, given the particular interest of FAO in the quality of the upcoming CLMS ETa product, WaPOR dekadal ETa values (WaPOR Level 1 version 3 - https://data.apps.fao.org/catalog//iso/7f4a7339-d56e-4393-8712-a8ffeffe2731, last accessed 04/08/2025) were extracted for the study sites and timeslots and included in the analysis presented in Table 8 and Figures 6 - 11. Based on those figures, the accuracy of the CLMS ETa is on the same level as that of WaPOR ETa. While the individual model runs of TSEB-PT or ETLook show poorer statistics than WaPOR, the Ensemble dataset has the best accuracy (apart from bias). It is also worth noting that the bias values of ETLook, TSEB-PT and the Ensemble ETa are negative whereas




WaPOR exhibited a slightly positive bias. Looking at Figure 8 it can be seen that WaPOR has particularly poor performance at the tropical sites, while it outperforms the two individual models (but not the Ensemble ETa) in Continental climate.

Another point worth noting is that WaPOR ETa is based on ETLook model and in all statistical measures it outperforms the CLMS ETLook ETa. Since the ETLook model setup should be fairly similar (the operational production setup of WaPOR is not public although FAO has created equivalent open-source package which produces similar results when run with the same input data), those differences can be attributed mainly to the input data. Here, three possible reasons appear most obvious:

- While both CLMS ETa and WaPOR ETa rely on DMS to improve the spatial resolution of LST, the original LST in CLMS is acquired by SLSTR sensor on board Sentinel-3 satellite with 1 km spatial resolution, while the original LST in WaPOR (version 3) is acquired by the VIIRS sensor on board of Suomi-NPP satellite with 375 m spatial resolution.

- The CLMS ETa dataset was produced in NRT mode, meaning that forecast meteorological forcing were used and gap-filling was performed using only preceding dates. On the other hand, WaPOR ETa is a reanalysis product which means that reanalysis meteorological forcing was used and gap-filling was performed using both preceding and succeeding dates.

- WaPOR ETLook implementation relies of some temporarily-static - spatially-distributed layers which parameterize the model. Those layers were used in CLMS ETLook implementation whenever available but there are still some layers which were either provided too late during the study (e.g. dry bare soil surface albedo) or which are not publicly disclosed (e.g. tenacity factor for plant soil moisture stress) in which case default constant values were used.

OpenET (Melton et al., 2022) is another dataset with which CLMS ETa can be compared. Although OpenET is produced with much higher spatial (30 m) and temporal (daily) resolutions and only in the western United States, similarly to CLMS dataset it contains a product which is an average (ensemble) of individual ET models (6 in case of OpenET). In a recent validation study, the RMSE of monthly ensemble product was between 12% and 30% lower than that of individual models while $r^2$ increased from 0.83 - 0.87 for individual models to 0.9 for ensemble (Volk et al., 2024). In our case the RMSE of dekadal Ensemble product was 9% lower than of TSEB-PT ET and 26% lower than of ETLook ET, while $r^2$ increased from 0.78 for individual models to 0.84 for Ensemble product. In another study, the daily mean Ensemble OpenET product had a RMSE of 0.96 mm/day, bias of -0.2 mm/day and $r^2$ of 0.84 (Melton et al., 2022) which is very similar to the results presented in Table 8 for the Ensemble dekadal ETa despite the increased uncertainty when validating 300 m product due to spatial-scale mismatch between flux tower footprint and pixels size.

## 4.2 Spatial intercomparison

The validation assessment presented in Section 3 was based on the comparison of TSEB-PT and ETLook ETa estimates with measurements at eddy covariance towers; i.e. a point location analysis. However, the view of the spatial patterns in the ETa calculation by the ETLook and TSEB-PT models can give interesting insights towards the design of an operational ETa product.



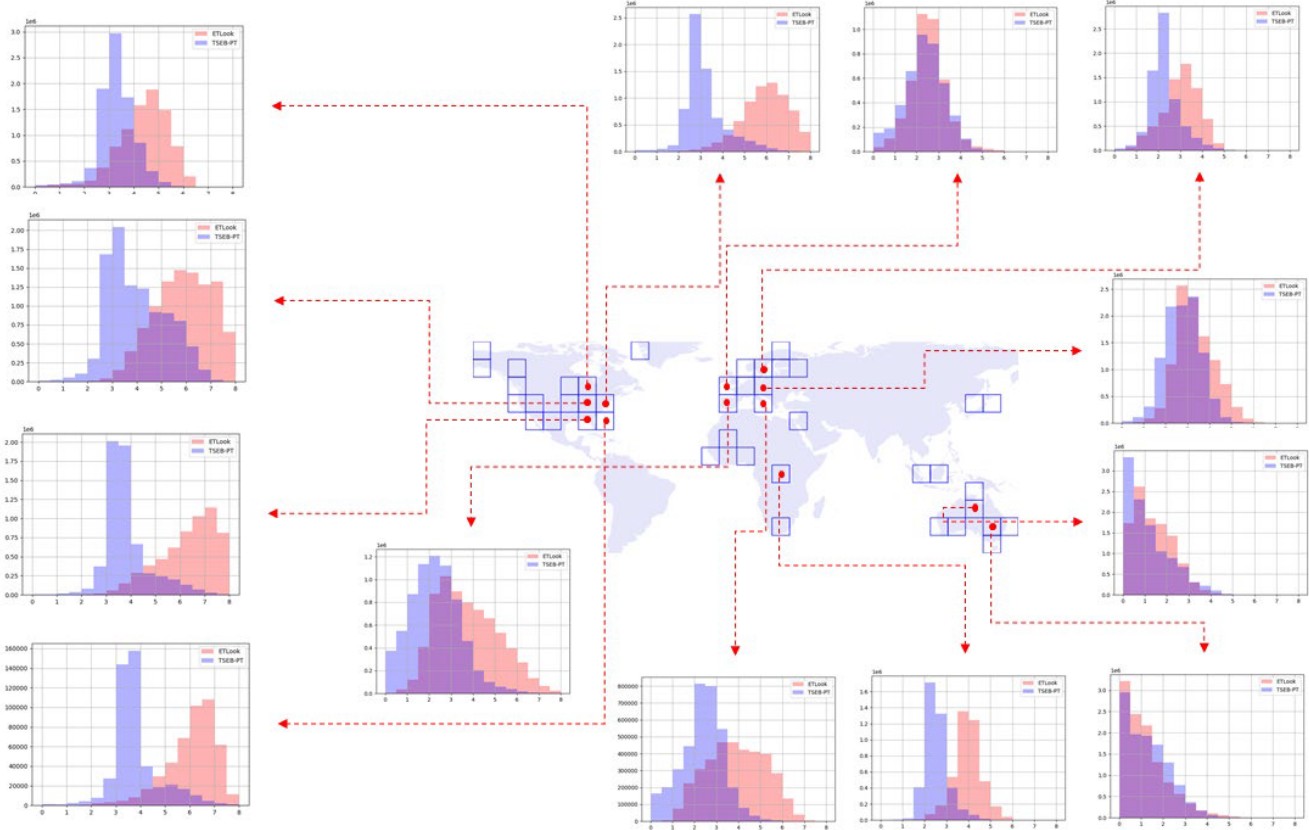

**Figure 15.** Histograms of TSEB-PT (blue) and ETLook (orange) dekadal ET in a dekad in the vegetation growing season across different 10° lat/lon tiles. 2020.

The modelled data under analysis here were generated on a per-tile basis whereby the globe is divided in areas with 10° lat/lon extent, as in other CLMS products. Without entering into detailed analyses of ETa spatial patterns (beyond the scope of 565 this study) simple inspection procedures across the tiles can deliver interesting information.

For instance, Figure 15 shows histograms on the dekadal ETa values as modelled by ETLook and TSEB-PT for a number of tiles. The histograms reflect the spread of ETa values in the middle of the vegetation growing season and are indicators of the difference in the magnitude of ETa estimated by each model. The histograms of Figure 15 are only a small subset of the different tiles on the globe and do not reflect the seasonal variation of ETa. Nevertheless, they already show very different 570 situations ranging from similar frequency distribution patterns in Australia and the northernmost tiles, through very clearly separated patterns in Congo and the southern United States, to the wider range of ETLook ETa values in the Iberian Peninsula and Italy as compared to the narrower range in the estimates by TSEB-PT.



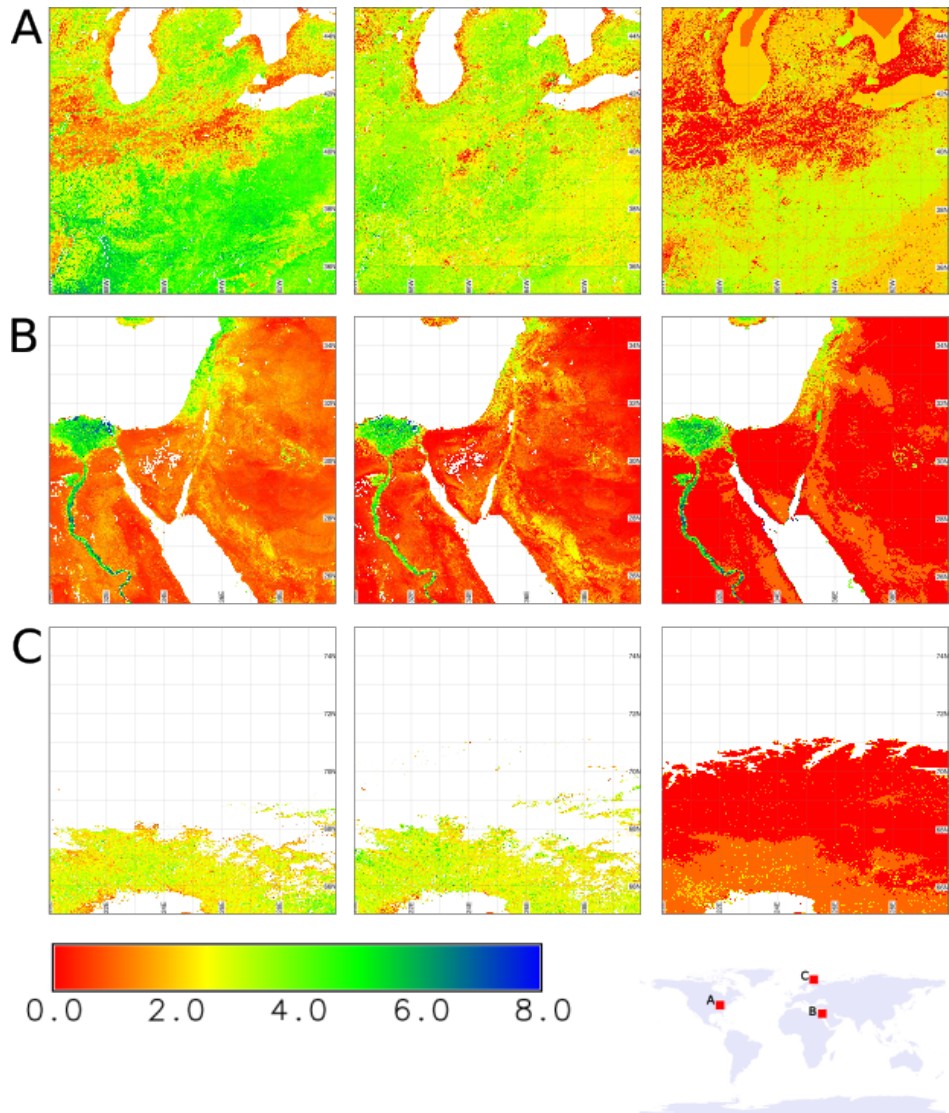

**Figure 16.** Dekadal ETa (mm/day) across three 10° lat/lon tiles as modelled by ETLook (left column), TSEB-PT (center column) and the WaPOR v3 (right column) product in the last dekad of May for panels A and C and the second dekad of July for panel B. 2020.

Although the sample is not large enough to extract solid conclusions, the difference in distribution patterns between models seems to be larger in temperate and tropical regions (like in Africa and the southern United States) as compared to latitudes 575 further away from the equator.

The visual inspection of the ETa values at tile level gives a complementary view to the histograms presented above. Figure 16 presents a small set of examples that allow the visual comparison of the dekadal ETa estimates by ETLook and TSEB-PT





and the WaPOR product at tile level. These images confirm what was mentioned earlier concerning differences/similarities in the output of ETLook and TSEB-PT. The panel A of Figure 16 illustrates the wider spread of ETa values pointed out earlier

which translates into regions of higher ETa in the ETLook estimations as compared to those of TSEB-PT.

The examples in Figure 16 show as well that the number of missing data in the output maps of ETLook and TSEB-PT is larger than in the WaPOR product. The reason for those gaps are the differences between NRT and reanalysis gap-filling (see Section 4.1) but also the different model inputs and treatments of inland water and snow. For example, the CLMS TOC reflectance product (principal input to ETa product) uses IDEPIX method for cloud-masking that it is known to frequently and

persistently mis-classify bright areas as potential clouds (Copernicus Land Monitoring Service, 2023). This explains the gaps seen in Figure 16-B in the desert areas. Regarding snow and inland water, neither TSEB-PT nor ETLook are designed to model evaporation of those surfaces and therefore both are masked out in the CLMS product (e.g. 16-A for water and 16-C for snow). WaPOR, on the other hand, estimates evaporation of those surfaces as a post-processing step.

### 4.3 Evaluation of CLMS ETa modelling framework components

Section 3 focused on the validation of the final CLMS ETa model outputs. During the development of the ETa modelling framework various design choices were made and intermediate products evaluated to justify those choices. In this section, we briefly present this evaluation.

#### 4.3.1 Biophysical traits, albedo and net shortwave radiation

The biophysical traits were obtained by random-forest inversion of PROSPECT-D+4SAIL RTM (Section 2.3). In order to

test the sensitivity and robustness of such inversion method, we ran an independent set of PROSPECT-D+4SAIL simulations to compare how the regression model predicts the biophysical traits. Random white noise was added to this test dataset, considering that retrieved TOC have relative uncertainty of 10% ($\rho_{test} = \rho_{ProSAIL}\left[1+\mathcal{N}\left(0,0.1\right)\right]$). This is shown as an example in Table 9.

A more computationally efficient alternative for the retrieval of LAI and $f_g$ (but not pigments used for albedo estimation)

would be to use the daily estimates of LAI and fraction of absorbed photosynthetically active radiation (fAPAR), which are internal CLMS datasets used to produce the 300 m dekadal biophysical CLMS product. In this case we could use a simple relationship between $f_g$, LAI and fAPAR (Fisher et al., 2008) in order to derive the fraction of LAI that is green ($f_g$).

$$f_g = \frac{fAPAR}{fIPAR} \tag{16}$$

$$fIPAR = 1 - e^{K(\theta_s)LAI} \tag{17}$$

where LAI is the total Leaf Area Index ($LAI = \frac{gLAI}{f_g}$), fIPAR is fraction of intercepted photosynthetically active radiation and $K(\theta_s)$ is the shortwave beam coefficient of extinction at solar zenith angle $\theta_s$.

Since CLMS LAI product actually represents the green LAI (Copernicus Land Monitoring Service, 2022), and with $f_g$ and total LAI unknown, an iterative procedure proposed by Guzinski et al. (2020) is performed to find the optimal value of $f_g$ based





**Table 9.** Evaluation performance for the Random Forest hybrid inversion of the PROSPECT-D+4SAIL radiative transfer model. The "observed" dataset corresponds to 40000 independent simulation of PROSPECT-D+4SAIL for a VZA=0°, SZA=37.5°, a standard atmosphere, and a relative uncertainty of 10% in the TOC reflectance retrievals. Cab, Car, Ant, Cbrown, are respectively the leaf concentrations of chlorophyll a+b, carotenoids, anthocyanins and brown pigments; Cm and Cw are respectively the leaf dry matter and water contents; LAI is the leaf area index and Leaf Angle is the Campbell (1990) mean leaf inclination angle. Physical units in the error metrics are consistent with those on Table 4

| Trait | N | bias | RMSE | r |
|---|---|---|---|---|
| Cab | | 0.011 | 8.480 | 0.96 |
| Car | | -0.013 | 4.477 | 0.90 |
| Cm | | -0.000 | 0.004 | 0.92 |
| Cw | 40000 | -0.000 | 0.008 | 0.92 |
| Ant | | -0.068 | 4.617 | 0.93 |
| Cbrown | | -0.002 | 0.232 | 0.93 |
| LAI | | 0.013 | 0.592 | 0.97 |
| Leaf Angle | | 0.020 | 6.436 | 0.93 |

on gLAI and fAPAR. An initial LAI is assumed equal to gLAI (i.e. $f_g = 1$), from which fIPAR is computed from Eq. 17 and

then $f_g$ recalculated with Eq. 16. This process is repeated until the $f_g$ value converges between iterations.

In order to evaluate whether both approaches provide consistent data, we ran the biophysical processor over sites included in the ICOS WarmWinter2020 database (Warm Winter 2020 Team et al., 2022). These selected sites are listed in Table B3. We thus compared both LAI and $f_g$ retrieved between 2019 and 2021 using the method described in Section 2.3 against the Fisher et al. (2008) and Guzinski et al. (2020) LAI/fAPAR approach using the CLMS FAPAR (Copernicus Land Monitoring Service,

2017a) and LAI (Copernicus Land Monitoring Service, 2017b) global products at 300m, version 1. The density plots of Figure 17 shows that overall the LAI products agree well, in particular at values lower than 2–3, with best agreement for green LAI (gLAI). Nonetheless, the larger scatter at higher LAI (i.e. denser vegetation) is not of great concern, as over these very dense canopies the interception (transmission) of radiation is already close to the maximum (minimum), i.e. near the light saturation, and thus these uncertainties have a minimal effect on ETa modelling. It is worth noting these results do not evaluate whether

any of the two approaches is better than the other. Indeed, the CLMS gLAI has been intensively validated with a wide dataset of *in situ* LAI measurements (Copernicus Land Monitoring Service, 2025a) and thus it can be trusted with great confidence.

However, the fraction of LAI that is green ($f_g$) shows larger scatter and with the point cloud in Figure 17 far from the 1:1 line. This deviation has indeed an effect on the comparison of total LAI with larger scatter and a slight positive bias towards the total LAI derived from the CLMS LAI/fAPAR. In order to better understand the different behavior of $f_g$, Figure

18 shows the timeseries for LAI, gLAI and $f_g$ over selected representative ICOS sites. These sites include both temperate and semi-arid conditions and a wide range of biomes: broadleaved and conifer forests, croplands, grasslands, savannas, and orchards/vineyards.



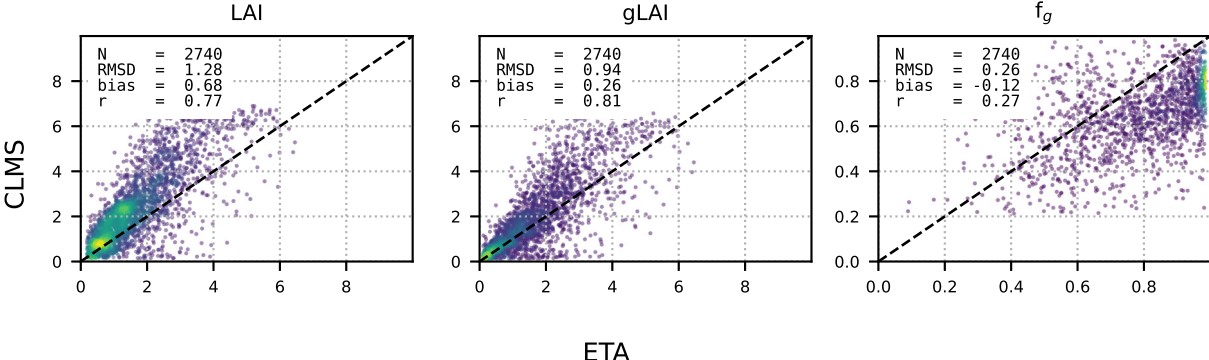

**Figure 17.** Density scatterplot intercomparison between the biophysical processor LAI, green LAI (gLAI) and $f_g$ and those derived from the alternative use of CLMS LAI/FAPAR 300m global products, version 1.

These timeseries confirm that LAI trends are mostly in agreement, with only some bias present in very dense vegetation: croplands during their peak development, broadleaved forests in summer and conifer forests. However, the behavior of $f_g$

shows discrepancies in certain cases. For instance, we would expect that herbaceous croplands (Figure 18a) remain mostly green during the growing phase (spring) with $f_g$ values very close to 1 which then decrease during crop senescence in summer. However, the $f_g$ data derived from the CLMS LAI/fAPAR products shows values significantly lower than one during spring, and sometimes values close to 1 when LAI decreases during senescence. Another inconsistent behaviour was found in the savanna site (Figure 18e), in which the CLMS LAI/fAPAR $f_g$ even shows an opposite trend as one would expect i.e. decrease

of $f_g$ in late spring reaching a minimum in summer where most of the grass layer in this site is dead and only the evergreen oak canopy remains green and then a re-greening with the first rains of autumn. In addition, the $f_g$ derived with the CLMS LAI/fAPAR product seems to be also underestimated in the temperate grassland (Figure 18b) and evergreen forest (Figure 18d), as over these two temperate biomes the canopy should remain mostly green all year round.

  PROSPECT-D model, through an emulator, was also used to derive leaf bihemispherical reflectances and transmittance

(Section 2.4). The evaluation of the performance of this emulator is shown in Figure 19. There is a larger scatter for the NIR reflectances and transmittances, likely due to the fact that we are intentionally excluding the PROSPECT-D leaf structural parameter, that basically controls the multiple scattering within the leaf tissues, which is of a larger magnitude in the NIR region than in the PAR. However, the uncertainties when deriving the reflectances and transmittances seem to be cancelled out when computing the leaf absorptance and therefore the conversion from pigments to leaf spectra seems sufficiently robust.

To derive soil bihemispherical reflectance we converted narrowband reflectance values measured by Sentinel-3 satellites into broadband reflectance in PAR and NIR spectral regions using a linear regression approach (Section 2.4). We evaluated the goodness of this approach by running an independent set of simulations and compared the retrieved broadbands using the coefficients of Table 5. For both platforms, the evaluation confirms the robustness of conversion from the Sentinel-3 TOC





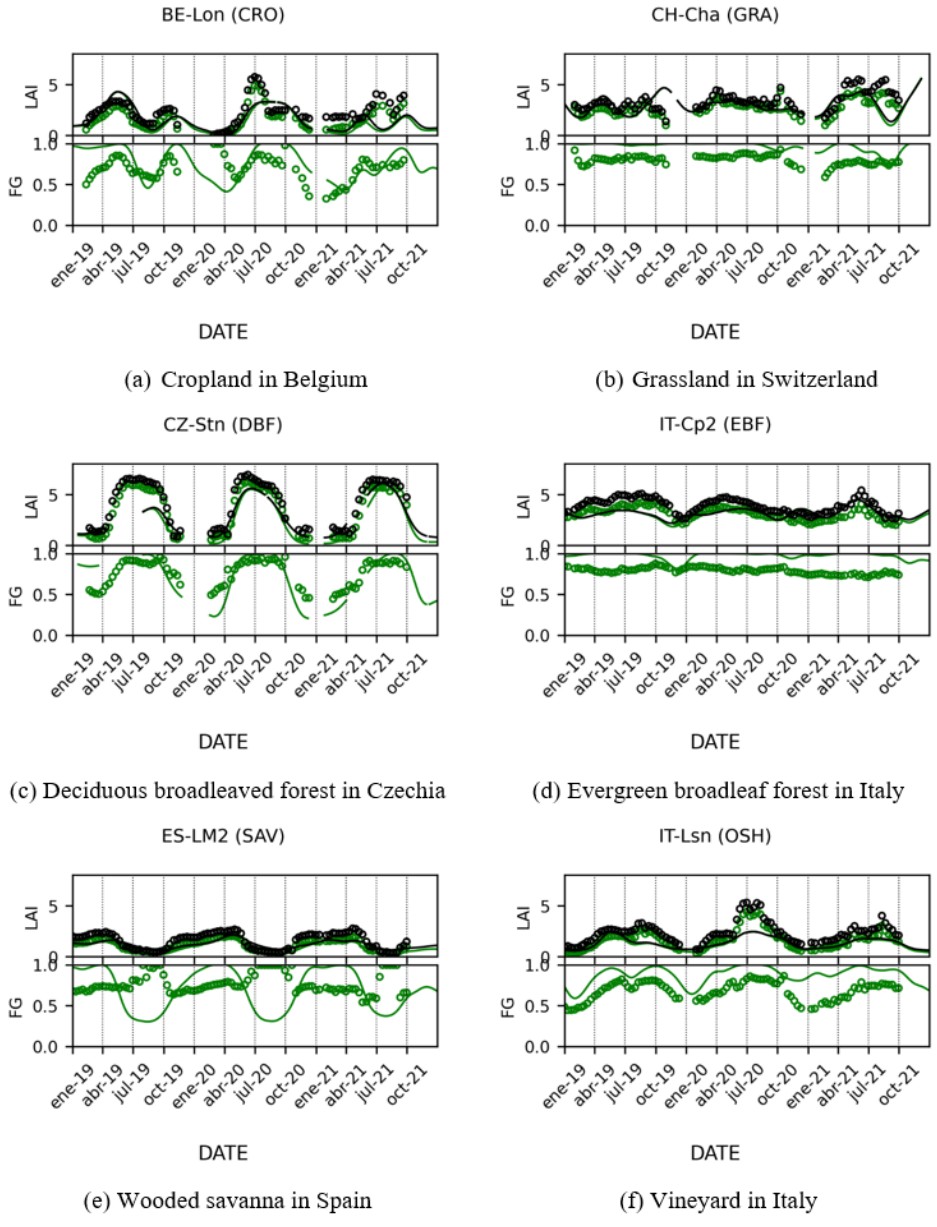

**Figure 18.** Timeseries intercomparison between the LAI (in black), gLAI (in green) and $f_g$ (in green) products retrieved from the biophysical processor (plain line) and those derived from the alternative use of CLMS LAI/fAPAR products (hollow circles).





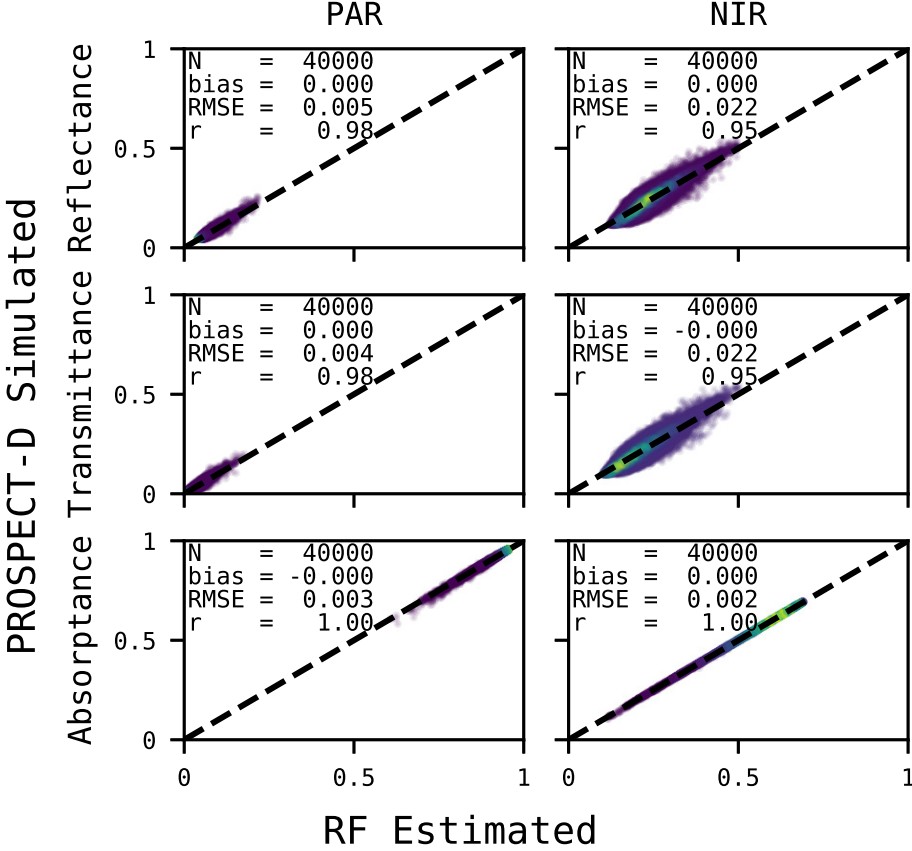

**Figure 19.** Emulator of PROSPECT-D leaf radiative transfer model for the retrieval of broadband leaf reflectance, transmittance and absorptance factors.

reflectances to the broadband PAR, NIR and SW spectral regions, with negligible mean bias ($\approx 0$), very low RMSE and very
good correlation ($\approx 1$) (Figure 20).

### 4.3.2    Land surface temperature sharpening

In the case of CLMS ETa product, the LST sharpening needs to be performed by a ratio of around 3 (i.e. from 1 km to
300 m). Because of this relatively small ratio, the sharpening could potentially be achieved with methods which are more
computationally efficient than DMS. Therefore, we also evaluated the performance of a classic and simple (and therefore
faster) sharpening method called TsHARP (Agam et al., 2007). It relies on finding a linear regression between the Normalized
Difference Vegetation Index (NDVI) and LST. The regression is derived on the whole LST scene to be sharpened with NDVI
resampled to the LST spatial resolution. Afterwards, the linear regression is applied to NDVI at its native resolution to estimate





**Figure 20.** Evaluation of the narrowband to broadband conversion for the estimated Sentinel-3A (top) and Sentinel-3B (bottom) coefficients using the Liang (2001) method. These results are obtained after applying the coefficents in Table 5 to 40000 PROSPECT-D+4SAIL independent simulations.

the representation of LST at this resolution. Finally, a bias correction step is applied to ensure the consistency of LST between the original and sharpened maps.

The two LST sharpening methods were tested in a number of geographically distributed areas of interest (AOI) (Fig. 21 left panel) and across different seasons (images from at least four dates were sharpened at each AOI) to ensure a robust comparison. SY_2_SYN___ (SYN) Sentinel-3 product was used as a proxy for CLMS TOC reflectance product since the latter was still in production at the time this analysis was performed. The two products share the same spectral bands and spatial resolutions. The DMS regression models are trained on the entire Sentinel-3 SYN 3-minute product data unit (PDU) (around 1400 km by

1200 km) as well as on subsets of 30 by 30 LST pixels in a moving window fashion. The evaluation of the sharpening methods was performed using the Sentinel-3 LST (as shown in Fig. 21 right panel) due to the lack of *in situ* LST data which could be used to validate satellite LST with 300 m spatial resolution. The SLSTR LST product was first resampled from 1 km to 3 km



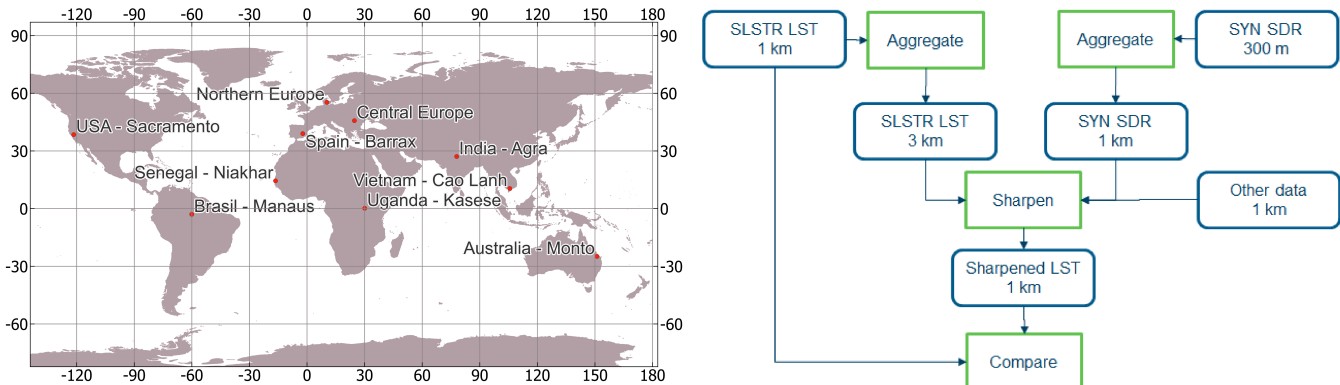

**Figure 21.** Locations of areas of interest (left) and outline of the framework (right) used to evaluate DMS and TsHARP approaches for sharpening SLSTR LST images.

**Table 10.** Accuracy statistics for thermal sharpening: coefficient of determination ($r^2$), Root Mean Square Error (RMSE), Mean Absolute Error (MAE), bias (modelled minus observed) and slope of the linear regression line between modelled and observed values. Model configurations are described in text.

| Model | $r^2$ | RMSE (K) | MAE (K) | Bias (K) | Slope |
|---|---|---|---|---|---|
| TsHARP | 0.93 | 0.85 | 0.59 | -0.02 | 0.99 |
| DMS - Reflectance | 0.95 | 0.74 | 0.52 | -0.02 | 1.00 |
| DMS - WaPOR | 0.95 | 0.75 | 0.53 | -0.01 | 1.00 |
| DMS - WaPOR selected | 0.95 | 0.75 | 0.52 | -0.01 | 1.00 |

and the SYN reflectance product was resampled from 300 m to 1 km, as was the ancillary data. The two sharpening methods were then used to recreate the LST at 1 km resolution, and the resulting map was compared with the original LST product.
This evaluation framework assumes that there are no significant differences in relations between the explanatory variables and the LST when sharpening from 3 km to 1 km and when sharpening from 1 km to 300 m.

The results of the comparison are summarized in Table 10. At all sites, the DMS method produces more accurate sharpened LST compared to TsHARP. The largest difference is at the Central Europe AOI where RMSE of DMS is up to 0.3 K lower than that of TsHARP and Mean Absolute Error (MAE) is up to 0.24 K lower. Looking at all sites, the RMSE of DMS is around 0.1
K lower compared to TsHARP (12% difference) and MAE is around 0.06 K lower (10% difference). Bias is minimal for both methods because bias correction is incorporated in both of them. Both methods also have similar and very high $r^2$, with DMS being slightly better, and the slope of the linear regression between sharpened and original LST very close to 1.

Regarding the three sets of DMS explanatory variables (see Section 2.5 for details), the differences between them are negligible. Looking at the details, there is no site in which "DMS - WaPOR" performs better than the other two configurations,
while "DMS - Reflectance" has slightly better performance at some sites and "DMS - WaPOR selected" at others. However,





the reduction in explanatory variables from 19 ("DMS - Reflectance") to 9 ("DMS - WaPOR selected") results in speed-up of the execution by a factor of around 2.3.

Qualitative assessment of the sharpened LST maps reveals that, in some cases, when variability of NDVI is low, TsHARP fails to find a meaningful relation between NDVI and LST (i.e. the linear regression has a slope close to 0). In such cases, the
resulting sharpened LST is mainly an output of the bias correction step and has a blurry appearance consistent with simple resampling of lower resolution LST to higher resolution. In those cases, the quantitative analysis might still result in good accuracy statistics due to a small difference in spatial resolution between 3 km and 1 km LST. DMS is less sensitive to, but not fully unaffected by, this issue because it relies on a range of spectral bands, indices, and DEM-based datasets. It is also noticeable that, in some cases, when there is insufficient information in the spectral data, DMS relies too heavily on
DEM-based datasets, which can result in artifacts in the sharpened LST. One of the root causes of this issue could be the insufficient or incorrect atmospheric correction applied to SYN spectral bands. For operational production of CLMS ETa (and for production of data validated in Section 3) the SYN product is replaced by CLMS TOC reflectance product. The latter has shown improved agreement with in-situ measurements (Copernicus Land Monitoring Service, 2025c) and could therefore lead to an improvement in LST sharpening in such cases.

### 4.3.3   Weather forcing and ETa gap-filling

The suitability of the topographically corrected CAMS forecast data (Section 2.6) for modelling of actual evapotranspiration was assessed by comparing modelled daily solar irradiance and reference ET against measurements from 45 EC flux towers located in western Europe (France, Belgium, western Germany, Switzerland Spain, and northern Italy), United States and Australia for the year 2020. Those towers represent various topographical conditions from flat and low-lying to mountainous
terrain as well as different climates, from arid to temperate. The statistical results of this comparison are shown in Fig. 22 and confirm the applicability of CAMS forecast for ETa modelling and suitability of the correction methods.

Apart from forcing the ET models, CAMS data was used for gap-filling of modelled ETa maps through the use of reference ET and precipitation (PR) (Section 2.9). We evaluated three gap-filling approaches: ignoring rainfall (called $GF_{ETr}$ - (Guzinski et al., 2021)); taking rainfall into account as described in Guzinski et al. (2023) (called $GF_{ETr+PR}$); conservative modification
of $GF_{ETr+PR}$ described in Section 2.9 (called $GF_{ETr+PR80}$). We also evaluated near real-time (NRT) gap-filling periods with different maximum durations: 15 days (as used by default in Sen-ET approach), 30 days and 60 days (as used in WaPOR during reprocessing). Finally, a non-time-critical (NTC) gap-filling with a maximum duration of 60 days (i.e. up to 60 days before and after target date) was also evaluated.

To evaluate the behaviour of the methods in diverse climates, we performed an analysis using in-situ ET measured at
geographically distributed ICOS stations (Table B3). The daily ET at ICOS stations was calculated by summing up good quality instantaneous ET at hourly or half-hourly timesteps. Dates on which more than two measurements were of poor quality were ignored and no correction for lack of energy balance closure was performed. The determination of quality of ET data was based on ICOS quality flags associated with each measurement. Reference ET and rainfall sums were calculated from





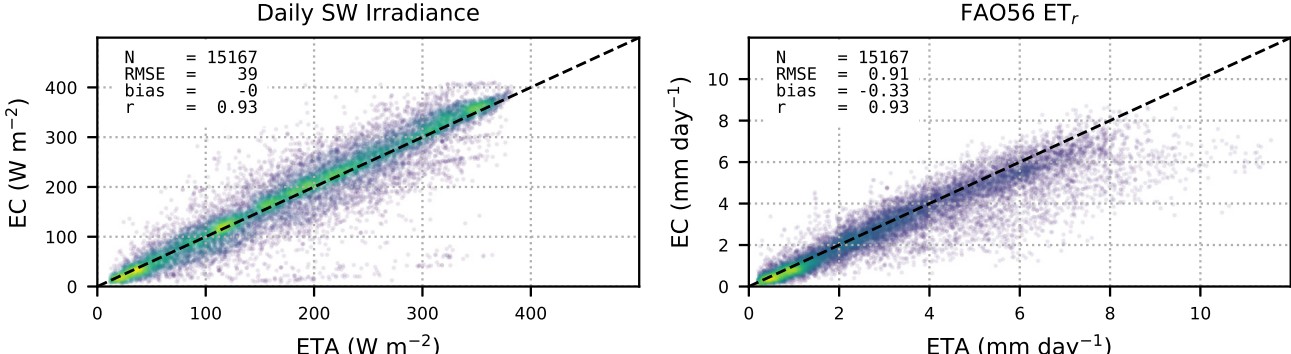

**Figure 22.** Comparison between daily solar irradiance (SW-IN-DD panel in $\mathrm{W m^2}$) and reference ET (ETR panel in $\mathrm{mm\,day^{-1}}$) modelled with topograhically corrected CAMS forecast data and measured in 45 representative Eddy Covariance stations located in western Europe, United States and Australia.

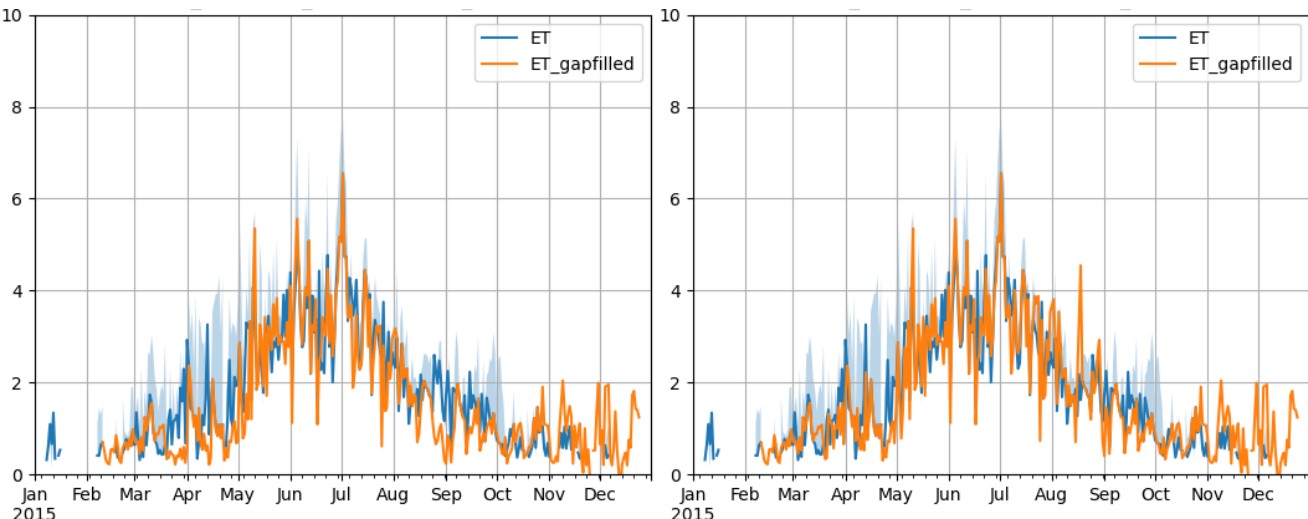

**Figure 23.** An example of gap-filled ET timeseries at Voulundgaard ICOS station in 2015. Maximum gap-filling window of 60 days was used with, at left, gap-filling with rainfall not taken into account (method $\mathrm{GF}_{ETr}$) and, at right, gap-filling with rainfall taken into account using the best performing method (method $\mathrm{GF}_{ETr+PR80}$). Most notable differences are in spring (March - April) and late summer (July - August).

the stations' meteorological measurements. Cloudy days were identified as those for which measured solar irradiance was less than 80% of theoretical surface solar irradiance in clear-sky conditions.

Table 11 presents the results on this evaluation. Statistics are calculated on daily basis and only for data points which were gap-filled and for which the rainfall adjustment is relevant. Neglecting rainfall can introduce an overall negative bias (underes-





**Table 11.** Accuracy statistics for gap-filling of evapotranspiration on cloudy days using ICOS in-situ data and different methods (described in text) and maximum window sizes (in NRT mode apart from last rows).

| Window size | Method | RMSE (mm day$^{-1}$) | Bias (mm day$^{-1}$) | r |
|---|---|---|---|---|
| 15 | $GF_{ETr}$ | 0.62 | -0.17 | 0.86 |
| | $GF_{ETr+PR}$ | **0.59** | **-0.05** | **0.87** |
| | $GF_{ETr+PR80}$ | **0.59** | -0.09 | **0.87** |
| 30 | $GF_{ETr}$ | 0.63 | -0.18 | 0.87 |
| | $GF_{ETr+PR}$ | 0.61 | **0.03** | 0.87 |
| | $GF_{ETr+PR80}$ | **0.59** | -0.06 | **0.88** |
| 60 | $GF_{ETr}$ | 0.63 | -0.17 | 0.87 |
| | $GF_{ETr+PR}$ | 0.65 | 0.13 | 0.86 |
| | $GF_{ETr+PR80}$ | **0.58** | **-0.02** | **0.88** |
| 60 NTC | $GF_{ETr}$ | 0.56 | -0.11 | 0.89 |
| | $GF_{ETr+PR}$ | 0.72 | 0.35 | 0.87 |
| | $GF_{ETr+PR80}$ | **0.54** | **0.04** | **0.90** |

timation) of up to -0.18 mm day$^{-1}$ on wet, gap-filled dates. When rainfall is taken into account the introduced bias can become negligible and this reduction in bias leads to a less significant reduction in introduced uncertainty. Method $GF_{ETr+PR}$ per-

forms well with shorter window size but performance degrades significantly as window size increases and for largest windows the uncertainty is higher than for method $GF_{ETr}$. Method $GF_{ETr+PR80}$ results in lowest introduced bias and uncertainty with 60 days window size, chosen for the CLMS ETa processing chain. Finally, the NTC gap-filling performs better than NRT gap-filling and with method $GF_{ETr+PR80}$ performing the best in this case. An example timeseries of ET gap-filled with methods $GF_{ETr}$ and $GF_{ETr+PR80}$ is shown in Figure 23.

**4.4   Potential improvements to the CLMS ETa product**

The CLMS ETa product will enter operational phase before the end of 2025. Therefore, it is not feasible to introduce significant changes for the initial version of the product. However, like most CLMS products, ETa might undergo evolution and reprocessing in the coming years and, therefore, it is worth to highlight some potential areas of improvement.

One of the main limitations in producing medium- and high spatial resolution ET products (below 1 km) is the lack of LST

observations with high spatio-temporal resolution, especially within the Copernicus Sentinel satellites. This situation should be resolved by the end of the decade when Land Surface Temperature Monitoring (LSTM) mission, with a primary objective of frequent monitoring of field-scale ETa, will join the Copernicus constellation (Koetz et al., 2018). In the meantime, it might be advantageous to make use of well established non-Copernicus sensors, such as VIIRS on board of Suomi-NPP satellite which can provide daily observations of LST with spatial resolution of 375 m.





Another potential improvement would be to compute a temporal running mean of biophysical parameters and TOC reflectance using the latest week of data. This should result in two benefits. Firstly, inverting the PROSPECT-D+4SAIL RTM (see Section 2.3) is an ill-posed problem, meaning that multiple combinations of biophysical parameters can result in the same reflectance. By taking a mean of multiple inversions, under the assumption that the biophysical conditions of vegetation do not change significantly over short time periods, the robustness of this inversion can be improved. Secondly, this will allow the full

use of SLSTR swath with VZA below 35°by gap-filling the OLCI image on the eastern edge of the swath (see Section 2.2).

       Producing a reanalysis dataset, e.g. two months after the completion of NRT production, might also lead to improvements in the accuracy of the CLMS ETa product. Firstly, because it would allow the use of ERA5 reanalysis meteorological data (Hersbach et al., 2020) instead of CAMS forecast as ET model forcing. The former has improved accuracy and increased spatial resolution compared to the latter and is therefore expected to result in more robust ET estimates. Secondly, gap-filling

could be performed using 60 days both preceding and succeeding the target date which will result in less cloud gaps in the final ETa maps but also in improved accuracy (compare two bottom rows of Table 11).

       Finally, modelling of evaporation of inland water bodies (and potentially of snow) can be important for many applications, such as water resources management or SDG reporting. Therefore, a post-processing step could be introduced during which this flux could be estimated, e.g. using Penman model (Monteith and Unsworth, 2013) parameterized for water.

**5   Conclusions**

A global and operational actual evapotranspiration product will be introduced to the portfolio of the Copernicus Land Monitoring Service by the end of 2025. This addition is motivated by a request from the Food and Agriculture Organization of the United Nations but the new product will have a multitude of applications in water resources management, SDG reporting, food security, forest management and other fields. The product is designed to have a 300 m spatial resolution, dekadal temporal

resolution, global extent and to be produced in near-real-time with 2-day delay (Table 1). As all other CLMS products, it will be distributed with a free and open license and will have guaranteed long-term continuity.

       The product will be based almost exclusively of Copernicus data, ranging from imagery acquired by OLCI and SLSTR sensors on board of Sentinel-3 satellites through meteorological forecast data provided by Copernicus Atmosphere Monitoring Service to higher-level products such as land cover maps produced by Copernicus Land Monitoring Service (Table 3).

Those data undergo significant pre-processing to turn them into input forcing (Table 2) for two ET models: TSEB-PT and ETLook (Section 2). The input data and the pre-processing methods were selected to be applicable globally and evaluation was performed to confirm that this is the case (Section 4.3).

       A demonstration dataset of one year was produced using the prototype end-to-end processing chain for the CLMS ETa product, and validated using measurements from 104 globally distributed eddy covariance stations (Section 3). Results of this

comparison are encouraging with overall best RMSE of 0.80 mm/day (for the ensemble TSEB-PT - ETLook ETa), bias of -0.12 mm/day (for TSEB-PT ETa) and coefficient of determination of 0.84 (for the ensemble TSEB-PT - ETLook ETa). The CLMS ETa prototype also compared favourably with the global WaPOR ETa maps produced by FAO, which it is meant to replace,



and other higher-resolution ETa datasets (Section 4.1). The addition of ETa product in the CLMS portfolio should therefore significantly enlarge the CLMS user community.

*Code availability.*  The biophysical processor based on PROSPECTD+4SAIL described in Section 2.3 is available at https://github.com/hectornieto/pyPro4Sail with the specific version used in this study being https://github.com/hectornieto/pypro4sail/releases/tag/v1.2 (Nieto, 2025).

The Data Mining Sharpener used to sharpen LST described in Section 2.5 is available at https://github.com/radosuav/pyDMS with the specific version used in this study being https://github.com/radosuav/pyDMS/releases/tag/v1.2.

The code used to access and topographically correct meteorological forcings described in Section 2.6 is available at https://github.com /hectornieto/meteo_utils/ with the specific version used in this study being https://github.com/hectornieto/meteo_utils/releases/tag/v2.1.1 (Nieto et al., 2025b) .

The implementation of TSEB-PT model described in Section 2.8.1 is available at https://github.com/hectornieto/pyTSEB the specific version used in this study being https://github.com/hectornieto/pyTSEB/releases/tag/v2.3 (Nieto et al., 2025a) .

The implementation of ETLook model described in Section 2.8.2 is available at https://bitbucket.org/cioapps/pywapor/ with the specific version used in this study being https://github.com/DHI-GRAS/pywapor/releases/tag/prototype.



**Appendix A:  Estimation of canopy transmittance and reflectance**

The direct-hemispherical spectral transmittance ($\tau_{C,DIR,\lambda}$) at a given solar zenith angle ($\theta_S$) is calculated following the equations of Campbell and Norman (1998) for a single layer canopy:

$$\tau_{C,DIR,\lambda}(\theta_S) = \frac{\left(\rho^*_{C,\lambda}(\theta_S)^2 - 1\right)\exp\left(-\sqrt{\zeta_\lambda}\kappa_b(\theta_S)LAI\right)}{\left(\rho^*_{C,\lambda}\rho_{S,\lambda} - 1\right) + \rho^*_{C,\lambda}(\psi)\left(\rho^*_{C,\lambda}(\theta_S) - \rho_{S,\lambda}\right)\exp\left(-2\sqrt{\zeta_\lambda}\kappa_b(\theta_S)LAI\right)} \quad (A1)$$

with $\lambda$ being either the PAR or NIR. $\rho^*_{C,\lambda}(\psi)$ is the beam spectral reflection coefficient for a deep canopy with non-horizontal leaves (see Eq. A2), $\zeta_\lambda$ is the leaf absorptivity, $\kappa_b$ is the extinction coefficient for direct-beam radiation (per LAI unit), and $\rho_{S,\lambda}$ is the soil spectral reflectance. The multiple scattering between the soil and the canopy is accounted for in the $\rho^*_{C,\lambda}$ and $\rho_{S,\lambda}$ terms.

$$\rho^*_{C,\lambda}(\theta_S) = \frac{2\kappa_b(\theta_S)\rho^H_\lambda}{\kappa_b(\theta_S) + 1} \quad (A2)$$

$\rho^H_\lambda = \frac{1-\sqrt{\zeta_\lambda}}{1+\sqrt{\zeta_\lambda}}$ is the reflectance factor for a canopy with horizontal leaves.

Finally, the canopy beam extinction $\kappa_b(\psi)$ is calculated based on the ellipsoidal LADF of Campbell (1990):

$$\kappa_b(\theta_S) = \frac{\sqrt{\chi^2 + \tan^2\theta_S}}{\chi + 1.774(\chi + 1.182)^{-0.733}} \quad (A3)$$

Diffuse spectral transmittance ($\tau_{C,DIF,\lambda}$) is calculated by numerically integrating $\kappa_b$ over the hemisphere:

$$\kappa_d = 2\int_0^\pi \kappa_b(\psi)\sin\psi\cos\psi d\psi \quad (A4)$$

and replacing $\kappa_b$ by $\kappa_d$ in Eq. A1.

Similarly the canopy direct spectral albedo is computed as:

$$\rho_{C,DIR,\lambda}(\theta_S) = \frac{\rho^*_{C,\lambda}(\theta_S) + \left[\frac{\rho^*_{C,\lambda}(\theta_S) - \rho_{S,\lambda}}{\rho^*_{C,\lambda}(\theta_S)\rho_{s,\lambda} - 1}\right]\exp\left(-2\sqrt{\zeta_\lambda}\kappa_b(\theta_S)LAI\right)}{1 + \rho^*_{C,\lambda}(\theta_S) + \left[\frac{\rho^*_{C,\lambda}(\theta_S) - \rho_{s,\lambda}}{\rho^*_{C,\lambda}(\theta_S)\rho_{S,\lambda} - 1}\right]\exp\left(-2\sqrt{\zeta_\lambda}\kappa_b(\theta_S)LAI\right)} \quad (A5)$$

and the diffuse canopy albedo ($\rho_{C,DIF,\lambda}$) by replacing $\kappa_b(\theta_S)$ by $\kappa_d$

**Appendix B:  List of flux towers**

Tables B1 and B2 lists all the EC flux tower sites used to validate CLMS ETa datasets. The ICOS datasets used in the analysis were the ETC L2 FLUXNET (ICOS RI et al., 2025) and the Warm Winter 2020 (ICOSww - Warm Winter 2020 Team et al.





(2022)). AsiaFlux site is described in Meijide et al. (2017). The DOIs of AmeriFlux sites are listed below:

CA-DBB: 10.17190/AMF/1881565

US-ARM: 10.17190/AMF/1854366

US-Bi1: 10.17190/AMF/1871134

US-Bi2: 10.17190/AMF/1871135

US-BZF: 10.17190/AMF/1881570

US-HB3: 10.17190/AMF/2229378

US-Jo1: 10.17190/AMF/1902833

US-Me6: 10.17190/AMF/2204871

US-Mo1: 10.17190/AMF/2229382

US-Mo2: 10.17190/AMF/2229383

US-Mo3: 10.17190/AMF/2229384

US-ONA: 10.17190/AMF/1832163

US-Rls: 10.17190/AMF/2229387

US-Ro4: 10.17190/AMF/1881589

US-Ro5: 10.17190/AMF/1818371

US-Ro6: 10.17190/AMF/1881590

US-Sne: 10.17190/AMF/1871144

US-SRG: 10.17190/AMF/2204877

US-Ton: 10.17190/AMF/2204880

US-Tw4: 10.17190/AMF/2204881

US-Var: 10.17190/AMF/1993904

US-Whs: 10.17190/AMF/1984574

US-Wkg: 10.17190/AMF/1984575

US-xAE: 10.17190/AMF/1985434

US-xDC: 10.17190/AMF/1985437

US-xDS: 10.17190/AMF/1985439

US-xJE: 10.17190/AMF/1985443

US-xSB: 10.17190/AMF/1985451

US-xSE: 10.17190/AMF/1985452

US-xST: 10.17190/AMF/1985454

US-xTA: 10.17190/AMF/1985455

US-xWD: 10.17190/AMF/2229412

Table B3 lists selected sites from ICOS WarmWinter2020 database which were used to evaluate biophysical modelling and ET gap-filling approaches.





**Table B1.** Geographical location, climate region (Köppen classification), plant functional type (PFT) and network/dataset of origin for EC flux towers used for CLMS ETa model validation - part A.

| Id | Site | Latitude | Longitude | Climate | PFT | Network |
|----|------|----------|-----------|---------|-----|---------|
| 1 | AT-Inn | 47.2641 | 11.3858 | Dfb | URB | EFDC |
| 2 | AT-Neu | 47.1167 | 11.3175 | Dfb | GRA | EFDC |
| 3 | AT-VnA | 48.1818 | 16.3909 | Dfb | URB | EFDC |
| 4 | AU-ASM | -22.2828 | 133.2493 | BWh | SAV | OzFlux |
| 5 | AU-Boy | -32.4771 | 116.9386 | Csa | EBF | OzFlux |
| 6 | AU-Cum | -33.6152 | 150.7236 | Cfa | EBF | OzFlux |
| 7 | AU-DaS | -14.1592 | 131.3881 | Aw | SAV | OzFlux |
| 8 | AU-Dry | -15.2588 | 132.3706 | Aw | SAV | OzFlux |
| 9 | AU-GWW | -30.1913 | 120.6541 | BWh | SAV | OzFlux |
| 10 | AU-Lit | -13.1790 | 130.7945 | Aw | SAV | OzFlux |
| 11 | AU-Rgf | -32.5061 | 116.9668 | Csa | CRO | OzFlux |
| 12 | AU-Stp | -17.1507 | 133.3502 | BSh | GRA | OzFlux |
| 13 | AU-Whr | -36.6732 | 145.0294 | BSk | EBF | OzFlux |
| 14 | AU-Wom | -37.4222 | 144.0944 | Cfb | EBF | OzFlux |
| 15 | BE-Lcr | 51.1122 | 3.8504 | Cfb | DBF | EFDC |
| 16 | BE-Lon | 50.5516 | 4.7462 | Cfb | CRO | ICOS |
| 17 | BE-Maa | 50.9801 | 5.6319 | Cfb | CSH | EFDC |
| 18 | CA-Cbo | 44.3167 | -79.9333 | Dfb | DBF | AmeriFlux |
| 19 | CA-DB2 | 49.1190 | -122.9951 | Csb | WET | AmeriFlux |
| 20 | CA-DBB | 49.1293 | -122.9849 | Csb | WET | AmeriFlux |
| 21 | CH-Aws | 46.5832 | 9.7904 | Dfc | GRA | EFDC |
| 22 | CH-Cha | 47.2102 | 8.4104 | Dfb | GRA | ICOSww |
| 23 | CH-Fru | 47.1158 | 8.5378 | Dfb | GRA | EFDC |
| 24 | CH-Oe2 | 47.2864 | 7.7338 | Dfb | CRO | EFDC |
| 25 | CZ-BK1 | 49.5021 | 18.5369 | Dfb | ENF | ICOSww |
| 26 | CZ-KrP | 49.5733 | 15.0788 | Dfb | CRO | ICOSww |
| 27 | CZ-RAJ | 49.4437 | 16.6965 | Dfb | ENF | ICOSww |
| 28 | CZ-wet | 49.0247 | 14.7704 | Dfb | WET | ICOSww |
| 29 | DE-BeR | 52.4572 | 13.3158 | Dfb | URB | EFDC |
| 30 | DE-Geb | 51.0997 | 10.9146 | Dfb | CRO | EFDC |
| 31 | DE-Gri | 50.9500 | 13.5126 | Dfb | GRA | ICOSww |
| 32 | DE-Hai | 51.0792 | 10.4522 | Dfb | DBF | ICOSww |
| 33 | DE-Hdn | 53.8683 | 13.2685 | Dfb | CRO | EFDC |
| 34 | DE-HoH | 52.0853 | 11.2192 | Dfb | DBF | ICOSww |
| 35 | DE-Kli | 50.8931 | 13.5224 | Dfb | CRO | ICOSww |
| 36 | DE-Obe | 50.7867 | 13.7213 | Dfb | ENF | ICOSww |
| 37 | DE-RuR | 50.6219 | 6.3041 | Dfb | GRA | ICOSww |
| 38 | DE-RuS | 50.8659 | 6.4471 | Cfb | CRO | ICOSww |
| 39 | DE-Tha | 50.9626 | 13.5652 | Dfb | ENF | ICOSww |
| 40 | DE-Zrk | 53.8759 | 12.8890 | Dfb | WET | EFDC |
| 41 | DK-Sor | 57.2331 | 9.8446 | Cfb | DBF | EFDC |
| 42 | ES-Abr | 38.7018 | -6.7859 | Csa | SAV | ICOSww |
| 43 | ES-Cnd | 39.2242 | -0.9031 | BSk | EBF | ICOSww |
| 44 | ES-LJu | 37.0979 | -2.9658 | BSh | OSH | ICOSww |
| 45 | ES-LM1 | 39.9427 | -5.7787 | BSk | SAV | ICOSww |
| 46 | ES-LM2 | 39.9346 | -5.7759 | BSk | SAV | ICOSww |
| 47 | FI-Hyy | 61.8474 | 24.2948 | Dfc | ENF | ICOS |
| 48 | FI-Kmp | 60.2029 | 24.9611 | Dfb | URB | EFDC |
| 49 | FI-Let | 60.6418 | 23.9595 | Dfc | ENF | EFDC |
| 50 | FI-Sii | 61.8327 | 24.1929 | Dfc | WET | ICOS |
| 51 | FR-Bil | 44.4937 | -0.9561 | Cfb | ENF | ICOS |
| 52 | FR-Fon | 48.4764 | 2.7801 | Cfb | DBF | ICOS |
| 53 | FR-Gri | 48.8442 | 1.9519 | Cfb | CRO | ICOSww |
| 54 | FR-Hes | 48.6741 | 7.0647 | Cfb | DBF | ICOSww |
| 55 | FR-Lam | 47.3229 | 2.2841 | Cfa | CRO | ICOSww |
| 56 | FR-LGt | 44.7171 | -0.7693 | Cfb | WET | EFDC |
| 57 | FR-Mej | 48.1184 | -1.7963 | Cfb | GRA | EFDC |
| 58 | IE-Cra | 53.3231 | -7.6418 | Cfb | WET | ICOSww |
| 59 | IL-Yat | 31.3450 | 35.0520 | BSh | ENF | ICOSww |
| 60 | IT-BCi | 40.5238 | 14.9574 | Csa | CRO | ICOSww |
| 61 | IT-Cp2 | 41.7043 | 12.3573 | Csa | EBF | ICOSww |
| 62 | IT-Lsn | 45.7405 | 12.7503 | Cfa | OSH | ICOSww |




**Table B2.** Geographical location, climate region (Köppen classification), plant functional type (PFT) and network/dataset of origin for EC flux towers used for CLMS ETa model validation - part B.

| Id | Site | Latitude | Longitude | Climate | PFT | Network |
|----|------|----------|-----------|---------|-----|---------|
| 63 | IT-OXm | 43.7745 | 11.2552 | Csa | URB | EFDC |
| 64 | IT-SR2 | 43.7320 | 10.2909 | Csa | ENF | EFDC |
| 65 | IT-Tor | 45.8444 | 7.5781 | Dfb | GRA | ICOSww |
| 66 | JOP | -1.6931 | 103.3914 | Af | CRO | AsiaFlux |
| 67 | RAGOLA | 14.4944 | -16.4563 | BSh | GRA | AMMA |
| 68 | RU-Fy2 | 56.4476 | 32.9019 | Dfb | ENF | ICOSww |
| 69 | SE-Deg | 64.1820 | 19.5565 | Dfc | WET | ICOSww |
| 70 | SE-Nor | 60.0865 | 17.4795 | Dfb | ENF | ICOS |
| 71 | SE-Ros | 64.1725 | 19.7380 | Dfc | ENF | ICOSww |
| 72 | SE-Svb | 64.2561 | 19.7745 | Dfc | ENF | ICOS |
| 73 | UK-LBT | 51.5215 | -0.1389 | Cfb | URB | EFDC |
| 74 | US-ARM | 36.6058 | -97.4888 | Cfa | CRO | AmeriFlux |
| 75 | US-Bi1 | 38.0992 | -121.4993 | Csa | CRO | AmeriFlux |
| 76 | US-Bi2 | 38.1091 | -121.5351 | Csa | CRO | AmeriFlux |
| 77 | US-BZF | 64.7013 | -148.3121 | Dfc | WET | AmeriFlux |
| 78 | US-HB3 | 33.3482 | -79.2322 | Cfa | ENF | AmeriFlux |
| 79 | US-Jo1 | 32.5820 | -106.6350 | BWk | OSH | AmeriFlux |
| 80 | US-Me6 | 44.3233 | -121.6078 | Dsb | ENF | AmeriFlux |
| 81 | US-Mo1 | 39.2298 | -92.1167 | Dfa | CRO | AmeriFlux |
| 82 | US-Mo2 | 38.9488 | -91.9945 | Dfa | GRA | AmeriFlux |
| 83 | US-Mo3 | 39.2322 | -92.1493 | Dfa | CRO | AmeriFlux |
| 84 | US-ONA | 27.3836 | -81.9509 | Cfa | GRA | AmeriFlux |
| 85 | US-Rls | 43.1439 | -116.7356 | BSk | CSH | AmeriFlux |
| 86 | US-Ro4 | 44.6781 | -93.0723 | Dfa | GRA | AmeriFlux |
| 87 | US-Ro5 | 44.6910 | -93.0576 | Dfa | CRO | AmeriFlux |
| 88 | US-Ro6 | 44.6946 | -93.0578 | Dfa | CRO | AmeriFlux |
| 89 | US-Sne | 38.0369 | -121.7547 | Csa | GRA | AmeriFlux |
| 90 | US-SRG | 31.7894 | -110.8277 | BSh | GRA | AmeriFlux |
| 91 | US-Ton | 38.4309 | -120.9660 | Csa | WSA | AmeriFlux |
| 92 | US-Tw4 | 38.1027 | -121.6413 | Csa | WET | AmeriFlux |
| 93 | US-Var | 38.4133 | -120.9508 | Csa | GRA | AmeriFlux |
| 94 | US-Whs | 31.7438 | -110.0522 | BSk | OSH | AmeriFlux |
| 95 | US-Wkg | 31.7365 | -109.9419 | BSk | GRA | AmeriFlux |
| 96 | US-xAE | 35.4106 | -99.0588 | Cfa | GRA | AmeriFlux |
| 97 | US-xDC | 47.1617 | -99.1066 | Dwb | GRA | AmeriFlux |
| 98 | US-xDS | 28.1250 | -81.4362 | Cfa | CVM | AmeriFlux |
| 99 | US-xJE | 31.1948 | -84.4686 | Cfa | ENF | AmeriFlux |
| 100 | US-xSB | 29.6893 | -81.9934 | Cfa | ENF | AmeriFlux |
| 101 | US-xSE | 38.8901 | -76.5600 | Cfa | DBF | AmeriFlux |
| 102 | US-xST | 45.5089 | -89.5864 | Dfb | DBF | AmeriFlux |
| 103 | US-xTA | 32.9505 | -87.3933 | Cfa | ENF | AmeriFlux |
| 104 | US-xWD | 47.1282 | -99.2414 | Dwb | GRA | AmeriFlux |

**Appendix C: Statistical metrics and Taylor plot scores for dekadal ETa per model, when only sites with fluxes corrected for the EBC problem (AmeriFlux, ICOS, OzFlux networks) were analyzed.**

Table C1 and Figure C1 show validation of modelled dekadal ETa (similarly to Table 8 and Figure 6) but only against the flux towers for which ECB correction was applied.





**Table B3.** Selected sites from ICOS WarmWinter2020 database (Warm Winter 2020 Team et al., 2022) used to evaluate biophysical modelling and ET gap-filling approaches. In addition to geographical location the table shows climate region (Köppen classification) and plant functional type (PFT) of each site.

| Site | Latitude | Longitude | Climate | PFT |
|------|----------|-----------|---------|-----|
| GF-Guy | 5.2787 | -52.9248 | Af | EBF |
| CD-Ygb | 0.8144 | 24.5024 | Af | MF |
| IT-MBo | 46.0146 | 11.0458 | Dfc | GRA |
| DK-Vng | 56.0374 | 9.1607 | Cfb | CRO |
| ES-LM2 | 39.9345 | -5.7758 | Csa | SAV |
| FR-Bil | 44.4936 | -0.9560 | Cfb | ENF |
| SE-Deg | 64.1820 | 19.5565 | Dfc | WET |
| FR-Lam | 43.4964 | 1.2378 | Cfb | CRO |
| CZ-Lnz | 48.6815 | 16.9463 | Dfb | DBF |
| SE-Nor | 60.0865 | 17.4795 | Dfb | ENF |
| ES-Agu | 36.9393 | -2.0340 | Csa | OSH |
| GL-Dsk | 69.2534 | -53.5141 | ET | OSH |
| JP-Bby | 43.3229 | 141.8107 | Cfa | WET |
| IT-BCi | 40.5237 | 14.9574 | Cfa | CRO |
| IL-Yat | 31.3450 | 35.0519 | BSh | ENF |
| GL-ZaH | 74.4733 | -20.5508 | ET | GRA |
| JP-Ozm | 34.5634 | 135.5334 | Dfa | URB |
| DE-Gri | 50.9500 | 13.5125 | Dfv | GRA |
| NL-Loo | 52.1664 | 5.7435 | Cfb | ENF |

**Table C1.** Bias (mm/day), RMSE (mm/day), and $r^2$ scores for dekadal ETa per model (all sites combined), and summary statistics at site level. rBias and rRMSE are relative metrics (i.e., divided by mean measured ET).

| | | | | | | | Summary scores at site level | | | | | |
|---|---|---|---|---|---|---|---|---|---|---|---|---|
| **Model** | **N** | **Bias** | **RMSE** | **rBias** | **rRMSE** | **$r^2$** | **Bias** | | **RMSE** | | **$r^2$** | |
| | | | | | | | min | max | min | max | mean | St.Dev |
| TSEB-PT | 2424 | -0.07 | 0.88 | -0.04 | 0.49 | 0.79 | -1.04 | 1.39 | 0.28 | 2.24 | 0.78 | 0.25 |
| ETLook | 2424 | -0.46 | 1.12 | -0.25 | 0.63 | 0.78 | -0.72 | 2.56 | 0.27 | 2.94 | 0.79 | 0.23 |
| Ensemble | 2424 | -0.26 | 0.80 | -0.15 | 0.45 | 0.85 | -0.66 | 1.98 | 0.28 | 2.22 | 0.83 | 0.20 |
| WaPOR | 2424 | 0.12 | 0.88 | 0.07 | 0.49 | 0.81 | -1.01 | 1.53 | 0.26 | 1.77 | 0.76 | 0.26 |



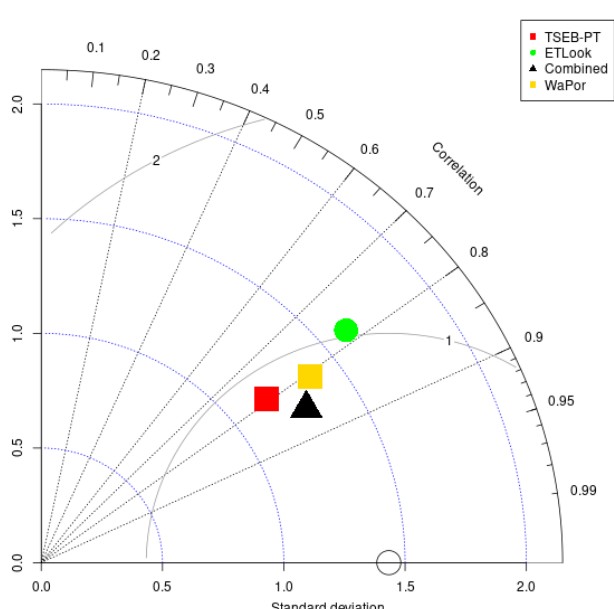

**Figure C1.** Taylor-plot general overview. The circle on the x-axis is the standard deviation of the eddy covariance towers measurements.



*Author contributions.* RG contributed to the input forcing analysis, integration of the processing chain and data production as well as the pyDMS code. HN contributed to the input forcing analysis and pyTSEB, pyPro4SAIL and meteo_utils code. JMB, FGM and JDP contributed to in-situ and WaPOR data collection and model validation. WG contributed to evaluation of LST sharpening methods. RL contributed to the
funding acquisition and coordinated the presented work.

*Competing interests.* Authors declare no competing interests.

*Acknowledgements.* The presented work was funded by Specific Contract No 4 of the implementing framework contract No 945120 – IPR – 2023 with the European Union represented by European Commission, Directorate-General Joint Research Centre. We would like to acknowledge the helpful interactions with Livia Peiser, Jippe Hoogeveen and Bert Coerver from FAO WaPOR team and their support with
access to the WaPOR/ETLook code.

This work utilized data from the Integrated Carbon Observation System (ICOS - https://www.icos-cp.eu/), the OzFlux network (https://www.ozflux.org.au/) which is supported by the Australian Terrestrial Ecosystem Research Network (TERN - http://www.tern.org.au), the AmeriFlux network (https://ameriflux.lbl.gov/, funding for the AmeriFlux data portal was provided by the U.S. Department of Energy Office
of Science), the European Fluxes Database Cluster (EFDC - https://www.europe-fluxdata.eu/), l'observatoire AMMA-CATCH (Analyse Multidisciplinaire de la Mousson Africaine - Couplage de l'Atmosphère Tropicale et du Cycle Hydrologique - https://www.amma-catch.org/) and the AsiaFlux network (https://asiaflux.net/index.php).





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
