# Peer review of "Towards a global actual evapotranspiration product for the Copernicus Land Monitoring Service"

_EGUsphere, 2025_

## Referee Comment (RC2)

I thoroughly enjoyed reading this paper; it is very informative and descriptive. I recommend that the editor consider it for publication, subject to addressing some clarification questions and minor corrections. In my view, the paper is somewhat lengthy because several figures present overlapping information, which could be streamlined to make the main text more concise.

- 1. Other than the requirements mentioned in line 50, which align with the applications of the WaPOR global ETa dataset, I was wondering why a 10-day interval was selected. Since the revisit time of Sentinel-3 instruments is around 2 days at the equator, a finer temporal resolution seems feasible. While the revisit frequency decreases at higher latitudes, the chances of acquiring clear observations, especially during morning overpasses, are already quite low. Therefore, I'm curious why a shorter revisit interval was not targeted. Or is it because of the number of cloud-free images available, as shown in the text? One of the main advantages of using Sentinel-3 over Sentinel-2 for such applications is its more frequent revisit capability. With a 10-day interval, that benefit seems somewhat limited. This is just a question out of curiosity.
- 2. How are PAR and NIR calculated? The input appears to include only shortwave and long wave downward, ssrd and strd, respectively, from the CAMS dataset. Are the direct and diffuse PAR and NIR components derived from using a radiative transfer model, such as 6S, mentioned in the text? I think this was not quite clear in the text.
- 3. Is Equation 2.4 correct? I think  $\varepsilon$  (emissivity) should only apply to the outgoing longwave radiation term.
- 4. On what basis was the climate classification into Tropical, Dry, Continental, and Temperate made? While presenting the data this way makes sense, it would be helpful to clarify the method used. For instance, most papers use either an already existing climate classification, like the Koppen-Greger, or some aridity indices-based classification.
- 5. In Table 6, I am not sure why topographic correction was not applied to the total precipitation. Since topographic correction was applied to other variables, it seems this could have been done for precipitation as well, as precipitation is also quite influenced by topography. Especially when downscaling from the 0.4-degree original CAMS resolution to the 300 m, this step might also help to include some local orographic effects. Just a thought.
- 6. In Line 441, it is mentioned that the in-situ flux data are filtered based on a "realistic range." Could you clarify how this range is determined? Is it based on the solar constant or some other criterion? Or is it just visual inspection?
- 7. The stress factors utilized in the ETLook model for the soil and canopy involve topsoil and root-zone soil moisture, respectively (Equations 13 and 13b). While the topsoil moisture was approximated using a trapezoidal construction (LST-fc), how was the root-zone soil moisture determined to calculate the stress factor?
- 8. Figure 6: The caption could be more descriptive (applies to all the figures, in my view caption should be self-explanatory). Additionally, it is not clear how the graph was generated. Were all observed and modeled ETa values across all sites combined to compute the statistics, or were the statistics first calculated per station and then aggregated across stations? For instance, the standard deviation on the x-axis for the EC towers—does it include data from all EC towers?

- 9. In addition to point no. 8 for all Taylor plots, in my opinion, when comparing Taylor plots across different plant-function types (PFTs) and climate types, normalizing the standard deviation with respect to the observation could facilitate the comparison. For example, in Figure 8, the in-situ standard deviations differ across climate types. While this does not change the underlying information, normalizing would likely make it easier for the reader to interpret and compare the results.
- 10. In my view, there are too many figures presenting overlapping information. While I understand that the author wants to illustrate different aspects in the text from different figures, it affects the paper's brevity. This is just my opinion and can be ignored if you disagree. For example, Figure 7: In particular, the PFT graph does not seem to add significant value to the overall narrative and could be moved to the supplementary material, as much of the information is already conveyed in the tables and the Taylor plots. Figure 12: Since it only shows results for TSEB-PT, it could also be moved to the supplementary material.
- 11. In TSEB-PT, there is no direct control on evapotranspiration from soil moisture. In that respect, I would have expected ETLook to perform at least as well as TSEB-PT in arid and tropical regions. Could you provide some insight or a hint as to why this is not the case? Probably because stress is only coming from the land surface temperature in the ETLook (already LST being accommodated even in the TSEB-PT), rather than any direct soil moisture observation? Could it be a potential outlook incorporating direct remote sensing-based soil moisture?
- 12. In line 537, since the paper does not explore the details of any of the reasons mentioned, I would suggest rephrasing "it is obvious" to "the reasons might be," or alternatively, providing a proper citation.
- 13. Figure 14: In my view, this figure could also be moved to the supplementary material.
- 14. Figure 16: The first two rows correspond to May, and the middle row to July. Since all of the locations are in the Northern Hemisphere, is there a particular reason why the same month was not chosen to illustrate the differences in spatial coverage of the dataset?
- 15. In Figure 18, the green line representing green LAI is not clearly visible. Would it be nicer to plot LAI on the top panel and fg in the bottom panel? This would make the figure easier to understand.
- **16.** Additionally, just out of curiosity, could you clarify why the existing CLMS 300 m LAI product was not directly utilized in the model? The intermediate variable LAI produced in this work is also a product, though unpublished at the end, which seems to me like a potentially redundant effort.
- 17. Figure 23: Does the shaded area represent the spread of the data?
- 18. At last, I think it would be very helpful to include a flowchart connecting all the steps from input to output, highlighting the process from obtaining TOC reflectance and LST to calculating ET, if possible.

Thank you!

Prajwal Khanal

---

## Referee Comment (RC4)

**GENERAL COMMENTS**

Congrats on your work. I was involved in the development of the WaPOR v3 database and I am well aware of the challenges you faced. Impressive work. The paper is clearly based on years of experience in the topic, and presents an overall thorough research.

My recommendation is to consider the paper for publication although some revisions are required.

**MAJOR COMMENTS**

Gapfilling procedure: Section "2.9 Output gap-filling" describes the procedure used to fill gaps. It is chosen to fill gaps in the outputs and not in the inputs with the argumentation that the satellite observations (e.g. LST, LAI, albedo) are all acquired at the same moment, and will have the same gaps. The output gaps are filled using KcKs\*RET. Ks can change daily or even within a day, especially when soil moisture is depleted, or rainfall/irrigation happens. You are likely to overestimate actual ET as Ks is reduced under cloudy conditions. Ks is likely to be higher during cloud-free periods when plant have higher water demands. The problem with this method is that KS is not a fixed crop property, but depends on soil moisture and atmospheric demand. It would be more logical to gapfill soil moisture, as this is more constant over time, and you preserve the physical relationship. I do not think the corrections for rainfall are sufficient to overcome this weakness. Adding to this, it is not clear how the KcKS method was used to create decadal data. Did you calculate an Eta value for each day, or did you create decadal KcKs values?

**Ensemble:** The suggestion for an Ensemble model does miss a proper defense, where are the complementary strengths of the models? When one model is overestimating, and the other underestimating, the ensemble may appear closer to observation, but not necessarily because it captures the underlying processes better, it is simple averaging out the opposite biases.

**Imbalance in explaining design choices**: The paper does show an imbalance in methodological detail that affects understanding the key design choices. PROSPECT modelling is described exhaustively while other critical decisions receive less attention:

- Gapfilling approach (see above)
- "The CLMS ETa product specification states a spatial resolution of 300 m. However, the spatial resolution of the SLSTR LST product is 1 km." I understand that CLMS has a strong preference for Copernicus datasets, but I do miss an explanation on why Sentintel-3 LST at 1km has been chosen instead of higher spatial resolution datasets such as VIIRS, and what is the impact of this decision, except in section 4.4.

- CAMS data processing needs clarification, also on how historical data is derived.
- Also it is unclear whether the PROSPECT derived inputs are different from the existing CLMS biophysical products? And if not, why they are calculated differently?

**Validation:** Although the authors use a large number of EC stations for the validation, additional evidence is required for the statement "The CLMS ETa prototype also compared favourably with the global WaPOR ETa maps produced by FAO, which it is meant to replace and other higher-resolution ETa datasets (Section 4.1). The addition of ETa product in the CLMS portfolio should therefore significantly enlarge the CLMS user community" Except for figure 16, the paper does not show how datasets compare for larger areas (spatial patterns).

Model vs framework: To improve clarity, I would advise to distinguish more explicitly between the model (the algorithms) and the framework (the processing system including input selection, gapfilling, and temporal aggregation). The paper would benefit from making this distinction as it helps to understand the design choices. For example, in the sentences "Preparatory activities required to develop an operational CLMS ETa product recommended that two ET modelling frameworks should be further investigated. The first one is the Sen-ET framework (Guzinski et al., 2020, 2021) developed to model ETa with Copernicus data at various spatial scales and using the Two-Source Energy Balance Priestley-Taylor (TSEB-PT) ET model (Norman et al., 1995; Kustas and Norman, 1999; Anderson et al., 2024). The second is the WaPOR framework developed by FAO through the WaPOR project and using the ETLook ETa model (Bastiaanssen et al., 2012). Both models, although conceptually different, estimate evaporation and transpiration and use LST as one of core input forcings." The reference should be to the WAPOR ETLook model instead of the WaPOR framework, as the approach is different from the WaPOR modelling framework with regards on input selection, gapfilling and temporal aggregation.

**SPECIFIC COMMENTS**

**Introduction**

- 30 "Since actual evapotranspiration is a direct proxy of plant water use it can be utilized for consistent irrigation water use monitoring across natural and political boundaries": Since distinguishing between rainfall and irrigation water use remains a challenge, please clarify this limitation or remove the specific reference to irrigation monitoring.
- 50 "In order to satisfy this wide range of potential users' needs, and for consistency with other global CLMS products, the CLMS ETa product will have a spatial resolution of 300 m and a dekadal temporal resolution." It is not entirely clear to which users the 300m product caters?

55 - "Another operational and global product which utilizes MODIS and VIIRS data is produced by United States Geological Survey using SSEBop energy balance model (Senay et al., 2020) with dekadal temporal resolution and 1 km spatial resolution." Consider mentioning FEWS as the dataset is available there.

**Data and methods**

Table 2: Perhaps specify which inputs are used for which model? Personally I think a figure showing how these inputs are used to generate the model inputs (e.g. LAI, albedo) would give more insight. I assume "100m" in the weather data means "at 100m above the surface" and not to the spatial resolution – this may be made more clear, or removed.

110 - "same or similar values in both cloudy and sunny conditions (e.g. leaf area index does not change day to day depending on cloudiness). Therefore, gaps in this data are highly suitable for filling using spatio-temporal gap-filling" I understand this makes the data suitable for temporal gap-filing, but it does not automatically make it suitable for spatial gapfilling?

121 - Please also introduce View Zenith Angle (VZA) in the text (it is currently only in the captions).

270 - "More details and the list of evaluated indices are available in the WaPOR wiki (https://bitbucket.org/cioapps/wapor-et-look/wiki/Intermediate\_Data\_Components/LST, last accessed: 22/07/2025)": This repository recently moved to <a href="https://github.com/unfao/wapor-et-look">https://github.com/unfao/wapor-et-look</a>, consider updating.

275 - "Finally, since we do not expect strong influence of aspect and slope on LST those two variables were removed from the WaPOR list and the resulting combination of 9 variables (called "DMS - WaPOR selected" in Section 4.3.2) is used in the ETa processing chain to sharpen the 1 km Sentinel-3 LST to the required 300 m spatial resolution." On what did you base this expectation?

255 - "ETLook model (Bastiaanssen et al., 2012) is used in the WaPOR framework and is described in detail in Section 5 of "WaPOR Data Manual, Evapotranspiration v2.2" (FRAME Consortium, 2020). ": Please note that the WaPOR data manual refers to the ETLook version 2, and mostly describes how the inputs are derived, while the methodology (the model) used in v2 is described in the methodology document (https://openknowledge.fao.org/server/api/core/bitstreams/d3db4794-fb5b-444c-9b3a-c5fb154c5f9f/content). For version 3 the data manual and methodology documentation were combined, with all updates and changes described in the Github page.

**3 Prototype product validation**

Figure 5: The lack of Eddy Covariance stations outside Northern America, Europe and Australia is an issue, with Africa, Asia and South America only represented by a few stations. This is mentioned in the text, and counterargued with that all major climate zones and plant functional types are represented by at least one EC station. In figure 5 you do show the number of dates available for each PFT and climate zone, but could you add the number of stations as well? I think that would improve our insight in which areas are still underrepresented in EC datasets.

400 - Temporal aggregation smooths errors. Why did you choose to validate the dekadal computations?

440 - "Missing data during the day were computed by linear interpolation if the number of valid timeslots during daytime was at least 50% of the total number of timeslots in that period. Otherwise, the day was discarded." This means you are interpolating both inputs and outputs, so I would mention that this interpolation may smooth variability and can influence error metrics.

470 - "Conversely, the models performed less good in the Tropical and Dry regions." I would not attribute the poorer performance to the models themselves as the causes are likely input related. In tropical regions, frequent cloud cover will result in missing remote sensing data inputs, while in dry regions it may be a result of missing short-term ET peaks after rainfall.

455-485 – The text describing the figures does not describe the WAPOR outputs while they are in the figures. The comparison with WaPOR is available in the discussion section. But since WaPOR is also based on the ETLook model, but uses other inputs, this would be an excellent opportunity to assess the impact of different inputs (sensors, datasets) and different input timesteps (daily vs decadal).

Figure 8: I would add the climate region to the individual plots (instead of A, B, C and D).

Figure 8/9: I would also add the number of sites used for each figure. Or the number of data points. Now they seem to have the same importance while some are based on more data points.

Figure 10: The reason for selecting the specific validation sites is not fully explained. If these sites are selected to illustrate the difference between the two models, this should be made explicit. The differences between the models (in particular T) requires further discussion. Moreover, I have some concerns regarding the choice for EBF (evergreen broadleaved forest) and DBF (deciduous broadleaved forest) sites as evapotranspiration modelling of forests is rather complicated for any ET model. For readability I would repeat the abbreviations like EBF more often, in particular in figures like figure 10.

**Discussion**

525 - The discussion on the differences between WaPOR Eta and CLMS ETLook Eta is very thorough.

- 538 "While both CLMS ETa and WaPOR ETa rely on DMS to improve the spatial resolution of LST, the original LST in CLMS is acquired by SLSTR sensor on board Sentinel-3 satellite with 1 km spatial resolution, while the original LST in WaPOR (version 3) is acquired by the VIIRS sensor on board of Suomi-NPP satellite with 375 m spatial resolution." => WaPOR L1 does not use DMS as VIIRS LST has a spatial resolution of 375m, and DMS would only introduce errors. DMS is only used for WaPOR L2 and L3.
- 543 Regarding point 2 (WaPOR being a reanalysis product) I have one remark:
  WaPOR is produced both NRT and after 6 dekads reprocessed. See also
  https://github.com/un-fao/wapor-et-look/wiki/Understanding%20the%20WaPOR%20Pipeline#wapor-database
- 545- Regarding point 3: the tenacity factor of WaPOR ETLook is 2: See <a href="https://github.com/un-fao/wapor-et-look/wiki/Release%20Notes">https://github.com/un-fao/wapor-et-look/wiki/Relative%20Root%20Zone%20Soil%20Moisture</a>.

581 - "The examples in Figure 16 show as well that the number of missing data in the output maps of ETLook and TSEB-PT is larger than in the WaPOR product. The reason for those gaps are the differences between NRT and reanalysis gap-filling (see Section 4.1) but also the different model inputs and treatments of inland water and snow" This explanation should be expanded to include the differences in gapfilling the inputs or outputs.

730 - "This situation should be resolved by the end of the decade when Land Surface Temperature Monitoring (LSTM) mission, with a primary objective of frequent monitoring of field-scale ETa, will join the Copernicus constellation (Koetz et al., 2018). "Is this approach realistic for an operational global product?

Thanks!

Annemarie Klaasse

---

## Author Comment (AC1)

Dear Prajwal,

Thank you for your positive and constructive feedback. We provide our response below each comment in green font.

 I thoroughly enjoyed reading this paper; it is very informative and descriptive. I recommend that the editor consider it for publication, subject to addressing some clarification questions and minor corrections. In my view, the paper is somewhat lengthy because several figures present overlapping information, which could be streamlined to make the main text more concise.

1.        Other than the requirements mentioned in line 50, which align with the applications of the WaPOR global ETa dataset, I was wondering why a 10-day interval was selected. Since the revisit time of Sentinel-3 instruments is around 2 days at the equator, a finer temporal resolution seems feasible. While the revisit frequency decreases at higher latitudes, the chances of acquiring clear observations, especially during morning overpasses, are already quite low. Therefore, I'm curious why a shorter revisit interval was not targeted. Or is it because of the number of cloud-free images available, as shown in the text? One of the main advantages of using Sentinel-3 over Sentinel-2 for such applications is its more frequent revisit capability. With a 10-day interval, that benefit seems somewhat limited. This is just a question out of curiosity.

There is a bit of misunderstanding here. The temporal resolution is 10-days but that does not mean that the used Sentinel-3 images are acquired every 10-days. In fact, we use every Sentinel-3 image within the 10-day compositing period to derive clear-sky ET estimates at a daily timescale, then gap-fill those to obtain all-sky daily ET estimates and finally composite (average) those to create a dekadal ET map (lines 375-378). The requirement for dekadal compositing is common among many CLMS global products and is partly based on user-needs, partly on cost-efficiency of storying and distributing global datasets, partly on improved robustness of products when temporal averaging is applied.  However, the CLMS ET product will also contain TSEB-PT derived instantaneous heat fluxes provided to users at a daily time-step.

To clarify the above points, we will edit the manuscript as follows.

Rename section 2.9 to "Output gap-filling and temporal compositing"

Add following sentence to the last paragraph of section 2.9 (line 398):

"The all-sky gap-filled daily estimates of evapotranspiration, evaporation and transpiration are then aggregated to dekadal timesteps by taking the mean of all valid values within the aggregation period."

Lines 754-755 in the conclusion will be modified to:

"The product is designed to have a 300 m spatial resolution, dekadal temporal resolution for the water fluxes (evapotranspiration and its components of evaporation and transpiration) and daily

temporal resolution for the heat fluxes (latent and sensible heat fluxes), global extent and to be produced in near-real-time with a 2-day delay (Table 1)."

2.      How are PAR and NIR calculated? The input appears to include only shortwave and long wave downward, ssrd and strd, respectively, from the CAMS dataset. Are the direct and diffuse PAR and NIR components derived from using a radiative transfer model, such as 6S, mentioned in the text? I think this was not quite clear in the text.

PAR and NIR instantaneous beam and diffuse shortwave irradiance are estimated thanks to the REST2 model (Gueymard 2008) using interpolated CAMS hourly AOD and TCWV as inputs, and then corrected for topography after Aguilar et al. 2010. This will be clarified in the revised manuscript.

References:

1. Gueymard, C.A. (2008). REST2: High-performance solar radiation model for cloudless-sky irradiance, illuminance, and photosynthetically active radiation – Validation with a benchmark dataset. Solar Energy 82, 272–285. https://doi.org/10.1016/j.solener.2007.04.008.

2. Aguilar, C., Herrero, J., and Polo, M.J. (2010). Topographic effects on solar radiation distribution in mountainous watersheds and their influence on reference evapotranspiration estimates at watershed scale. Hydrol. Earth Syst. Sci. 14, 2479–2494. https://doi.org/10.5194/hess-14-2479-2010.

3.      Is Equation 2.4 correct? I think ε (emissivity) should only apply to the outgoing longwave radiation term.

Reflectivity (thermal albedo) is the inverse of emissivity: refl = 1 – ε. Longwave net radiation consists of incoming minus reflected radiation (similarly to net shortwave radiation) plus the emitted radiation. Therefore, the equation is correct.

4.      On what basis was the climate classification into Tropical, Dry, Continental, and Temperate made? While presenting the data this way makes sense, it would be helpful to clarify the method used. For instance, most papers use either an already existing climate classification, like the Koppen-Greger, or some aridity indices-based classification.

The Tropical, Dry, Continental and Temperate classes correspond to the main groups of the Köppen climate classification system.  In the nomenclature of the Köppen system these groups get the labels A, B, C and D, respectively.  We will mention more explicitly in the revised version of the document that those categories were obtained from the Köppen climate classification system.

5.      In Table 6, I am not sure why topographic correction was not applied to the total precipitation. Since topographic correction was applied to other variables, it seems this could have been done for precipitation as well, as precipitation is also quite influenced by topography. Especially when downscaling from the 0.4-degree original CAMS resolution to the 300 m, this step might also help to include some local orographic effects. Just a thought.

Topography certainly has an influence on rainfall, even at local scales (i.e. 300 m). However, correcting for this is complex and as far as we are aware not something which can be done operationally at global extent and daily timescale within reasonable computing costs. At the same time, rainfall is not a core input used by either of the two models but is just used to improve the gap-filling procedure.

6.      In Line 441, it is mentioned that the in-situ flux data are filtered based on a "realistic range." Could you clarify how this range is determined? Is it based on the solar constant or some other criterion? Or is it just visual inspection?

The realistic range we refer to in the manuscript is intended to consider the situations in which eddy covariance data providers use fill values to indicate that data is missing. For instance, the use of values like "–999.9" for float data types of "255" for the 8-bit integer data type might be found in datasets to indicate missing data. In the data selection step we removed records with fill data in addition to the use of quality flags. In the new version of the manuscript we will rephrase and replace the expression 'realistic range' to avoid misunderstandings.

7.      The stress factors utilized in the ETLook model for the soil and canopy involve topsoil and root-zone soil moisture, respectively (Equations 13 and 13b). While the topsoil moisture was approximated using a trapezoidal construction (LST-fc), how was the root-zone soil moisture determined to calculate the stress factor?

The top-soil and root-zone soil moisture are assumed to have the same value and are estimated using the same trapezoid. We will clarify this by modifying lines 371-372 to the following: "Then the soil moisture of a pixel (representing both top-soil and root-zone) is estimated using the relative location of LST and fC of that pixel within that trapezoid."

8.      Figure 6: The caption could be more descriptive (applies to all the figures, in my view caption should be self-explanatory). Additionally, it is not clear how the graph was generated. Were all observed and modeled ETa values across all sites combined to compute the statistics, or were the statistics first calculated per station and then aggregated across stations? For instance, the standard deviation on the x-axis for the EC towers—does it include data from all EC towers?

Thank you for the suggestion, we will revise all the captions to be more self-explanatory. The Taylor diagram in Figure 6 was generated from the full dataset of observed and modelled ETa pairs across all the validation sites. Therefore, the standard deviation on the x-axis, as well as the other metrics in the Taylor diagram, include data from all the eddy covariance stations considered in the analysis. We will also clarify this in the revised manuscript.

9.      In addition to point no. 8 for all Taylor plots, in my opinion, when comparing Taylor plots across different plant-function types (PFTs) and climate types, normalizing the standard deviation with respect to the observation could facilitate the comparison. For example, in Figure 8, the in-situ standard deviations differ across climate types. While this does not change the underlying information, normalizing would likely make it easier for the reader to interpret and compare the results.

Thank you for the suggestion. Normalizing standard deviation might make comparison between PFTs and climates easier but it also removes some information from the plot (the actual value of standard deviation). We will consider this suggestion when revising the manuscript.

10.      In my view, there are too many figures presenting overlapping information. While I understand that the author wants to illustrate different aspects in the text from different figures, it affects the paper's brevity. This is just my opinion and can be ignored if you disagree. For example, Figure 7: In particular, the PFT graph does not seem to add significant value to the overall narrative and could be moved to the supplementary material, as much of the information is already conveyed in the tables and the Taylor plots. Figure 12: Since it only shows results for TSEB-PT, it could also be moved to the supplementary material.

Thank you for the suggestions and we agree that the manuscript is a bit on the longer side. When revising, we will carefully consider which parts can be moved to the supplementary material including the two mentioned figures.

11.      In TSEB-PT, there is no direct control on evapotranspiration from soil moisture. In that respect, I would have expected ETLook to perform at least as well as TSEB-PT in arid and tropical regions. Could you provide some insight or a hint as to why this is not the case? Probably because stress is only coming from the land surface temperature in the ETLook (already LST being accommodated even in the TSEB-PT), rather than any direct soil moisture observation? Could it be a potential outlook incorporating direct remote sensing-based soil moisture?

Both ETLook and TSEB-PT make use of LST to account for soil moisture. ETLook does it more explicitly, due to the derivation of soil moisture using the trapezoid approach, while in TSEB-PT it is more implicit through calculation of the temperatures and energy fluxes of the vegetation and the soil.  In the revised manuscript we will add a new section to the discussion where we analyze the assumptions and limitations of the two models through their performance in different

climates and plant functional types. Regarding incorporation of direct (radar-based) soil moisture into evapotranspiration models, this is an active research topic, including by some authors of this manuscript, but is out of scope to include here.

12.     In line 537, since the paper does not explore the details of any of the reasons mentioned, I would suggest rephrasing "it is obvious" to "the reasons might be," or alternatively, providing a proper citation.
We agree with the suggestions and will change the manuscript accordingly.

13.     Figure 14: In my view, this figure could also be moved to the supplementary material.
We will consider this suggestion when revising the manuscript as mentioned in reply to comment 10.

14.     Figure 16: The first two rows correspond to May, and the middle row to July. Since all of the locations are in the Northern Hemisphere, is there a particular reason why the same month was not chosen to illustrate the differences in spatial coverage of the dataset?
There was no particular reason apart from the selected spatial and temporal subsets nicely illustrating the points described on lines 581-588. The figure would look very similar if panel B was from May and since this figure is only for qualitative descriptive purposes, the actual date does not matter too much. Therefore we would prefer to keep it as is.

15.     In Figure 18, the green line representing green LAI is not clearly visible. Would it be nicer to plot LAI on the top panel and fg in the bottom panel? This would make the figure easier to understand.
In the top panel of each sub-figure, we plot four parameters: green LAI derived in biophysical processor (green line), green LAI derived with use of CLMS LAI/fAPAR (green circles), LAI derived with biophysical processor (black line) and CLMS LAI (black circles). Many of those parameters are close to each other, especially during some parts of the year, which is why not all datasets are always visible on top. The bottom panel shows fg from biophysical processor (green line) and derived with the use of CLMS LAI/fAPAR (green circles). In the revised manuscript, we will try to make the caption and the figure clearer.

16.     Additionally, just out of curiosity, could you clarify why the existing CLMS 300 m LAI product was not directly utilized in the model? The intermediate variable LAI produced in this work is also a product, though unpublished at the end, which seems to me like a potentially redundant effort.
The biophysical processor derives LAI, fg as well as pigments used for albedo estimation. The two first parameters could be obtained from CLMS LAI/fAPAR product but the pigments could not (lines 599-601). While the green LAI derived with biophysical processor agrees quite well with the

CLMS LAI (Figure 17, middle panel), the fg derived with both methods are quite different (Figure 17, left panel) with fg from biophysical processor better capturing the expected behavior (Figure 18 and lines 627 – 638). This does not imply that the CLMS LAI is less accurate, just that we needed to use a simplified method to derive fg with the CLMS LAI and FAPAR products. Therefore, since we needed to derive the pigments and fg with biophysical processor we used it also to derive LAI.

17.      Figure 23: Does the shaded area represent the spread of the data?
The shaded area represents the uncertainty range (energy balance closure error) of the data. We will add this information to the caption in the revised manuscript.

18.      At last, I think it would be very helpful to include a flowchart connecting all the steps from input to output, highlighting the process from obtaining TOC reflectance and LST to calculating ET, if possible.
This is a good suggestion and we will add such figure to the revised manuscript.

Thank you!

Prajwal Khanal

---

## Author Comment (AC2)

Dear Reviewer,

We would like to thank the reviewer for the constructive feedback.  We provide our response below each comment in green font.

The article describes the procedure to develop the new ET product of the Copernicus Land Monitoring Service (CLMS), which is currently missing. The description of data inputs and the modeling chain are well described, but several points need to be addressed before the paper can be accepted.

Developing a global, operational product is constrained by certain programmatic, financial and technical limitations. The CLMS products are designed for long-term continuity and are paid for by the European Commission therefore issues like compute efficiency, data storage and dissemination costs or long-term availability of sources of core input data need to be considered. Taking those constraints into account, and in order to satisfy wide range of potential users needs and for consistency with other global CLMS products, the European Commission set the requirements for the CLMS ETa product, as well as its partitioning between soil evaporation and canopy transpiration, of a spatial resolution of 300 m and a dekadal temporal resolution and data availability in near-real-time (NRT). Other product requirements are shown in Table 1. Those requirements in turn impact the design choices presented in this study for the first version of the CLMS ETa product.

A similar clarification will be added to the Introduction (inserted around line 47) of the updated manuscript. However, despite those constraints we believe that the study adds to the scientific understanding of challenges and solutions applicable to global modelling of evapotranspiration.

We provide more details at the specific points below.

Comments

1.  In the abstract the main innovation could be more explicitly included
    Thank you for the suggestion, we will modify the abstract to better highlight the main innovation which is the development of global, operational and near-real-time ET product.

2.  Temporal and spatial resolution: 300 m is not the original resolution of either the vegetation and the land surface temperature products. 10 days of temporal resolution is probably not enough to capture the variations of ET, especially in agricultural areas or for water management applications, as the authors state. If daily meteorological forcings are available, with the gap filling technique described in the paper, an almost daily product cloud be produced. A detailed explanation of these aspects is needed to identify the weakness of the

product.

As described in the introduction, the 300 m spatial resolution and 10-day temporal resolution are based on cost-benefit analysis conducted by European Comission when deriving requirements of the product and they are the constraints within which we conducted this study. However, a couple of points can be highlighted:
- 300 m is indeed the spatial resolution of the Sentinel-3 OLCI sensor and therefore the vegetation products.
- 10 day temporal resolution of the FAO's WaPOR product has been successfully used in multiple studies to assess the variations of ET in agricultural areas and water management.
- The manuscript describes the design of a global product which aims to have long-term continuity. Increasing the spatial or temporal resolutions would have an impact on the data storage and distribution costs.
- Despite of the above, and in accordance with the European Commission requirements, daily heat flux dataset (instantaneous sensible and latent heat fluxes at the satellite overpass time) derived with TSEB-PT model will be distributed as part of the CLMS ET product. We will add this clarification to Section 1.

3. The choice of the two models (TSEB-PT and ETLook) should be better justified. Both models are based on the energy balance, limiting the strength of the analyses. The research community have developed and compared many models based on different approaches (energy balance, water balance, machine learning …) agreeing that it doesn't exist a model that clearly outperforms all the others (like the OpenET approach). A clear justification from a methodological, hypotheses and inputs data analyses is missing, when the selection of only these two models has been done.

For the justification of the selection of the two models, please see the reply to comment number 8. However, we believe that even with the selection of just the two models there is a clear scientific interest in exploring the selection and pre-processing of the various input data as the same inputs can be used in the other models. This is especially relevant since the pre-processing methods are publicly available as open-source Python packages (see Code Availability section). Similarly, looking at the individual and ensemble outputs of just those two models can lead to improved scientific understanding of the models themselves, but also to potential improvements in overall modelling of land surface water and energy fluxes.

4. It's interesting to see that the ensemble approach outperforms each of the two models. But I think the authors should justify why not more ET models based on different assumptions could not create a more accurate ensemble ET.

We agree that, as a general rule, the more models included in the ensemble the more accurate the final output will be as well as a better understanding of the model uncertainties. For the justification of the selection of the two models, please see the reply to comment number 8. Introduction of more models increases the costs of both computing and data storage and dissemination (both the individual model outputs and the ensemble will be distributed to the users). Therefore, for this first version of CLMS ET product only two models are used in the ensemble. If users and studies show clear benefit of additional models, then the European Commission might be more inclined to increase the ensemble model size in potential future evolution of this product. This is also the hope of the authors.

5. Lines52-55: actually there are many other models available in literature. A more comprehensive analysis should be done, especially many models have higher temporal resolution.
We should have been clearer here. The short review does not focus on available ET models, of which there are many, but on ET datasets which are currently produced operationally (i.e. at regular intervals, using well established models and validated outputs, etc.) with global coverage and in near-real-time. We will update line 51-52 to the following:
"..., global ETa datasets that are currently produced operationally and updated in NRT with closest..."

6. Line 59: ETmonitor is operational
We agree that ETMonitor is an operational modelling approach. However, the dataset finishes in 2021 so it is not produced in near-real-time. We will modify the end of this line to read "...or are not produced in NRT (e.g. ETMonitor - Zheng et al. (2022))."

7. Line 60: who decided and why only two models should be considered?
The decision was made by the European Commission. Please see the reply to comment number 8.

8. I suggest adding a section describing the proper selection of only two models based on the energy balance approach
We will modify the paragraph on lines 60-65 to the following:
Preparatory activities conducted by European Commission to develop an operational CLMS ETa product recommended that two ET modelling frameworks should be further investigated. The first one is the Sen-ET framework (Guzinski et al., 2020, 2021) developed to model ETa with Copernicus data at various spatial scales and using the Two-Source Energy Balance Priestley-Taylor (TSEB-PT) ET model (Norman et al., 1995; Kustas and Norman, 1999; Anderson et al., 2024). The second is the WaPOR framework (FRAME Consortium & FAO,

2024) developed by FAO through the WaPOR project and using the ETLook ETa model (Bastiaanssen et al., 2012). Both models, although conceptually different, estimate evaporation and transpiration and use LST as one of core input forcings. This recommendation was mainly based on the availability of mature open-source implementations of the two ET models, on previous studies demonstrating the applicability of both models with Copernicus data sources (i.e. Sentinel-3 imagery and meteorological data from European Center for Medium Range Weather Forecasts) (Guzinski et al., 2025) and on FAO's familiarity with both approaches. This does not imply that those two modelling frameworks clearly outperform all other approaches and indeed it has been shown that the performance of an individual model depends on the landcover and climatic conditions (Reitz et al, 2025). However, due to constraints on developing a publicly funded global and operational dataset (outlined previously) there was a need to limit the design of the first version of the CLMS ETa product to those two frameworks.

References

1. Guzinski, R., Nieto, H., Sandholt, I., and Karamitilios, G. (2020). Modelling High-Resolution Actual Evapotranspiration through Sentinel-2 and Sentinel-3 Data Fusion. Remote Sensing 12, 1433. https://doi.org/10.3390/rs12091433.

2. Guzinski, R., Nieto, H., Sánchez, J.M., López-Urrea, R., Boujnah, D.M., and Boulet, G. (2021). Utility of Copernicus-Based Inputs for Actual Evapotranspiration Modeling in Support of Sustainable Water Use in Agriculture. IEEE Journal of Selected Topics in Applied Earth Observations and Remote Sensing 14, 11466–11484. https://doi.org/10.1109/JSTARS.2021.3122573.

3. Norman, J.M., Kustas, W.P., and Humes, K.S. (1995). Source approach for estimating soil and vegetation energy fluxes in observations of directional radiometric surface temperature. Agricultural and Forest Meteorology 77, 263–293. https://doi.org/10.1016/0168-1923(95)02265-Y.

4. Kustas, W.P., and Norman, J.M. (1999). Evaluation of soil and vegetation heat flux predictions using a simple two-source model with radiometric temperatures for partial canopy cover. Agricultural and Forest Meteorology 94, 13–29. https://doi.org/10.1016/S0168-1923(99)00005-2.

5. FRAME Consortium & FAO: WaPOR methodology, https://github.com/un-fao/wapor-et-look/wiki, 2024.

6. Bastiaanssen, W.G.M., Cheema, M.J.M., Immerzeel, W.W., Miltenburg, I.J., and Pelgrum, H. (2012). Surface energy balance and actual evapotranspiration of the transboundary Indus Basin estimated from satellite measurements and the ETLook model. Water Resources Research 48. https://doi.org/10.1029/2011WR010482.

7. Reitz, M., Volk, J.M., Ott, T., Anderson, M., Senay, G.B., Melton, F., Kilic, A., Allen, R., Fisher, J.B., Ruhoff, A., et al. (2025). Performance Mapping and Weighting for the Evapotranspiration Models of the OpenET Ensemble. Water Resources Research 61, e2024WR038899. https://doi.org/10.1029/2024WR038899.

9. Line 188: please explain "that Ln is usually computed internally by each ET model". How the models compute especially the incoming longwave radiation is important due to the high uncertainty. In addition, how emissivity is computed/measured?

   In the TSEB-PT model, Ln is computed according to equation 2a from Kustas & Norman (1999) using CAMS surface_thermal_radiation_downwards field together with internally computed leaf and soil temperatures, constant emissivity values for the vegetation and soil (0.98 and 0.95 respectively) and effective leaf area index. WaPOR-ETLook estimates it following the FAO-56 approach and daily values of air temperature, vapour pressure and transmissivity (FRAME Consortium & FAO, 2024).

   References:

   1. Kustas, W.P., and Norman, J.M. (1999). Evaluation of soil and vegetation heat flux predictions using a simple two-source model with radiometric temperatures for partial canopy cover. Agricultural and Forest Meteorology 94, 13–29. https://doi.org/10.1016/S0168-1923(99)00005-2.

   2. FRAME Consortium & FAO: WaPOR methodology, https://github.com/un-fao/wapor-et-look/wiki, 2024.

10. 12 how the aerodynamic resistances are computed?

    We will edit lines 361-362 to read:

    "..., $r_{a,S}$ and $r_{a,C}$ are aerodynamic resistances for heat turbulent transport for soil and canopy respectively calculated following (Allen, 1998) and adjusted for buoyancy using Monin-Obukhov similarity theory in unstable conditions, and $r_S$ and ...."

    Reference

    Allen, R. G., ed.: Crop evapotranspiration: guidelines for computing crop water requirements, no. 56 in FAO irrigation and drainage paper, Food and Agriculture Organization of the United Nations, Rome, repr edn., ISBN 978-92-5-104219-9, 1998.

11. Equation 13.a add the reference of the soil resistance equation. A definition of the empirical parameters a,b,c, is missing. Are they defined with soil texture? Setop is a stress factor or soil moisture?

    We will clarify the mentioned points by edition lines 363-368 to:

    The resistance of soil to vapour transfer is calculated as

$$r_S = b(S_e^{top})^c$$

where b and c are soil resistance parameters set to constant values of 800 and -2.1 respectively and $S_e^{top}$ is relative top-soil soil moisture (Camillo & Gurney, 1996].

The resistance of canopy to vapour transfer is affected by air temperature stress (St), vapour pressure stress (Sv), radiation stress (Sr) and root-zone soil moisture stress (Sm):

$$r_C = (\frac{r_{s,min}}{LAI_{eff}})(\frac{1}{S_t S_v S_r S_m})$$

where rs,min (s m$^{-1}$) is the minimum stomatal resistance, LAIeff is effective leaf area index (Jarvis, 1976; Stewart, 1988).
References:
1. Camillo, P.J., and Gurney, R.J. (1986). A RESISTANCE PARAMETER FOR BARE-SOIL EVAPORATION MODELS: Soil Science 141, 95–105. https://doi.org/10.1097/00010694-198602000-00001.
2. Jarvis, P.G. (1976). The interpretation of the variations in leaf water potential and stomatal conductance found in canopies in the field. Phil. Trans. R. Soc. Lond. B 273, 593–610. https://doi.org/10.1098/rstb.1976.0035.
3. Stewart, J.B. (1988). Modelling surface conductance of pine forest. Agricultural and Forest Meteorology 43, 19–35. https://doi.org/10.1016/0168-1923(88)90003-2.

12. From figure10 and 11, it seems that the biggest errors are present in the forest areas, especially big differences between the two models are found in transpiration during summer time. Probably the parametrization of aerodynamic resistance is not proper for high vegetation. In addition, it always seems that ET from ETLook is always higher than TSEB-PT, leading to an ensemble which is an average of the two.
It is indeed the case that largest differences are between the models is in the transpiration, which we highlight on lines 486-490. We will add a new sub-section to the discussion in which we will look at those differences in the context of the assumptions and limitations of each of the modelling approaches. It should also be noted that in the revised manuscript we will use updated parameterization of the two models which is expected to reduce the differences between them in the forests.

13. In general there is a long description of the errors between models/climatic conditions/vegetation, but a more detailed explanation of the models hypotheses on the obtained results is needed to understand the model accuracies.
This is a valid suggestion and we will add a section in the discussion where we put the differences between the models in the context of their different assumptions and modelling

approaches.

14. Line 509: The high errors in the cities should be better discussed, as both models don't have a specific module for the urban energy balance.
It is correct that neither of the models can deal with ET estimation in urban areas, and we highlight this point on lines 498-501. We will mention it again here.

15. Figure13: the text in the figure is not readable
Thank you for highlighting this, it will be corrected in the updated manuscript.

16. The use of weather forcings in respect to reanalyse data should be clearly defined, and a discussion on the possible loose of accuracy in the model ET estimates discussed.
In section 4.3.3 and in particular in Figure 22 we show that the forecast CAMS weather forcing is suitable for a global ET product to be delivered with 1-2 days timeliness. Reanalysis (ERA-5) data does not meet this timeliness requirement and therefore comparison between ERA-5 and CAMS was not conducted as part of this study.

---

## Author Comment (AC3)

Dear Annemarie,

Thank you for your positive review and valuable suggestions to improve the manuscript. . We provide our response below each comment in green font.

GENERAL COMMENTS

Congrats on your work. I was involved in the development of the WaPOR v3 database and I am well aware of the challenges you faced. Impressive work. The paper is clearly based on years of experience in the topic, and presents an overall thorough research.

My recommendation is to consider the paper for publication although some revisions are required.

MAJOR COMMENTS

**Gapfilling procedure:** Section "2.9 Output gap-filling" describes the procedure used to fill gaps. It is chosen to fill gaps in the outputs and not in the inputs with the argumentation that the satellite observations (e.g. LST, LAI, albedo) are all acquired at the same moment, and will have the same gaps. The output gaps are filled using KcKs*RET. Ks can change daily or even within a day, especially when soil moisture is depleted, or rainfall/irrigation happens. You are likely to overestimate actual ET as Ks is reduced under cloudy conditions. Ks is likely to be higher during cloud-free periods when plant have higher water demands. The problem with this method is that KS is not a fixed crop property, but depends on soil moisture and atmospheric demand. It would be more logical to gapfill soil moisture, as this is more constant over time, and you preserve the physical relationship. I do not think the corrections for rainfall are sufficient to overcome this weakness. Adding to this, it is not clear how the KcKS method was used to create decadal data. Did you calculate an Eta value for each day, or did you create decadal KcKs values?

We agree that no gap-filling method is perfect and using $K_{c,s}$*RET also has its limitations. Nevertheless, it is a widely used method to gap-fill actual evapotranspiration estimates (including in the Sen-ET framework) and is of comparable skill to other widely used gap-filling methods (see e.g. Delogu et al., 2021). A large part of the changes in atmospheric demand are already captured in the RET calculation and we try to minimize the effect of soil moisture by using $K_{c,s}$ derived as closely as possible to the target date. We further try to account for changes in SM by e.g. taking rainfall into account. Gap-filling soil moisture might work for ETLook (but not for TSEB which does not use it as input) but itself contains limitations. For example, the Whittaker smoother used in the WaPOR framework assumes smooth variability in SM which might not be necessary true during rainfall, irrigation or drought events.

Regarding the creation of dekadal values, we will add following sentence to the last paragraph of

section 2.9 (line 398):

"The all-sky gap-filled daily estimates of evapotranspiration, evaporation and transpiration are then aggregated to dekadal timesteps by taking the mean of all valid values within the aggregation period."

References:

1. Delogu, E., Olioso, A., Alliès, A., Demarty, J., and Boulet, G. (2021). Evaluation of Multiple Methods for the Production of Continuous Evapotranspiration Estimates from TIR Remote Sensing. Remote Sensing 13, 1086. https://doi.org/10.3390/rs13061086.

**Ensemble:** The suggestion for an Ensemble model does miss a proper defense, where are the complementary strengths of the models? When one model is overestimating, and the other underestimating, the ensemble may appear closer to observation, but not necessarily because it captures the underlying processes better, it is simple averaging out the opposite biases.

Ensemble models (or ensembles of model outputs) often perform better than individual models due to cancellation of random errors, and sometimes bias, produced by the individual models (Volk et al. 2024). Indeed this technique is widely applied in weather forecasting, climate change predictions and hydrological modelling. This does not necessarily mean that they better capture the underlying processes. In the revised manuscript we will add a new section to the discussion in which we analyze the assumptions and limitations of the two models through their performance in different climates and plant functional types and which might help to address this issue. We will also add a paragraph in section 4.4 (Potential improvements) on a more robust method for producing ensemble which takes the model performance into account based on the recent experience of the openET consortium (Reitz et al, 2025).

References:

1. Volk, J.M., Huntington, J.L., Melton, F.S., Allen, R., Anderson, M., Fisher, J.B., Kilic, A., Ruhoff, A., Senay, G.B., Minor, B., et al. (2024). Assessing the accuracy of OpenET satellite-based evapotranspiration data to support water resource and land management applications. Nat Water 2, 193–205. https://doi.org/10.1038/s44221-023-00181-7.

2. Reitz, M., Volk, J.M., Ott, T., Anderson, M., Senay, G.B., Melton, F., Kilic, A., Allen, R., Fisher, J.B., Ruhoff, A., et al. (2025). Performance Mapping and Weighting for the Evapotranspiration Models of the OpenET Ensemble. Water Resources Research 61, e2024WR038899. https://doi.org/10.1029/2024WR038899.

**Imbalance in explaining design choices**: The paper does show an imbalance in methodological detail that affects understanding the key design choices. PROSPECT modelling is described exhaustively while other critical decisions receive less attention:

Gapfilling approach (see above)

Due to the already lengthy nature of this manuscript, we tried to focus on areas which were not previously described in detail in other papers. For topics which were described in previous publications, such as the two ET models, gap-filling or thermal sharpening, we provide a shorter description and references to those other studies.

"The CLMS ETa product specification states a spatial resolution of 300 m. However, the spatial resolution of the SLSTR LST product is 1 km." I understand that CLMS has a strong preference for Copernicus datasets, but I do miss an explanation on why Sentintel-3 LST at 1km has been chosen instead of higher spatial resolution datasets such as VIIRS, and what is the impact of this decision, except in section 4.4.

The use of Sentinel-3 LST was one of the constraints imposed on the CLMS ET product by the European Commission. The major reasons for this are mentioned on lines 81-82 ("This is to ensure the free and open license conditions, long-term future continuity and consistency across the CLMS portfolio"). In particular, the long-term continuity of VIIRs is not guaranteed. Since this was a constraint within which the study was conducted, an analysis of potential impacts on this decision was not performed. However, we briefly mention a potential impact when discussing the differences between WaPOR and CLMS ETLook outputs in section 4.1. In the revised manuscript we will add a paragraph to the introduction explaining those constraints.

CAMS data processing – needs clarification, also on how historical data is derived.

The CAMS data is used for both NRT and historical production, in both cases using the shortest possible forecast duration of 12 hours. We will add this clarification to section 2.6. The reason for this is that the first version of CLMS ET product will be produced only in the near-real-time mode, e.g. there will be no reprocessing 2-3 months after the production date to perform improved gap-filling or use reanalysis meteorological data (like is done e.g. in WaPOR). Therefore, using data source other than CAMS for data produced before November 2025 (when operational production started) and CAMS for data produced after that date could lead to discontinuity of the timeseries. In Figure 22 we demonstrate that the CAMS data is highly suitable for use in this application. Nevertheless we also mention the production of reanalysis dataset, which could use ERA 5 instead of CAMS, as one the potential improvements in Section 4.4 (lines 741-746).

Also it is unclear whether the PROSPECT derived inputs are different from the existing CLMS biophysical products? And if not, why they are calculated differently?

The biophysical processor (PROSPECT-4SAIL) derives LAI, fg as well as pigments used for leaf albedo estimation. The two first parameters could be obtained from CLMS LAI/fAPAR (biophysical) products but the pigments could not (lines 599-601). While the green LAI derived

with biophysical processor agrees quite well with the CLMS LAI (Figure 17, middle panel), the fg derived with both methods are quite different (Figure 17, left panel) with fg from biophysical processor better capturing the expected behavior (Figure 18 and lines 627 – 638). This does not imply that the CLMS LAI is less accurate, just that we needed to use a simplified method to derive fg with the CLMS LAI and FAPAR products. Therefore, since we needed to derive the pigments and fg with biophysical processor we used it also to derive LAI.

**Validation:** Although the authors use a large number of EC stations for the validation, additional evidence is required for the statement "The CLMS ETa prototype also compared favourably with the global WaPOR ETa maps produced by FAO, which it is meant to replace and other higher-resolution ETa datasets (Section 4.1). The addition of ETa product in the CLMS portfolio should therefore significantly enlarge the CLMS user community" Except for figure 16, the paper does not show how datasets compare for larger areas (spatial patterns).

Thank you for this valid suggestion. For the revised manuscript, we will conduct a more thorough intercomparison of spatial patterns in the WaPOR and CLMSE ET maps.

**Model vs framework**: To improve clarity, I would advise to distinguish more explicitly between the model (the algorithms) and the framework (the processing system including input selection, gapfilling, and temporal aggregation). The paper would benefit from making this distinction as it helps to understand the design choices. For example, in the sentences *"Preparatory activities required to develop an operational CLMS ETa product recommended that two ET modelling frameworks should be further investigated. The first one is the Sen-ET framework (Guzinski et al., 2020, 2021) developed to model ETa with Copernicus data at various spatial scales and using the Two-Source Energy Balance Priestley-Taylor (TSEB-PT) ET model (Norman et al., 1995; Kustas and Norman, 1999; Anderson et al., 2024). The second is the WaPOR framework developed by FAO through the WaPOR project and using the ETLook ETa model (Bastiaanssen et al., 2012). Both models, although conceptually different, estimate evaporation and transpiration and use LST as one of core input forcings."* The reference should be to the WAPOR ETLook model instead of the WaPOR framework, as the approach is different from the WaPOR modelling framework with regards on input selection, gapfilling and temporal aggregation.

Thank you for pointing this out. We will review the whole manuscript to make this distinction clearer.

SPECIFIC COMMENTS

**Introduction**

30 - "Since actual evapotranspiration is a direct proxy of plant water use it can be utilized for consistent irrigation water use monitoring across natural and political boundaries": Since distinguishing between rainfall and irrigation water use remains a challenge, please clarify this limitation or remove the specific reference to irrigation monitoring.

Yes this is true and we will change "irrigation water use monitoring" to "agricultural water use monitoring".

50 - "In order to satisfy this wide range of potential users' needs, and for consistency with other global CLMS products, the CLMS ETa product will have a spatial resolution of 300 m and a dekadal temporal resolution." It is not entirely clear to which users the 300m product caters?

Different applications which could benefit from ETa dataset are mentioned on lines 32-39 with further applications of WaPOR data mentioned on lines 43-45. When performing analysis on global or (large) regional extent the resolution of 300 m is in many cases sufficient. Those are the user needs referred to in the sentence. However, we will try to modify this statement to avoid confusion. In addition, we will add a paragraph in the introduction explaining the constraints imposed on the global CLMS products, including the 300 m resolution.

55 - "Another operational and global product which utilizes MODIS and VIIRS data is produced by United States Geological Survey using SSEBop energy balance model (Senay et al., 2020) with dekadal temporal resolution and 1 km spatial resolution." Consider mentioning FEWS as the dataset is available there.

Thank you for pointing this out, we will mention FEWS in the revised manuscript.

**Data and methods**

Table 2: Perhaps specify which inputs are used for which model? Personally I think a figure showing how these inputs are used to generate the model inputs (e.g. LAI, albedo) would give more insight. I assume "100m" in the weather data means "at 100m above the surface" and not to the spatial resolution – this may be made more clear, or removed.

Thank you for the suggestion. In the revised manuscript we will make it clearer which inputs are used by which model, either by modifying the table, adding a figure or explanatory text. And indeed, 100m refers to 100m above the surface and this will also be clarified.

110 - "same or similar values in both cloudy and sunny conditions (e.g. leaf area index does not change day to day depending on cloudiness). Therefore, gaps in this data are highly suitable for filling using spatio-temporal gap-filling" I understand this makes the data suitable for temporal gap-filing, but it does not automatically make it suitable for spatial gapfilling**?**

The second sentence refers to the full first sentence, the first half of which ("The reflectance

values, as well as biophysical traits, usually show a clear seasonal cycle and spatial similarity ")
is on the previous page (line 107) and therefore could have been missed. The spatial similarity
makes the data suitable for spatio-temporal gap-filling (such as StarFM which was referred to)
which is different to spatial gap-filling (such as interpolation between known data).

121 - Please also introduce View Zenith Angle (VZA) in the text (it is currently only in the captions).
Yes, we will

270 - "More details and the list of evaluated indices are available in the WaPOR wiki
(https://bitbucket.org/cioapps/wapor-et-look/wiki/Intermediate_Data_Components/LST, last
accessed: 22/07/2025)" : This repository recently moved to https://github.com/un-fao/wapor-et-
look , consider updating.
Thank you for pointing this out, we will update accordingly.

275 - "Finally, since we do not expect strong influence of aspect and slope on LST those two
variables were removed from the WaPOR list and the resulting combination of 9 variables (called
"DMS - WaPOR selected" in Section 4.3.2) is used in the ETa processing chain to sharpen the 1
km Sentinel-3 LST to the required 300 m spatial resolution." On what did you base this
expectation?
From physical considerations, the slope and aspect influence LST mainly through the solar
illumination geometry and this is already partially taken into account in the cosine solar
inclination angle. We will add this clarification on lines 275-276 in the revised manuscript. In
addition, we performed evaluation of LST sharpening using all WaPOR explanatory variables and
selected WaPOR explanatory variables (i.e. without slope and aspect) which is presented in
Section 4.3.2 and Table 10. Our expectation was based on this analysis.

255 - "ETLook model (Bastiaanssen et al., 2012) is used in the WaPOR framework and is
described in detail in Section 5 of "WaPOR Data Manual, Evapotranspiration v2.2" (FRAME
Consortium, 2020). ": Please note that the WaPOR data manual refers to the ETLook version 2,
and mostly describes how the inputs are derived, while the methodology (the model) used in v2
is described in the methodology document
(https://openknowledge.fao.org/server/api/core/bitstreams/d3db4794-fb5b-444c-9b3a-
c5fb154c5f9f/content). For version 3 the data manual and methodology documentation were
combined, with all updates and changes described in the Github page.
Thank you again, we will also update this accordingly.

**3 Prototype product validation**

Figure 5: The lack of Eddy Covariance stations outside Northern America, Europe and Australia is an issue, with Africa, Asia and South America only represented by a few stations. This is mentioned in the text, and counterargued with that all major climate zones and plant functional types are represented by at least one EC station. In figure 5 you do show the number of dates available for each PFT and climate zone, but could you add the number of stations as well? I think that would improve our insight in which areas are still underrepresented in EC datasets.

Figure 5 actually shows the number of sites available in each PFT and climate zone. We will increase that number for the revised manuscript as we are evaluating new sites in South America, Africa and Asia. Moreover, the analysis will cover a longer period.

400 - Temporal aggregation smooths errors. Why did you choose to validate the dekadal computations?

It is true that temporal aggregation smooths errors. However, the CLMS ET product has dekadal timestep which is why we validated the data at the same timestep.

440 - "Missing data during the day were computed by linear interpolation if the number of valid timeslots during daytime was at least 50% of the total number of timeslots in that period. Otherwise, the day was discarded." This means you are interpolating both inputs and outputs, so I would mention that this interpolation may smooth variability and can influence error metrics.

The interpolation described in this paragraph does not refer to model input or output. It was applied to a reduced number of eddy covariance data that were only available at sub-daily time step. The procedure is intended to ensure that there is a minimum number of actual *in situ* observations in a day before sub-daily values are aggregated to estimate daily ET. We will try to make this clearer in the revised manuscript.

470 - "Conversely, the models performed less good in the Tropical and Dry regions." I would not attribute the poorer performance to the models themselves as the causes are likely input related. In tropical regions, frequent cloud cover will result in missing remote sensing data inputs, while in dry regions it may be a result of missing short-term ET peaks after rainfall.

We will change this to "model outputs" which captures both the models and their input data.

455-485 – The text describing the figures does not describe the WAPOR outputs while they are in the figures. The comparison with WaPOR is available in the discussion section. But since WaPOR is also based on the ETLook model, but uses other inputs, this would be an excellent opportunity to assess the impact of different inputs (sensors, datasets) and different input timesteps (daily vs decadal).

Since WaPOR ET is not a product of this study, we do not think that it should be analysed in the results section. This could be part of another study in which spatio-temporal evaluation of

different products could be assessed in detail (e.g.CLMS-ET, WaPOR, MOD16, GLEAM, LSA-SAF…). At the same time, we put it in the figures to avoid duplication and to make the analysis performed in the Discussion easier. In the revised manuscript, we will either add a clarification on line 455 that WaPOR shown in Figures 6 – 11 will be discussed further in Section 4.1 or remove WaPOR from those plots. Direct use of WaPOR and CLMS ETLook outputs to assess the different inputs might not be straightforward for the reasons mentioned in Section 4.1 (i.e. reanalysis versus NRT mode and that not all temporarily-static, spatially distributed parameters used in WaPOR are public [note that in the revised manuscript we will used new CLMS ETLook data obtained with the use of the dry bare soil and wet-vegetated surface albedo layers and also other parameterization changes and we expect the results to be much closer to WaPOR maps]) but we will try to expand on this discussion. Finally, CLMS uses daily input timestep and the aggregation to dekadal is performed on the output ET maps, as will be clarified in Section 2.9.

Figure 8: I would add the climate region to the individual plots (instead of A, B, C and D).
We will modify the figure accordingly.

Figure 8/9: I would also add the number of sites used for each figure. Or the number of data points. Now they seem to have the same importance while some are based on more data points. This is a valid point. We do show the number of sites per climatic zone and PFT in Figure 5 and we also mention it when discussing the results (e.g. lines 473-474). However, in the revised manuscript we will make sure that this is clearer.

Figure 10: The reason for selecting the specific validation sites is not fully explained. If these sites are selected to illustrate the difference between the two models, this should be made explicit. The differences between the models (in particular T) requires further discussion. Moreover, I have some concerns regarding the choice for EBF (evergreen broadleaved forest) and DBF (deciduous broadleaved forest) sites as evapotranspiration modelling of forests is rather complicated for any ET model. For readability I would repeat the abbreviations like EBF more often, in particular in figures like figure 10.
The sites shown in Figure 10 and 11 were chosen precisely because modelling of ET in forest is complicated and for this reason the two models gave quite divergent estimates (line 486-489). We agree however that this should be made clearer and we will address this in the revised manuscript. We will also add another section to the discussion in which we look in more detail at the performances of the two models in different ecosystems and climates. Finally, in the revised manuscript we will use updated parameterization of the two models which is expected to reduce the differences between them in the forests.

**Discussion**

525 - The discussion on the differences between WaPOR Eta and CLMS ETLook Eta is very thorough.

Thank you

538 - "While both CLMS ETa and WaPOR ETa rely on DMS to improve the spatial resolution of LST, the original LST in CLMS is acquired by SLSTR sensor on board Sentinel-3 satellite with 1 km spatial resolution, while the original LST in WaPOR (version 3) is acquired by the VIIRS sensor on board of Suomi-NPP satellite with 375 m spatial resolution." => WaPOR L1 does not use DMS as VIIRS LST has a spatial resolution of 375m, and DMS would only introduce errors. DMS is only used for WaPOR L2 and L3.

Thank you for the clarification, we will correct the text in the revised manuscript.

543 - Regarding point 2 (WaPOR being a reanalysis product) I have one remark: WaPOR is produced both NRT and after 6 dekads reprocessed. See also https://github.com/un-fao/wapor-et-look/wiki/Understanding%20the%20WaPOR%20Pipeline#wapor-database

This is an imprecise statement from our side. What we meant is that the WaPOR ET data used by us was from the reanalysis dataset because it came from year 2020. This will be corrected in the revised manuscript.

545- Regarding point 3: the tenacity factor of WaPOR ETLook is 2: See https://github.com/un-fao/wapor-et-look/wiki/Release%20Notes & https://github.com/un-fao/wapor-et-look/wiki/Relative%20Root%20Zone%20Soil%20Moisture.

Thank you for pointing this out. In the revised manuscript we will also use CLMS ETLook data produced using maps of surface albedo of full vegetation and dry bare soil provided by the WaPOR consortium and other WaPOR-ETLook parameterization and we expect the results to be more consistent with WaPOR ET.

581 - "The examples in Figure 16 show as well that the number of missing data in the output maps of ETLook and TSEB-PT is larger than in the WaPOR product. The reason for those gaps are the differences between NRT and reanalysis gap-filling (see Section 4.1) but also the different model inputs and treatments of inland water and snow" This explanation should be expanded to include the differences in gapfilling the inputs or outputs. ,

We will mention the WaPOR gapfilling approach in this discussion.

730 - "This situation should be resolved by the end of the decade when Land Surface Temperature Monitoring (LSTM) mission, with a primary objective of frequent monitoring of fieldscale ETa, will join the Copernicus constellation (Koetz et al., 2018). " Is this approach realistic for an operational global product?

Producing an operational global product at ET 50 m (if that becomes the final spatial resolution of LSTM) might not be realistic but the LST could be resampled to 300 m in which case the need for sharpening of Sentinel-3 LST would be avoided.

Thanks!

Annemarie Klaasse

---

## Author Comment (AC4)

Dear Reviewer,

Thank you for your constructive feedback. We provide our response below each comment in green font.

This paper describes the development of Copernicus Land Monitoring Service (CLMS) ETa data products and validation of the prototype product. I appreciate the authors' transparency in documenting these choices and opening the discussion on EGUsphere. However, the paper reads more like a combination of Algorithm Theoretical Basis Document and Product Quality Assessment Document, which should be published with CLMS data products anyway. Therefore, I'm questioning the scientific significance of this paper.

Although there is necessarily an overlap between some of the information content of this paper and ATBD and PQAD we do not think this detracts from the scientific contribution of this manuscript. The different documents have a different focus and while ATBD and PQAD are about informing the user of the product, this manuscript aims to focus on the scientific context of the product, e.g. through the content presented in Introduction and Discussion sections. For example, we believe that there is a clear scientific interest in exploring the selection and pre-processing of the various input data as the same inputs can be used in the other models. This is especially relevant since the data is free and open (coming mainly from Copernicus) and pre-processing methods are publicly available as open-source Python packages (see Code Availability section). Similarly, looking at the individual and ensemble outputs of just TSEB and ETLook models can lead to improved scientific understanding of the models themselves, but also to potential improvements in overall modelling of land surface water and energy fluxes.

General comments

1. I suggest the authors focus on specific research questions, rather than merely presenting the paper as "describing the design choices". Some lengthy details could be included in Annex to keep the focus on such research questions. For example, the paper could focus on investigating

   Perhaps a better phrase here would be "describing and justifying the design choices" since we not only described the methods used to prepare the various input data but also evaluate their performance and compare with alternatives (Section 4.3)

• Why the average of two models perform better than individual model?

• Where and when the ensemble average neutralizes biases of two models?

• What are the uncertainties and gaps in input data and processing that could influence model performance?

Those are all very valid suggestions and we will add a new section in the discussion where we analyze the performance of the two individual models and the ensemble in different plant functional types and climates and link that to the different model assumptions and limitations and uncertainties in input data.

2. The paper lacks a clear definition of ETa. You referred to Bojinski et al. (2014) for the definition of ECVs. However, in that article, "water use" was mentioned instead of "actual evapotranspiration". Table 1 shows that CLMS products do not include evaporation from canopy interception. Since interception contribution to total evaporation is substantial (even more than soil evaporation in many cases) (Savenije, 2004), and very an important process to hydrology (Dingman, 2015), I think it should not be neglected by CLMS.
The reviewer is right that water stored in the canopy by interception can sometimes be significant. However, some challenges are still open when estimating the evaporation in the canopy, such as i) consideration of not only rainfall but also dew, fog and overhead irrigation, which the latter ones are very difficult to use as inputs operationally, ii) there is no general model/framework for accounting for the amount of interception, LAI usually plays a role in such semi-empirical approaches, but also leaf size/shape as well as wind speed and rainfall intensity should be considered, furthermore iii) there is a lack of interception data that could be used to validate these models covering a wide range of canopies and climates, iv) the separation between soil evaporation, canopy transpiration and evaporation of stored water in the canopy is challenging (and in most cases missing) for in-situ reference EC datasets, and v) efforts to incorporate evaporation from intercepted water in the canopy, such as those implemented in GLEAM, ETMonitor or WaPOR assume that all available energy at the canopy is used to evaporate all intercepted water, without considering that in order for evaporation to happen the air must be dry enough (i.e. VPD also plays a role according to the Penman equation: $E = \frac{\Delta R_n + \rho_a c_p VPD}{\lambda(\Delta + \gamma) r_a}$).
The latter issue implies that under a moist atmosphere, sensible heat flux can even be more significant than latent heat and thus temperature may increase in a canopy with intercepted water.

Furthermore, evaporation from canopy interception is implicitly integrated in the evaporation component on the energy balance, when using thermal infrared data as input, as this excess evaporation reduces the surface/canopy temperature. Should this process be considered robustly in energy balance models, a novel multi-source energy balance should be developed, which is not part of the scope of this manuscript.

3. Since canopy transpiration [mm/day] and soil evaporation [mm/day] are also CLMS products, effort should be made to evaluate their quality. You mentioned in L407 that these could not be contrasted against *in situ* However, there are references that could be used as proxy, e.g., partition of Eddy Covariance measurements (Nelson et al., 2020), and recently SAPFLUXNET (Poyatos et al., 2016). The ETLook model overestimates transpiration (compared to TSEB-PT) created large bias in ETa for specific cases, as you showed. The reason could also be that ETa from EC tower is underestimated due to energy balance closure (which was not discussed). In any case, investigating the accuracy of both models in transpiration estimation could add significant values in the discussion.

Regarding the partition of evapotranspiration into its components, we agree that it would be an interesting exercise, however we do not think that it can be performed at a globally representative scale including the different climatic zones and plant functional types. The EC flux partitioning methods have rather high uncertainty, often require raw high frequency data (which is not usually available at higher level products of ICOS, Ameriflux and OxFlux sites) or site-specific data preprocessing and selection. Regarding the sap-flow measurements, there is the issue of spatial-scale mismatch when comparing against 300 m transpiration. For that reason, in the Product User Manual we specify that E and T partition is unvalidated and therefore has a larger uncertainty and we will add similar warning to the manuscript in the Discussion / Conclusion. We do however hope that once the dataset is public it will be used in various studies around the globe and the characterization of the partition will become more certain.

Regarding energy balance closure we briefly discuss it on lines 411-420. However, in the revised manuscript we will make it more explicit. The validation results could potentially change depending on the energy balance closure (EBC) method and variants. In order to ensure transparency and minimize any scientific bias by us, we prioritize the use of the energy balance correction provided by the dataset, which usually is the Bowen Ratio method adopted by the FLUXNET community (Pastorello, 2020) and implemented in different datasets: ICOS, Ameriflux, OzFlux. In other stations we leave the flux uncorrected. How to deal with the EBC when validating ET models is still an open question for scientific debate.

References:

1. Pastorello, G., Trotta, C., Canfora, E., Chu, H., Christianson, D., Cheah, Y.-W., Poindexter, C., Chen, J., Elbashandy, A., Humphrey, M., et al. (2020). The FLUXNET2015 dataset and the ONEFlux processing pipeline for eddy covariance data. Sci Data 7, 225. https://doi.org/10.1038/s41597-020-0534-3.

4. The validation of the prototype is most limited in the tropical climate (Figure 5). The validation results also show lower performance than other regions. Meanwhile, the number of cloud-free images are lowest in this climate (Figure 2), even limiting eastern swath does not change much this fact. This points to a serious gap in a global data product like CLMS, which should be highlighted.

   Thank you for pointing this out. For the revised manuscript we are performing additional validation in South America, Asia and Africa and we will also highlight this limitation in the Discussion/Conclusion.

Specific comments

1. L14 "overall best" => "average" (of all site-dekads)

   What we meant by "overall best" was best out of TSEB-PT, ETLook and Ensemble. We will reword to make this clearer.

2. L16 which are the similar global ETa dataset mentioned?

   It is the WaPOR dataset. We will also make this clearer.

3. L21 What is the definition of "operational" in this context? Does it mean "near-real time monitoring" or something else?

   Copernicus definition of operational is (paraphrased) produced at regular intervals in near-real-time using well established and validated methods and input data sources and with guaranteed long-term continuity. It also means that the products largely satisfy the applicable requirements, that their limitations are non-critical and documented, and that regular quality monitoring and assessments are performed to check and ensure the stability of product quality. We will define it properly in the revised manuscript.

4. L45-47 Could you provide some references?

   This is based on official and unofficial discussions with FAO and therefore does not have references. FAO is aware of this manuscript.

5. L48 What are the other global CLMS products? What are their spatial and temporal resolution?

   Some example of those products are top-of-canopy reflectance (300 m, daily), vegetation products (Leaf Area Index, Fraction of Absorbed Photosynthetically Active Radiation, NDVI - 300 m, dekadal) and primary production products (GPP, NPP, Gross Dry Matter Productivity - 300 m, dekadal) - https://land.copernicus.eu/en/products/vegetation. We will mention this in the revised manuscript.

6. L48-L59 This paragraph seems to justify the spatial and temporal resolution of CLMS (300m, 10-d). If continuing WaPORv3L1 is the main factor, it should be expressed more explicitly. Other products can be presented as reference for state-of-the-art.
In this paragraph we present examples of ET datasets (known to us) which are produced operationally and in near-real-time with a global extent or satisfy at least some of those conditions. This is done to place CLMS ET product in the context of other, similar products and not to justify the spatial and temporal resolutions. Those resolutions were determined (requested) by European Commission and were the constraints within which we conducted the study and were based on a number of factors:
- Other CLMS products have the same spatial and/or temporal resolutions.
- 300 m is the spatial resolution of the Sentinel-3 OLCI sensor
- The manuscript describes design of a global product which aims to have long-term continuity. Increasing the spatial or temporal resolutions would have an impact on the data storage and distribution costs.
 The fact that WaPORv3L1 and CLMS spatio-temporal resolutions align is a coincidence or could point to WaPOR being influenced by existing CLMS products at the time its L1 product was being defined.
We will improve the justification for the spatial and temporal resolutions in the revised manuscript.

7. L60 What are these preparatory activities? Who recommended two ET modelling frameworks? Why were these two selected?
We will modify the paragraph on lines 60-65 to the following:
Preparatory activities conducted by European Commission to develop an operational CLMS ETa product recommended that two ET modelling frameworks should be further investigated. The first one is the Sen-ET framework (Guzinski et al., 2020, 2021) developed to model ETa with Copernicus data at various spatial scales and using the Two-Source Energy Balance Priestley-Taylor (TSEB-PT) ET model (Norman et al., 1995; Kustas and Norman, 1999; Anderson et al., 2024). The second is the WaPOR framework (FRAME Consortium & FAO, 2024) developed by FAO through the WaPOR project and using the ETLook ETa model (Bastiaanssen et al., 2012). Both models, although conceptually different, estimate evaporation and transpiration and use LST as one of core input forcings. This recommendation was mainly based on the availability of mature open-source implementations of the two ET models, on previous studies demonstrating the applicability of both models with Copernicus data sources (i.e. Sentinel-3 imagery and meteorological data from European Center for Medium Range Weather Forecasts) (Guzinski et al., 2025) and on FAO's familiarity with both approaches. This does not imply

that those two modelling frameworks clearly outperform all other approaches and indeed it has been shown that the performance of an individual model depends on the landcover and climatic conditions (Reitz et al, 2025). However, due to constraints on developing a publicly funded global and operational dataset (outlined previously) there was a need to limit the design of the first version of the CLMS ETa product to those two frameworks.

References:

1. Guzinski, R., Nieto, H., Sandholt, I., and Karamitilios, G. (2020). Modelling High-Resolution Actual Evapotranspiration through Sentinel-2 and Sentinel-3 Data Fusion. Remote Sensing 12, 1433. https://doi.org/10.3390/rs12091433.

2. Guzinski, R., Nieto, H., Sánchez, J.M., López-Urrea, R., Boujnah, D.M., and Boulet, G. (2021). Utility of Copernicus-Based Inputs for Actual Evapotranspiration Modeling in Support of Sustainable Water Use in Agriculture. IEEE Journal of Selected Topics in Applied Earth Observations and Remote Sensing 14, 11466–11484. https://doi.org/10.1109/JSTARS.2021.3122573.

3. Norman, J.M., Kustas, W.P., and Humes, K.S. (1995). Source approach for estimating soil and vegetation energy fluxes in observations of directional radiometric surface temperature. Agricultural and Forest Meteorology 77, 263–293. https://doi.org/10.1016/0168-1923(95)02265-Y

4. Kustas, W.P., and Norman, J.M. (1999). Evaluation of soil and vegetation heat flux predictions using a simple two-source model with radiometric temperatures for partial canopy cover. Agricultural and Forest Meteorology 94, 13–29. https://doi.org/10.1016/S0168-1923(99)00005-2.

5. FRAME Consortium & FAO: WaPOR methodology, https://github.com/un-fao/wapor-et-look/wiki, 2024.

6. Bastiaanssen, W.G.M., Cheema, M.J.M., Immerzeel, W.W., Miltenburg, I.J., and Pelgrum, H. (2012). Surface energy balance and actual evapotranspiration of the transboundary Indus Basin estimated from satellite measurements and the ETLook model. Water Resources Research 48. https://doi.org/10.1029/2011WR010482.

7. Reitz, M., Volk, J.M., Ott, T., Anderson, M., Senay, G.B., Melton, F., Kilic, A., Allen, R., Fisher, J.B., Ruhoff, A., et al. (2025). Performance Mapping and Weighting for the Evapotranspiration Models of the OpenET Ensemble. Water Resources Research 61, e2024WR038899. https://doi.org/10.1029/2024WR038899.

8. L66 Related to my first general comment, if the paper only "aims to give an overview of the design choices", it could be just a technical documentation. To make a clearer scientific significance, this paper should focus on investigating further the two model frameworks (L61).

We agree and will therefore add a section in the discussion focusing on the two modelling frameworks (as mentioned previously). However, we would again like to highlight that we believe that describing, and justifying, the forcing pre-processing step of a global ET product has scientific value in itself.

9. Section 2. I would start with 2.8 first to provide an introduction about the two modelling frameworks, clearly indicate which input variables are required for both/each framework. Then continue with 2.1 and link required variables with input data sources in Table 2, indicating which one is used in which model.
   Thank you for this suggestion, we will reformat this section as suggested.

10. Table 2. How were 100m wind speed, air temperature, and water vapour pressure converted to near surface level (e.g., 2m is commonly used).
    In our model setup we use the wind speed, air temperature and water vapour at 100 m above the surface. Due to the low spatial resolution of those fields we assumed that they represent conditions at atmospheric blending height (set to be 100 m above ground) where impact of local surface conditions on those parameters is not so direct (Guzinski et al., 2021).
    References:
    1. Guzinski, R., Nieto, H., Sánchez, J.M., López-Urrea, R., Boujnah, D.M., and Boulet, G. (2021). Utility of Copernicus-Based Inputs for Actual Evapotranspiration Modeling in Support of Sustainable Water Use in Agriculture. IEEE Journal of Selected Topics in Applied Earth Observations and Remote Sensing 14, 11466–11484. https://doi.org/10.1109/JSTARS.2021.3122573.

11. L81 Is this in the mandate of CLMS?
    The mandate of CLMS and Copernicus is (from lines 81-82) "to ensure the free and open license conditions, long-term future continuity and consistency across the CLMS portfolio". One way of achieving this is to base input data predominantly on Copernicus data sources.

12. If Copernicus data product should be based on Copernicus data, why the canopy height map was derived from a static map by ETH Zurich and not derived from other Copernicus data dynamically (e.g., Sentinel-2)?
    The ETH Zurich product is based on GEDI LiDAR measurements combined with Sentinel-2 imagery. There is no Copernicus equivalent of GEDI and, therefore, in this case there was no alternative. However, the tree canopy height is an ancillary product that remains largely static (in a global perspective) and we further filter it through the CLMS landcover product

(lines 313-317). We should clarify thus that canopy for herbaceous and annual canopies are dynamic and estimated based on LAI.

13. Table 3. The native resolution of these input data sources should be included.
Thank you for the suggestion. We will add this information to the table.

14. L104 missing reference for semi-Bayesian approach
The reference will be added:
Bulgin, C.E., Sembhi, H., Ghent, D., Remedios, J., & Merchant, C.J. (2014). Cloud Clearing Techniques over Land for Land Surface Temperature Retrieval from the Advanced Along Track Scanning Radiometer. International Journal of Remote Sensing, 35, 3594-3615

15. L108 What if LAI changes in case of crop harvesting, forest fire?
All gap-filling approaches have their limitations and this is certainly one of them. But in large majority of cases vegetation follows the phenological cycle and this is frequently used during gap-filling (e.g. the popular TIMESAT method or Whittaker smoother used in WaPOR).

16. L109 Can the gaps in LAI be suitable for filling in areas where it is cloudy most of the year?
Similarly to previous answer, gap-filling over long gaps or with few data points is for sure challenging and in general the data quality decreases with longer gap periods.

17. L112 needs reference
We will add following reference:
Abbasi, B., Qin, Z., Du, W., Li, S., Fan, J., and Zhao, S. (2020). Effects of Cloud on Land Surface Temperature (LST) Change in Thermal Infrared Remote Sensing Images: a Case Study of Landsat 8 Data. In IGARSS 2020 - 2020 IEEE International Geoscience and Remote Sensing Symposium (IEEE), pp. 5430–5433. https://doi.org/10.1109/IGARSS39084.2020.9324415.

18. L185 needs references. Not clear to what degree of accuracy, and from which EO data.
We will rephrase this sentence. We just wanted to stress that both albedo emissivity and LST are routinely estimated with EO data, but we should never forget that even these rather mature products are not free for uncertainties as any other remote sensing variable

19. L187 How do these models differ in Ln calculation?
In the TSEB-PT model, Ln is computed according to equation 2a from Kustas & Norman (1999) using CAMS surface_thermal_radiation_downwards field together with internally computed leaf and soil temperatures, constant emissivity values for the vegetation and soil (0.98 and 0.95 respectively) and effective leaf area index. WaPOR-ETLook estimates it

following the FAO-56 approach and daily values of air temperature, vapour pressure and transmissivity (FRAME Consortium & FAO, 2024).

References:

1. Kustas, W.P., and Norman, J.M. (1999). Evaluation of soil and vegetation heat flux predictions using a simple two-source model with radiometric temperatures for partial canopy cover. Agricultural and Forest Meteorology 94, 13–29. https://doi.org/10.1016/S0168-1923(99)00005-2.

2. FRAME Consortium & FAO: WaPOR methodology, https://github.com/un-fao/wapor-et-look/wiki, 2024.

20. L2555 needs reference for "most approaches"

Here we refer to DMS, TsHARP and other similar methods (see e.g. Gao et al., 2012; Agam et al., 2007; Sanchez et al, 2020). We will add those references to the revised manuscript.

References:

1. Gao, F., Kustas, W.P., and Anderson, M.C. (2012). A Data Mining Approach for Sharpening Thermal Satellite Imagery over Land. Remote Sensing 4, 3287–3319. https://doi.org/10.3390/rs4113287.

2. Agam, N., Kustas, W.P., Anderson, M.C., Li, F., and Neale, C.M.U. (2007). A vegetation index based technique for spatial sharpening of thermal imagery. Remote Sensing of Environment 107, 545–558. https://doi.org/10.1016/j.rse.2006.10.006.

3. Sánchez, J.M., Galve, J.M., González-Piqueras, J., López-Urrea, R., Niclòs, R., and Calera, A. (2020). Monitoring 10-m LST from the Combination MODIS/Sentinel-2, Validation in a High Contrast Semi-Arid Agroecosystem. Remote Sensing 12, 1453. https://doi.org/10.3390/rs12091453.

21. L269 needs reference

We will add the missing reference:

FRAME Consortium & FAO: WaPOR methodology, https://github.com/un-fao/wapor-et-look/wiki, 2024.

22. L276 Is this statement based on sensitivity analysis? I would expect aspect and slope influence shaded relief (related to shadow effect, intensity of sunlight exposure), which influence LST. (e.g., Peng et al., 2020)

This is true, but this is already account for in the "cosine solar inclination angle" parameter which is derived from aspect and slope. We will add this clarification in the revised manuscript. This is backed up by analysis shown in section 4.3.2.

23. L291 Only topographic correction?

Yes, only topographic correction of the meteorological variables shown in Table 6 was performed. However, this results in meteorological forcings suitable for evapotranspiration modelling as shown in Figure 22.

24. L313 resampled by averaging?

Yes by averaging, we will make this clear in the revised manuscript.

25. L398 How close is the closest non-gap-filled dates? Do you apply a threshold for when the date is too far?

The closest gap-filled date varies by location and season and can range from 1-day to 60-days (line 394).

26. L416 Although it is mentioned here that you searched for more diversity in the reference datasets, only Eddy Covariance methods are considered. Meanwhile, many other references have been used to (qualitatively) validate remotely sensed ET, especially in regions lacking Eddy Covariance measurements (Tran et al., 2023)

Please see reply to comment number 27.

27. For example, a large area in Figure 4 has no stations at all. These areas would require alternative effort and methodology to evaluate CLMS data products. For examples, see de Andrade et al. (2024) for South America, see Weerasinghe et al. (2020) and Blatchford et al., (2020) for Africa, see Athira et al. (2025) for India, see He et al. (2025) Southeast Asia, see Cogill et al. (2025) for South Africa. I think at least, the authors should refer to these studies for future validation of CLMS data products.

Thank you for the reference. It is true that many areas do not have Eddy Covariance (EC) measurements but the focus of qualitative validation was specifically on this data as it is the most widely used for validation of satellite-based ET estimates. Also please note that in the revised manuscript, we will add extra EC sites in South America, Africa and Asia. In addition, we will mentioned the suggested references when discussing the limits of validation in the revised manuscript.

28. L440 Which quality flag was excluded? And what was considered realistic? Based on what?

No quality flag was excluded. On the contrary, the quality information provided with each dataset was effectively used to ensure the selection of good quality data. The point on realistic range refers to cases in which missing data are filled with numerical values (*e.g.* -999.9) that are meant to indicate that the timeslot does not contain a valid value. The occurrence of such values goes normally together with a bad quality score in the quality

flag but using both approaches gave an additional warranty of good data selection. We will clarify this in the revised manuscript.

29. L446 What I missed here is whether ETa values were extracted at the 300-m pixel containing the station or flux footprint was considered. The mismatch in spatial support although not affect overall results but would affect some cases where the flux site is located in heterogenous area (e.g., IT-BCi).
The ETa values were extracted from the 300m pixels containing the station. Evapotranspiration validation is performed during the day, therefore under unstable conditions. In these cases, the footprint is generally limited to a few hundred meters or less. We will add this information to the revised manuscript. However, spatial scale mismatch can indeed affect some sites and this will be mentioned in the revised manuscript.

30. Table 8. rBias should also be included for site level
Thank you for this suggestion. Table 8 will be modified accordingly in the new version of the manuscript.

31. L462 The overall statistics in Appendix C would not change much, but in my experience, bias and rbias would differ remarkably for some sites that energy balance disclosure is high.
Yes this is true, here we were focusing on the overall statistics and we will mention this in the revised manuscript.

32. L481 "better than" what?
Better than individual models. This will be clarified in the revised manuscript.

33. L483 what do you mean by "the variability of the fluxes"? temporal or spatial variability?
We mean the standard deviation shown in Taylor plots of Figure 9.

34. L484 reference to Figure 7?
Again here we refer to the standard deviation shown in the SAV plot of Figure 9, where standard deviation of ETLook values is larger than in situ (variability is overestimated) while standard deviation of TSEB-PT is smaller than in situ (variability is underestimated). We will improve the clarity of this in the revised manuscript.

35. L486 Interesting. I think this should be investigated further in future development.
We agree that this should be investigated further since forests are generally complex ecosystems for which it is most difficult to model ET. Please note that the data which will

be used in the updated manuscript will be derived with TSEB-PT and ETLook models in which parameterization was updated and this is expected to reduce the magnitude of this difference.

36. Figure 10. Could you also plot the EB-corrected data for tower? What does the error band around "Tower" line represent?
The line with the label 'Tower' in that Figure is actually the EB-corrected data. That clarification will be added to the caption of the Figure in the revised version of the manuscript. The band around the line is the random uncertainty. The random uncertainty –when provided in the eddy covariance dataset- for a particular timeslot is derived from flux measurements with similar meteorological conditions within a time window of +/- 5 days around the date and within a window of +/- 1 hour around the time of the day, as described in Pastorello et al, 2020.
Reference:
1. Pastorello, G., Trotta, C., Canfora, E., Chu, H., Christianson, D., Cheah, Y.-W., Poindexter, C., Chen, J., Elbashandy, A., Humphrey, M., et al. (2020). The FLUXNET2015 dataset and the ONEFlux processing pipeline for eddy covariance data. Sci Data 7, 225. https://doi.org/10.1038/s41597-020-0534-3.

37. L491, ETLook is not visible in the Taylor plot (Fig9-EBF).
The figure will be corrected in the revised manuscript.

38. L496 Here it says the ensemble formula is a suitable option in the case of DBF (Figure 11). But in the case of EBF (Figure 10- IT-Cp2), the ensemble is worse than TSEB-PT, so would you say the ensemble formula is not applicable everywhere?
Yes we agree that ensemble does not perform better than the individual models in all sites. However, overall it still presents the most accurate results. It should be noted that in the CLMS ET product the user will have access to ET produced by ensemble and the individual models. We will add this information to the revised manuscript.

39. L525 The description of WaPOR global ETa dataset should be mentioned before the validation results (Section 3)
We did not consider comparison with WaPOR to be validation which is why this is only described in the Discussion. However, we decided to include WaPOR ET in the plots and tables of Section 3.2 not to duplicate the figures and to make the comparison easier. We agree that this might be confusing and therefore we will make a reference and brief description of WaPOR ET before Section 3.2 but discuss the comparison still in Section 4 or we will remove WaPOR results from the Tables and Figures presented in section 3.

40. L530 a low bias presents high accuracy, so I would say "performance" instead of "accuracy" here.

We will correct this in the revised manuscript.

41. L566 The selection of tiles seems arbitrary to me. I would suggest more strategic selection, focusing on tiles without flux sites, tiles with high heterogeneity (e.g., indicated by coefficient of variation of vegetation index), and tiles represent climatic zones (especially Tropical and Arid, since these are lacking validation sites).

Thank you for the suggestion, we will consider it when updating the figure for the revised manuscript.

42. Figure 15. Although it might make the histogram look crowded, I'd suggest including the ensemble.

Again thank you for the suggestion, we will take it into consideration.

43. Figure 17. What does ETA stand for?

On the y-axis are LAI, gLAI and fg derived with the CLMS LAI/FAPAR (labeled as CLMS) dataset and on the X from the biophysical model produced as part of the CLMS ET processing (labeled as ETA). We agree that this might be confusing and it will be corrected in the revised manuscript.

44. L618-619 This explanation is not clear to me. The uncertainty in LAI at higher LAI should affect ETLook model more since LAI is used to partition available energy for transpiration and soil evaporation in ETLook. I think saying that it has "minimal effect on ETa modelling" requires some local sensitivity analysis or uncertainty analysis.

LAI also affects the partitioning between E and T in the TSEB-PT model, in particular LAI is also used in the Campbell radiative transfer model to partition the (net) radiation into canopy and soil (see Eqs. 3a and 3b). Indeed, at high LAI values most of the gap fraction becomes negligible according to the exponential law of Beer-Lambert, and thus most of the partitioning is already assigned to the canopy and most of the water flux to T. Therefore, a change of LAI between e.g. 4 and 5 does not significantly affect this partitioning.

45. L628-630 I'm connecting this larger bias in LAI in dense vegetation (e.g., broadleaf forest) with the lower performance of ETLook in EBF and DBF discussed in L486-489. Maybe a simple local sensitivity analysis could help address this.

This could be part of the reason but LAI is equally important parameter for TSEB-PT model (see the answer above). We have recently discovered some parameterization issues in our implementation of ETLook which we think could be major contributors to this lower

performance. In the revised manuscript we will use data produced with updated parameterization.

46. L633-656 Similarly, the inconsistency of CLMS LAI at SAV site could contribute to why the r2 of ETLook is lower than 0.5 for SAV (Figure 7)
Please see the answer above.

47. L660 How were these AOIs selected? Based on which criteria? Why not selecting AOI based on level of heterogeneity/homogeneity?
This was part of the selection process, as was the aim to cover different climatic zones and ecosystems distributed around the globe. We will add this information to the revised manuscript

48. L672-L680 The site-level results were discussed but not presented or referenced.
The table with site-level results was not included in the manuscript for brevity but will now be added to supplementary material.

49. L690 Interesting. Could you show some examples? Where were these artifacts commonly observed? I think this would be important for users to know since the use of products like CLMS are often for spatial analysis.
We observed them e.g. in tropical rainforest with uniform dense vegetation. In the revised manuscript we will show an example in the Supplementary Material.

50. L696-L700 Do these 45 sites mainly in Europe also represent Tropical and coastal regions?
We will make a correction since there were 43 sites, but there are 3 sites in Europe closer than 25km to the Mediterranean (ES-Agu, ES-LJu, IT-Lsn), and 4 in US close to the Pacific (US-CGG, US-Myb, US-Sne and US-Snf). However, only one site is on tropical region AU-Dry (Aw Köppen climate)

[Figure]

51. Figure 22. Could you say something about the cluster of points with EC below 100 Wm-2 but spread from 100 to 350 Wm-3 for ETA?

Those points most probably represent situations with persistent (on a daily time-scale) local cloud/fog which was not representative of the conditions seen in the CAMS pixels with its 0.4° spatial resolution. However, it could also be due to dust, spider webs or other obstructions on the upward pointing radiation sensors mounted on the EC tower.

52. Overall Section 4.3 presents a lot of methodological details and could be more focused on the impact on the prototype products. I would suggest methods or trials of methods to be described in Methodology section or Annex.

In our opinion the trials of methods present some of the largest scientific input of this paper (apart from the ETa product itself), since they (and the lessons learned) can be applied to other global and regional products. This is especially true since the core of those methods are all open-source as listed in the "Code availability" at the end of the manuscript. However, we also agree that more emphasis should be placed on evaluating the strengths and weaknesses of the two models and the ensemble and therefore we will add a dedicated section on this topic to the discussion in the revised manuscript.

53. L733 ECOSTRESS could also be considered, then.

ECOSTRESS is for sure a very useful sensor for prototyping of different methods. However, with its irregular acquisition times, frequency and coverage of the globe and with its

limited duration on board of the International Space Station it is not suitable for global operational applications focused on long-term continuity.

54. L747 Also canopy interception

As mentioned in the response to comment number 1, interception is already implicitly captured in the evaporation component. Explicitly separating evaporation from canopy interception is scientifically challenging, however we will add it to potential improvements in the revised manuscript.

55. L761 Could you say that the pre-processing methods are "applicable globally" when the evaluation in Tropical climate (areas with least validation sites and most data gaps) was not included?

The selected methods rely on globally available data sources and do not undergo any localized fine-tuning or specific parameterization. In that sense they are applicable globally. However, we agree that the quality of the outputs is not well characterized globally and the models might perform better in some areas than others. We will add such statement to the revised manuscript. We would also like to point out that some stations in the tropical climate were already used (northern Australia and Indonesia) and in the revised manuscript we will use additional stations from South America, Africa and Asia.

**References**

Athira, K.V., Rajasekaran, E., Boulet, G., Nigam, R. and Bhattacharya, B.K., 2025. Multiscale evaluation of three remote sensing-based evapotranspiration models across the humid to arid tropics: a study over India. *Hydrological Sciences Journal*, pp.1-22.

Blatchford, M.L., Mannaerts, C.M., Njuki, S.M., Nouri, H., Zeng, Y., Pelgrum, H., Wonink, S. and Karimi, P., 2020. Evaluation of WaPOR V2 evapotranspiration products across Africa. *Hydrological processes*, *34*(15), pp.3200-3221.

Cogill, L.S., Toucher, M., Wolski, P., Esler, K.J. and Rebelo, A.J., 2025. Evaluating the performance of satellite-derived evapotranspiration products across varying bioclimates in South Africa. *Remote Sensing Applications: Society and Environment*, p.101612.

de Andrade, Bruno Comini, Leonardo Laipelt, Ayan Fleischmann, Justin Huntington, Charles Morton, Forrest Melton, Tyler Erickson et al. "geeSEBAL-MODIS: Continental-scale evapotranspiration based on the surface energy balance for South America." *ISPRS Journal of Photogrammetry and Remote Sensing* 207 (2024): 141-163.

Dingman, S.L., 2015. *Physical hydrology*. Waveland press.

He, S., Yang, Q., Zhang, L., Shi, Z., Wang, X. and Lv, B., 2025. Consistency Assessment and Uncertainty Analysis of Spatial-temporal Characteristics of Evaporation Data in the Greater Mekong Subregion. *Journal of Hydrometeorology*.

Nelson, J.A., Pérez-Priego, O., Zhou, S., Poyatos, R., Zhang, Y., Blanken, P.D., Gimeno, T.E., Wohlfahrt, G., Desai, A.R., Gioli, B. and Limousin, J.M., 2020. Ecosystem transpiration and evaporation: Insights from three water flux partitioning methods across FLUXNET sites. *Global change biology*, *26*(12), pp.6916-6930.

Peng, X., Wu, W., Zheng, Y., Sun, J., Hu, T. and Wang, P., 2020. Correlation analysis of land surface temperature and topographic elements in Hangzhou, China. *Scientific Reports*, *10*(1), p.10451.

Poyatos, R., Granda, V., Molowny-Horas, R., Mencuccini, M., Steppe, K. and Martínez-Vilalta, J., 2016. SAPFLUXNET: towards a global database of sap flow measurements. *Tree physiology*, *36*(12), pp.1449-1455.

Savenije, H.H., 2004. The importance of interception and why we should delete the term evapotranspiration from our vocabulary.

Tran, B.N., Van Der Kwast, J., Seyoum, S., Uijlenhoet, R., Jewitt, G. and Mul, M., 2023. Uncertainty assessment of satellite remote-sensing-based evapotranspiration estimates: a systematic review of methods and gaps. *Hydrology and Earth System Sciences*, *27*(24), pp.4505-4528.

Weerasinghe, I., Bastiaanssen, W., Mul, M., Jia, L. and Van Griensven, A., 2020. Can we trust remote sensing evapotranspiration products over Africa?. *Hydrology and Earth System Sciences*, *24*(3), pp.1565-1586.